# Construction of Hierarchical Neural Architecture Search Spaces based on Context-free Grammars

**Simon Schrodi**[1]    **Danny Stoll**[1]    **Binxin Ru**[2]
**Rhea Sanjay Sukthanker**[1]    **Thomas Brox**[1]    **Frank Hutter**[1]

[1]University of Freiburg    [2]University of Oxford

{schrodi,stolld,sukthank,brox,fh}@cs.uni-freiburg.de    robin@robots.ox.ac.uk

## Abstract

The discovery of neural architectures from simple building blocks is a long-standing goal of Neural Architecture Search (NAS). Hierarchical search spaces are a promising step towards this goal but lack a unifying search space design framework and typically only search over some limited aspect of architectures. In this work, we introduce a unifying search space design framework based on context-free grammars that can naturally and compactly generate expressive hierarchical search spaces that are 100s of orders of magnitude larger than common spaces from the literature. By enhancing and using their properties, we effectively enable search over the complete architecture and can foster regularity. Further, we propose an efficient hierarchical kernel design for a Bayesian Optimization search strategy to efficiently search over such huge spaces. We demonstrate the versatility of our search space design framework and show that our search strategy can be superior to existing NAS approaches. Code is available at https://github.com/automl/hierarchical_nas_construction.

## 1    Introduction

Neural Architecture Search (NAS) aims to automatically discover neural architectures with state-of-the-art performance. While numerous NAS papers have already demonstrated finding state-of-the-art architectures (prominently, e.g., Tan and Le [1], Liu et al. [2]), relatively little attention has been paid to understanding the impact of architectural design decisions on performance, such as the repetition of the same building blocks. Moreover, despite the fact that NAS is a heavily-researched field with over 1 000 papers in the last two years [3, 4], NAS has primarily been applied to over-engineered, restrictive search spaces (e.g., cell-based ones) that did not give rise to *truly novel* architectural patterns. In fact, Yang et al. [5] showed that in the prominent DARTS search space [6] the manually-defined macro architecture is more important than the searched cells, while Xie et al. [7] and Ru et al. [8] achieved competitive performance with randomly wired neural architectures that do not adhere to common search space limitations.

Hierarchical search spaces are a promising step towards overcoming these limitations, while keeping the search space of architectures more controllable compared to global, unrestricted search spaces. However, previous works limited themselves to search only for a hierarchical cell [9], linear macro topologies [1, 2, 10], or required post-hoc checking and adjustment of architectures [8, 11]. This limited their applicability in understanding the impact of architectural design choices on performance as well as search over all (abstraction) levels of neural architectures.

In this work, we propose a *functional* view of neural architectures and a unifying search space design framework for the efficient construction of (hierarchical) search spaces based on *Context-Free Grammars (CFGs)*. We compose architectures from simple multivariate functions in a hierarchical manner

37th Conference on Neural Information Processing Systems (NeurIPS 2023).

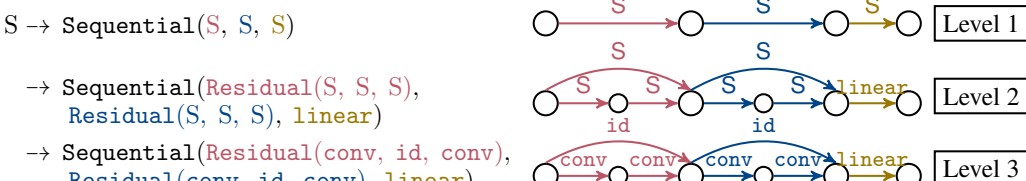

```
S → Sequential(S, S, S)

  → Sequential(Residual(S, S, S),
      Residual(S, S, S), linear)
  → Sequential(Residual(conv, id, conv),
      Residual(conv, id, conv), linear)
```

Figure 1: Derivation of the function composition of the neural architecture from Equation 1 (left). Note that the derivations correspond to edge replacements [14–16] in the computational graph representation (right). The intermediate derivations provide various granularities of a neural architecture. Appendix A provides the vocabulary of primitive computations and topological operators.

using the recursive nature of CFGs and consequently obtain a function composition representing the architecture. Further, we enhance CFGs with mechanisms to efficiently define search spaces over the complete architecture with non-linear macro topology (e.g., see Figure 4 or Figure 10 in Appendix J for examples), foster *regularity* by exploiting that context-free languages are closed under substitution, and demonstrate how to integrate user-defined search space *constraints*.

However, since the number of architectures scales exponentially in the number of hierarchical levels – leading to search spaces 100s of orders of magnitude larger than commonly used ones in NAS – for many prior approaches search becomes either infeasible (e.g., DARTS [6]) or challenging (e.g., regularized evolution [12]). As a remedy, we propose Bayesian Optimization for Hierarchical Neural Architecture Search (BOHNAS), which constructs a hierarchical kernel upon various granularities of the architectures, and show its efficiency through extensive experimental evaluation.

**Our contributions** We summarize our key contributions below:

1. We propose a *unifying search space design framework* for (hierarchical) NAS based on CFGs that enables us to search across the *complete* architecture, i.e., from micro to macro, *foster regularity*, i.e., repetition of architectural patterns, and incorporate user-defined *constraints* (Section 3). We demonstrate its *versatility* in Sections 4 and 6.

2. We propose a *hierarchical extension of NAS-Bench-201* (as well as derivatives) to allow us to study search over various aspects of the architecture, e.g., the macro architecture (Section 4).

3. We propose *BOHNAS* to efficiently search over large hierarchical search spaces (Section 5).

4. We thoroughly show how our search space design framework can be used to study the impact of architectural design principles on performance across granularities.

5. We show the superiority of BOHNAS over common baselines on 6/8 (others on par) datasets above state-of-the-art methods, using the same training protocol, including, e.g., a $4.99\%$ improvement on ImageNet-16-120 using the NAS-Bench-201 training protocol. Further, we show that we can effectively search on different types of search spaces (convolutional networks, transformers, or both) (Section 6).

6. We adhere to the NAS best practice checklist [13] and provide code at https://github.com/automl/hierarchical_nas_construction to foster *reproducible NAS research* (Appendix N).

## 2   Related work

We discuss related works in Neural Architecture Search (NAS) below and discuss works beyond NAS in Appendix B. Table 4 in Appendix B summarizes the differences between our proposed search space design based on CFGs and previous works.

Most previous works focused on global [17, 18], hyperparameter-based [1], chain-structured [19–22], or cell-based [23] search space designs. Hierarchical search spaces subsume the aforementioned spaces while being more expressive and effective in reducing search complexity. Prior works considered $n$-level hierarchical assembly [2, 9, 24], parameterization of a hierarchy of random

graph generators [8], or evolution of topologies and repetitive blocks [11]. In contrast to these prior works, we search over all (abstraction) levels of neural architectures and do not require any post-generation testing and/or adaptation of the architecture [8, 11]. Further, we can incorporate user-defined constraints and foster regularity in the search space design.

Other works used formal "systems": string rewriting systems [25, 26], cellular (or tree-structured) encoding schemes [27–30], hyperedge replacement graph grammars [31, 32], attribute grammars [33], CFGs [10, 34–40], And-Or-grammars [41], or a search space design language [42]. Different to these prior works, we search over the complete architecture with non-linear macro topologies, can incorporate user-defined constraints, and explicitly foster regularity.

For search, previous works, e.g., used reinforcement learning [17, 43], evolution [44], gradient descent [6, 45, 46], or Bayesian Optimization (BO) [18, 47, 48]. To enable the effective use of BO on graph-like inputs for NAS, previous works have proposed to use a GP with specialized kernels [18, 48, 49], encoding schemes [47, 50], or graph neural networks as surrogate model [51–53]. In contrast to these previous works, we explicitly leverage the hierarchical nature of architectures in the performance modeling of the surrogate model.

# 3 Unifying search space design framework

To efficiently construct expressive (hierarchical) NAS spaces, we review two common neural architecture representations and describe how they are connected (Section 3.1). In Sections 3.2 and 3.3 we propose to use CFGs and enhance them (or exploit their properties) to efficiently construct expressive (hierarchical) search spaces.

## 3.1 Neural architecture representations

**Representation as computational graphs**   A neural architecture can be represented as an edge-attributed (or equivalently node-attributed) directed acyclic (computational) graph $G(V, E, o_e, m_v)$, where $o_e$ is a primitive computation (e.g., convolution) or a computational graph itself (e.g., residual block) applied at edge $e \in E$, and $m_v$ is a merging operation for incident edges at node $v \in V$. Below is an example of an architecture with two residual blocks followed by a linear layer:

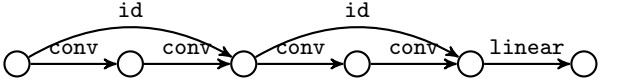

,

where the edge attributes `conv`, `id`, and `linear` correspond to convolutional blocks, skip connections, or linear layer, respectively. This architecture can be decomposed into sub-components/graphs across multiple *hierarchical levels* (see Figure 1 on the previous page): the operations convolutional blocks and skip connection construct the residual blocks, which construct with the linear layer the entire architecture.

**Representation as function compositions**   The example above can also naturally be represented as a composition of multivariate functions:

$$\texttt{Sequential}(\texttt{Residual}(\texttt{conv, id, conv}), \texttt{Residual}(\texttt{conv, id, conv}), \texttt{linear}) \quad , \quad (1)$$

where the functions `conv`, `id`, `linear`, as well as `Sequential` and `Residual` are defined with an arity of zero or three, respectively. We refer to functions with arity of zero as *primitive computations* and functions with non-zero arity as *topological operators* that define the (functional/graph) structure. Note the close resemblance to the way architectures are typically implemented in code. To construct the associated computational graph representation (or executable neural architecture program), we define a bijection $\Phi$ between function symbols and the corresponding computational graph (or function implemented in code, e.g., in Python), using a predefined vocabulary. Appendix D shows exemplary how a functional representation is mapped to its computational graph representation.

Finally, we introduce *intermediate variables* $x_i$ that can share architectural patterns across an architecture, e.g., residual blocks $x_{res} = \texttt{Residual}(\texttt{conv, id, conv})$. Thus, we can rewrite the architecture from Equation 1 as follows:

$$\texttt{Sequential}(x_{res}, x_{res}, \texttt{linear}) \quad . \quad (2)$$

## 3.2 Construction based on context-free grammars

To construct neural architectures as compositions of (multivariate) functions, we propose to use *Context-Free Grammars (CFGs)* [54] since they naturally and in a formally grounded way generate (expressive) languages that are (hierarchically) composed from an alphabet, i.e., the set of (multivariate) functions. CFGs also provide a simple and formally grounded mechanism to evolve architectures that ensure that evolved architectures stay within the defined search space (see Section 5 for details). With our enhancements of CFGs (Section 3.3), we provide a unifying search space design framework that is able to *represent all search spaces* we are aware of from the literature. Appendix F provides examples for NAS-Bench-101 [50], DARTS [6], Auto-DeepLab [2], hierarchical cell space [9], Mobile-net space [55], and hierarchical random graph generator space [8].

Formally, a CFG $\mathcal{G} = \langle N, \Sigma, P, S \rangle$ consists of a finite set of nonterminals $N$ and terminals $\Sigma$ (i.e., the alphabet of functions) with $N \cap \Sigma = \emptyset$, a finite set of production rules $P = \{A \to \beta | A \in N, \beta \in (N \cup \Sigma)^*\}$, where the asterisk denotes the Kleene star [56], and a start symbol $S \in N$. To generate a string (i.e., the function composition), starting from the start symbol $S$, we recursively replace nonterminals with the right-hand side of a production rule, until the resulting string does not contain any nonterminals. For example, consider the following CFG in extended Backus-Naur form [57] (refer to Appendix C for background on the extended Backus-Naur form):

$$S \ ::= \ \texttt{Sequential(S, S, S)} \ | \ \texttt{Residual(S, S, S)} \ | \ \texttt{conv} \ | \ \texttt{id} \ | \ \texttt{linear} \quad . \quad (3)$$

Figure 1 shows how we can derive the function composition of the neural architecture from Equation 1 from this CFG and makes the connection to its computational graph representation explicit. The set of all (potentially infinite) function compositions generated by a CFG $\mathcal{G}$ is the language $L(\mathcal{G})$, which naturally forms our search space. Thus, the NAS problem can be formulated as follows:

$$\underset{\omega \in L(\mathcal{G})}{\arg\min} \ \mathcal{L}(\Phi(\omega), \ \mathcal{D}) \quad , \quad (4)$$

where $\mathcal{L}$ is an error measure that we seek to minimize for some data $\mathcal{D}$, e.g., final validation error of a fixed training protocol.

## 3.3 Enhancements

Below, we enhance CFGs or utilize their properties to efficiently model changes in the spatial resolution, foster regularity, and incorporate constraints.

**Flexible spatial resolution flow**   Neural architectures commonly build a hierarchy of features that are gradually downsampled, e.g., by pooling operations. However, many NAS works do not search over the macro topology of architectures (e.g., Zoph et al. [23]), only consider linear macro topologies (e.g., Liu et al. [2]), or require post-generation testing for resolution mismatches with an adjustment scheme (e.g., Ru et al. [8], Miikkulainen et al. [11]).

To overcome these limitations, we propose a simple mechanism to search over the macro topology with flexible spatial resolution flow by *overloading nonterminals*: We assign to each nonterminal the number of downsampling operations required in its subsequent derivations. This effectively distributes the downsampling operations recursively across the architecture.

For example, the nonterminals $D1, D2$ of the production rule $D2 \to \texttt{Residual(D1, D2, D1)}$ indicate that 1 or 2 downsampling operations must be applied in their subsequent derivations, respectively. Thus, the input features of the residual topological operator $\texttt{Residual}$ will be downsampled twice in both of its paths and, consequently, the merging paths will have the same spatial resolution. Appendix D provides an example that also makes the connection to the computational graph explicit.

**Fostering regularity through substitution**   To foster regularity, i.e., reuse of architectural patterns, we implement intermediate variables (Section 3.1) by exploiting the property that context-free languages are closed under *substitution*. More specifically, we can substitute an intermediate variable $x_i \in \Sigma$ with a string $\omega$ of another language $L'$, e.g., constructing cell topologies. By substituting the same intermediate variable multiple times, we reuse the same architectural pattern ($\omega$) and, thereby, effectively foster regularity. Note that the language $L'$ may in turn have its own intermediate variables that map to languages constructing other architectural patterns, e.g., activation functions.

For example, consider the languages $L(\mathcal{G}_m)$ and $L(\mathcal{G}_c)$ constructing the macro or cell topology of a neural architecture, respectively. Further, we add a single intermediate variable $x_1$ to the terminals $\Sigma_{\mathcal{G}_m}$ that map to the string $\omega_1 \in L(\mathcal{G}_c)$, e.g., the searchable cell. Thus, after substituting all $x_1$ with $\omega_1$, we effectively share the same cell topology across the macro topology of the architecture.

**Constraints** When designing a search space, we often want to adhere to constraints. For example, we may only want to have two incident edges per node – as in the DARTS search space [6] – or ensure that for every neural architecture the input is associated with its output. Note that such constraints implicate *context-sensitivity* but CFGs by design are context-free. Thus, to still allow for constraints, we extend the sampling and evolution procedures of CFGs by using a (one-step) *lookahead* to ensure that the next step(s) in sampling procedure (or evolution) does not violate the constraint. We provide more details and a comprehensive example in Appendix E.

## 4 Example: Hierarchical NAS-Bench-201 and its derivatives

In this section, we propose a hierarchical extension to NAS-Bench-201 [58]: *hierarchical NAS-Bench-201* that subsumes (cell-based) NAS-Bench-201 [58] and additionally includes a search over the *macro topology* as well as the *parameterization of the convolutional blocks*, i.e., type of convolution, activation, and normalization. Below, is the definition of our proposed hierarchical NAS-Bench-201:

$$
\begin{aligned}
\text{D2} &::= \texttt{Sequential3(D1, D1, D0)} \mid \texttt{Sequential3(D0, D1, D1)} \mid \texttt{Sequential4(D1, D1, D0, D0)} \\
\text{D1} &::= \texttt{Sequential3(C, C, D)} \mid \texttt{Sequential4(C, C, C, D)} \mid \texttt{Residual3(C, C, D, D)} \\
\text{D0} &::= \texttt{Sequential3(C, C, CL)} \mid \texttt{Sequential4(C, C, C, CL)} \mid \texttt{Residual3(C, C, CL, CL)} \\
\text{D} &::= \texttt{Sequential2(CL, down)} \mid \texttt{Sequential3(CL, CL, down)} \mid \texttt{Residual2(CL, down, down)} \\
\text{C} &::= \texttt{Sequential2(CL, CL)} \mid \texttt{Sequential3(CL, CL, CL)} \mid \texttt{Residual2(CL, CL, CL)} \\
\text{CL} &::= \texttt{Cell(OP, OP, OP, OP, OP, OP)} \\
\text{OP} &::= \texttt{zero} \mid \texttt{id} \mid \texttt{CONVBLOCK} \mid \texttt{avg\_pool} \\
\text{CONVBLOCK} &::= \texttt{Sequential3(ACT, CONV, NORM)} \\
\text{ACT} &::= \texttt{relu} \mid \texttt{hardswish} \mid \texttt{mish} \\
\text{CONV} &::= \texttt{conv1x1} \mid \texttt{conv3x3} \mid \texttt{dconv3x3} \\
\text{NORM} &::= \texttt{batch} \mid \texttt{instance} \mid \texttt{layer} \quad .
\end{aligned}
$$
(5)

The blue productions of the nonterminals $\{\text{D2}, \text{D1}, \text{D0}, \text{D}, \text{C}\}$ construct the (non-linear) macro topology with flexible spatial resolution flow, possibly containing multiple branches. The red and yellow productions of the nonterminals $\{\text{CL}, \text{OP}\}$ construct the NAS-Bench-201 cell and $\{\text{CONVBLOCK}, \text{ACT}, \text{CONV}, \text{NORM}\}$ parameterize the convolutional block. Note that the red productions correspond to the original NAS-Bench-201 cell-based (sub)space [58]. Appendix A provides the vocabulary of topological operators and primitive computations and Appendix D provides a comprehensive example on the construction of the macro topology.

We omit the stem (i.e., 3x3 convolution followed by batch normalization) and classifier head (i.e., batch normalization followed by ReLU, global average pooling, and linear layer) for simplicity. We used element-wise summation as merge operation. For the number of channels, we adopted the common design to double the number of channels whenever we halve the spatial resolution. Alternatively, we could handle a varying number of channels by using, e.g., depthwise concatenation as merge operation; thereby also subsuming NATS-Bench [59]. Finally, we added a constraint to ensure that the input is associated with the output since `zero` could disassociate the input from the output.

**Search space capacity** The search space consists of ca. $\mathbf{10^{446}}$ architectures (Appendix G describes how to compute the search space size), which is hundreds of orders of magnitude larger than other popular (finite) search spaces from the literature, e.g., the NAS-Bench-201 or DARTS search spaces only entail ca. $1.5 \cdot 10^4$ and $10^{18}$ architectures, respectively.

**Derivatives** We can derive several variants of our hierarchical NAS-Bench-201 search space (`hierarchical`). This allows us to investigate search space as well as architectural design principles in Section 6. We briefly sketch them below and refer for their formal definitions to Appendix J.2:

- `fixed+shared` (cell-based): Fixed macro topology (only leftmost blue productions) with the single, shared NB-201 cell (red productions); equivalent to NAS-Bench-201 [58].

- `hierarchical+shared`: Hierarchical macro search (blue productions) with a single, shared cell (red & yellow productions).
- `hierarchical+non-linear`: Hierarchical macro search with more non-linear macro topologies (i.e., some `Sequential` are replaced by `Diamond` topological operators), allowing for more branching at the macro-level of architectures.
- `hierarchical+shared+non-linear`: Hierarchical macro search with more non-linear macro topologies (more branching at the macro-level) with a single, shared cell.

## 5  Bayesian Optimization for hierarchical NAS

Expressive (hierarchical) search spaces present challenges for NAS search strategies due to their huge size. In particular, gradient-based methods (without any yet unknown novel adoption) do not scale to expressive hierarchical search spaces since the supernet would yield an exponential number of weights (Appendix G provides an extensive discussion). Reinforcement learning approaches would also necessitate a different controller network. Further, we found that evolutionary and zero-cost methods did not perform particularly well on these search spaces (Section 6). Most Bayesian Optimization (BO) methods also did not work well [18, 52] or were not applicable (adjacency [50] or path encoding [47]), except for NASBOWL [48] with its Weisfeiler-Lehman (WL) graph kernel design in the surrogate model.

Thus, we propose the novel BO strategy Bayesian Optimization for Hierarchical Neural Architecture Search (BOHNAS), which (i) uses a novel *hierarchical kernel* that constructs a kernel upon different granularities of the architectures to improve performance modeling, and (ii) adopts ideas from grammar-guided genetic programming [60, 61] for acquisition function optimization of the discrete space of architectures. Below, we describe these components and provide more details in Appendix H.

**Hierarchical Weisfeiler-Lehman kernel (hWL)**   We adopt the WL graph kernel design [48, 62] for performance modeling.However, modeling solely based on the final computational graph of the architecture, similar to Ru et al. [48], ignores the useful hierarchical information inherent in our construction (Section 3). Moreover, the large size of the architectures also makes it difficult to use a single WL kernel to capture the more global topological patterns.

As a remedy, we propose the *hierarchical WL kernel (hWL)* that hierarchically constructs a kernel upon various granularities of the architectures. It efficiently captures the information in all hierarchical levels, which substantially improves search and surrogate regression performance (Section 6). To compute the kernel, we introduce fold operators $F_l$ that remove all substrings (i.e., inner functions) beyond the $l$-th hierarchical level, yielding partial function compositions (i.e., granularities of the architecture). E.g., the folds $F_1$, $F_2$ and $F_3$ for the function composition $\omega$ from Equation 1 are:

$$F_3(\omega) = \texttt{Sequential(Residual(conv, id, conv), Residual(conv, id, conv), linear)} ,$$
$$F_2(\omega) = \texttt{Sequential(Residual, Residual, linear)}  , \tag{6}$$
$$F_1(\omega) = \texttt{Sequential}  .$$

Note that $F_3(\omega) = \omega$ and observe the similarity to the derivations in Figure 1. We define hWL for two function compositions $\omega_i, \omega_j$ constructed over $L$ hierarchical levels, as follows:

$$k_{hWL}(\omega_i, \omega_j) = \sum_{l=2}^{L} \lambda_l \cdot k_{WL}(\Phi(F_l(\omega_i)), \Phi(F_l(\omega_j))) , \tag{7}$$

where $\Phi$ bijectively maps the function compositions to their computational graphs. The weights $\lambda_l$ govern the importance of the learned graph information at different hierarchical levels $l$ (granularities of the architecture) and can be optimized (along with other GP hyperparameters) by maximizing the marginal likelihood. We omit the fold $F_1(\omega)$ since it does not contain any edge features. Appendix H.2 provides more details on our proposed hWL.

**Grammar-guided acquisition function optimization**   Due to the discrete nature of the function compositions, we adopt ideas from grammar-based genetic programming [60, 61] for acquisition function optimization. For mutation, we randomly replace a substring (i.e., part of the function composition) with a new, randomly generated string with the same nonterminal as start symbol. For

Table 1: Test errors [%] of the best found architectures of BOHNAS (ours) compared with the respective best methods from the literature using the same training protocol. Table 6 (Appendix J.3), Table 2, and Tables 9 and 10 (Appendix L.3) provide the full tables for the NAS-Bench-201, activation function search, or DARTS training protocols, respectively.

| Dataset | C10 | C100 | IM16-120 | CTile | AddNIST | C10 | C10 | IM |
| Training prot. | NB201$^\dagger$ | NB201$^\dagger$ | NB201$^\dagger$ | NB201$^\dagger$ | NB201$^\dagger$ | Act. func. | DARTS | DARTS (transfer) |
|---|---|---|---|---|---|---|---|---|
| Previous best | 5.63 | 26.51 | 53.15 | 35.75 | 7.4 | **8.32** | **2.65**$^*$ | 24.63$^*$ |
| BOHNAS | **5.02** | **25.41** | **48.16** | **30.33** | **4.57** | 8.31 | 2.68 | **24.48** |
| Difference | +0.51 | +1.1 | +4.99 | +5.42 | +2.83 | +0.01 | -0.03 | +0.15 |

$^\dagger$ includes hierarchical search space variants. $^*$ reproduced results using the reported genotype.

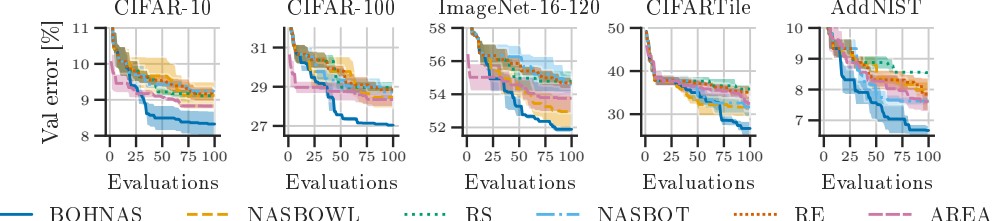

Figure 2: Mean $\pm$ standard error on the hierarchical NAS-Bench-201 search space. Appendix J.3 provides further results and extensive analyses.

crossover, we randomly swap two substrings produced by the same nonterminal as start symbol. We consider two crossover operators: a novel *self-crossover* operation swaps two substrings of the *same* string, and the common crossover operation swaps substrings of two different strings. Importantly, all evolutionary operations by design only result in valid function compositions (architectures) of the generated language. We provide visual examples for the evolutionary operations in Appendix H.

In our experiments, we used expected improvement as acquisition function and the Kriging Believer [63] to make use of parallel compute resources to reduce wallclock time. The Kriging Believer hallucinates function evaluations of pending evaluations at each iteration to avoid redundant evaluations.

## 6  Experiments

In the section, we show the versatility of our search space design framework, show how we can study the impact of architectural design choices on performance, and show the search efficiency of our search strategy BOHNAS by answering the following research questions:

**RQ1** How does our search strategy BOHNAS compare to other search strategies?

**RQ2** How important is the incorporation of hierarchical information in the kernel design?

**RQ3** How well do zero-cost proxies perform in large hierarchical search spaces?

**RQ4** Can we find well-performing transformer architectures for, e.g., language modeling?

**RQ5** Can we discover novel architectural patterns (e.g., activation functions) from scratch?

**RQ6** Can we find better-performing architectures in huge hierarchical search spaces with a limited number of evaluations, despite search being more complex than, e.g., in cell-based spaces?

**RQ7** How does the popular uniformity architecture design principle, i.e., repetition of similar architectural patterns, affect performance?

**RQ8** Can we improve performance further by allowing for more non-linear macro architectures while still employing the uniformity architecture design principle?

To answer these questions, we conducted extensive experiments on the hierarchical NAS-Bench-201 (Section 4), activation function [64], DARTS [6], and newly designed transformer-based search spaces (Appendix M.2). The activation function search shows the versatility of our search space design framework, where we search not for architectures, but for the composition of simple mathematical

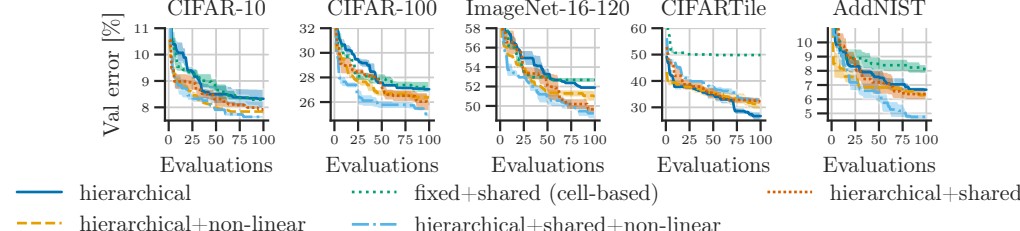

Figure 3: Mean $\pm 1$ standard error for several architecture search space designs (see Section 4 and Appendix J.2 for their CFGs) using BOHNAS for search.

functions. The search on the DARTS search space shows that BOHNAS is also backwards-compatible with cell-based spaces and achieves on-par or even superior performance to state-of-the-art gradient-based methods; even though these methods are over-optimized for the DARTS search space due to successive works targeting the same space. Search on the transformer space further shows that our search space design framework is also capable to cover the ubiquitous transformer architecture. To promote reproducibility, we discuss adherence to the NAS research checklist in Appendix N. We provide supplementary results (and ablations) in Appendices J.3, K.3, L.3 and M.3.

## 6.1 Evaluation details

We search for a total of 100 or 1000 evaluations with a random initial design of 10 or 50 architectures on three seeds {777, 888, 999} or one seed {777} on the hierarchical NAS-Bench-201 or activation function search space, respectively. We followed the training protocols and experimental setups of Dong and Yang [58] or Ramachandran et al. [64]. For search on the DARTS space, we search for one day with a random initial design of 10 on four seeds {666, 777, 888, 999} and followed the training protocol and experimental setup of Liu et al. [6], Chen et al. [45]. For searches on transformer-based spaces, we ran experiments each for one day with a random initial design of 10 on one seed {777}. In each evaluation, we fully trained the architectures or activation functions (using ResNet-20) and recorded the final validation error. We provide full training details and the experimental setups for each space in Appendices J.1, K.2, L.2 and M.1. We picked the best architecture, activation function, DARTS cells, or transformer architectures based on the final validation error (for NAS-Bench-201, activation function, and transformer-based search experiments) or on the average of the five last validation errors (for DARTS). All search experiments used 8 asynchronous workers, each with a single NVIDIA RTX 2080 Ti GPU.

We chose the search strategies Random Search (RS), Regularized Evolution (RE) [12], AREA [65], NASBOT [18], and NASBOWL [48] as baselines. Note that we could not use gradient-based approaches for our experiments on the hierarchical NAS-Bench-201 search space since they do not scale to large hierarchical search spaces without any novel adoption (see Appendix G for a discussion). Also note that we could not apply AREA to the activation function search since it uses binary activation codes of ReLU as zero-cost proxy. Appendix I provides the implementation details of the search strategies.

## 6.2 Results

In the following we answer all of the questions **RQ1**-**RQ8**. Figure 2 shows that BOHNAS finds superior architectures on the hierarchical NAS-Bench-201 search space compared to common baselines (answering **RQ1**); including NASBOWL, which does not use hierarchical information in its kernel design (partly answering **RQ2**), and zero-cost-based search strategy AREA (partly answering **RQ3**). We further investigated the hierarchical kernel design in Figure 12 in Appendix J.3 and found that it substantially improves regression performance of the surrogate model, especially on smaller amounts of training data (further answering **RQ2**). We also investigated other zero-cost proxies but they were mostly inferior to the simple baselines l2-norm [66] or flops [67] (Table 7 in Appendix J.3, further answering **RQ3**). We also found that BOHNAS is backwards compatible with cell-based search spaces. Table 1 shows that we found DARTS cells that are on-par on CIFAR-10 and (slightly) superior on ImageNet (with search on CIFAR-10) to state-of-the-art gradient-based methods; even though those are over-optimized due to successive works targeting this particular space.

Our searches on the transformer-based search spaces show that we can indeed find well-performing transformer architectures for, e.g., language modeling or sentiment analysis (**RQ4**). Specifically, we found a transformer for generative language modeling that achieved a best validation loss of *1.4386* with only $3.33\,\mathrm{M}$ parameters. For comparison, Karpathy [68] reported a best validation loss of *1.4697* with a transformer with $10.65\,\mathrm{M}$ parameters. Note that as we also searched for the embedding dimensionality, that acts globally on the architecture, we combined our hierarchical graph kernel (hWL) with a Hamming kernel. This demonstrates that our search space design framework and search strategy, BOHNAS, are applicable not only to NAS but also to joint NAS and Hyperparameter Optimization (HPO). The found classifier head for sentiment analysis achieved a test accuracy of $92.26\,\%$. We depict the found transformer and classifier head in Appendix M.3.

Table 2: Results of the activation function search on CIFAR-10 with ResNet-20.

| Search strategy | Test error [%] |
| --- | --- |
| ReLU | 8.93 |
| Swish | 8.61 |
| RS | 8.91 |
| RE | 8.47 |
| NASBOWL | **8.32** |
| BOHNAS | **8.31** |

The experiments on activation function search (see Table 2) show that our search space design framework as well as BOHNAS can be effectively used to search for architectural patterns from even more primitive, mathematical operations (answering **RQ1** & **RQ5**). Unsurprisingly, in this case, NASBOWL is on par with BOHNAS, since the computational graphs of the activation functions are small. This result motivates further steps in searching in expressive search spaces from even more atomic primitive computations.

To study the impact of the search space design and, thus, the architectural design choices on performance (**RQ6**-**RQ8**), we used various derivatives of our proposed hierarchical NAS-Bench-201 search space (see Section 4 and Appendix J.2 for the formal definitions of these search spaces). Figure 3 shows that we can indeed find better-performing architectures in hierarchical search spaces compared to simpler cell-based spaces, even though search is more complex due to the substantially larger search space (answering **RQ6**). Further, we find that the popular architectural uniformity design principle of reusing the same shared building block across the architecture (search spaces with keyword `shared` in Figure 3) improves performance (answering **RQ7**). This aligns well with the research in (manual and automated) architecture engineering over the last decade. However, we also found that adding more non-linear macro topologies (search spaces with keyword `non-linear` in Figure 3), thereby allowing for more branching at the macro-level of architectures, surprisingly further improves performance (answering **RQ8**). This is in contrast to the common linear macro architectural design without higher degree of branching at the macro-level; except for few notable exceptions [69, 70].

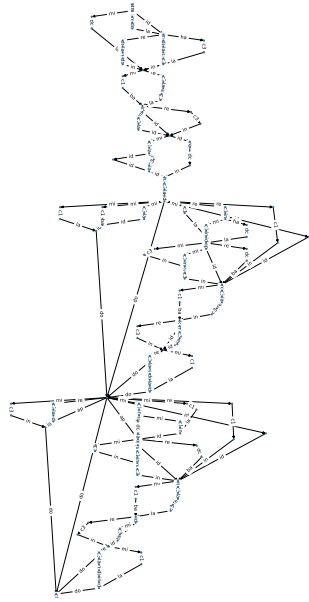

Figure 4: Best architecture found by BOHNAS for ImageNet-16-120 in the hierarchical NAS-Bench-201 search space. Best viewed with zoom. Figure 10 in Appendix J.3 provides the abbreviations of the operations and the architectures for the other datasets.

While most previous works in NAS focused on cell-based search spaces, Table 1 reveals that our found architectures are superior in 6/8 cases (others are on par) to the architectures found by previous works. On the NAS-Bench-201 training protocol, our best found architectures on CIFAR-10, CIFAR-100, and ImageNet-16-120 (depicted in Figure 4) reduce the test error by $0.51\,\%$, $1.1\,\%$, and $4.99\,\%$ to the best reported numbers from the literature, respectively. Notably, this is even better than the *optimal* cells in the cell-based NAS-Bench-201 search space (further answering **RQ6**). This clearly emphasizes the potential of NAS going beyond cell-based search spaces (**RQ6**) and prompts us to rethink macro architectural design choices (**RQ7**, **RQ8**).

# 7 Limitations

Our versatile search space design framework based on CFGs (Section 3) unifies *all* search spaces we are aware of from literature in a single framework; see Appendix F for several exemplar search spaces, and allows search over all (abstraction) levels of neural architectures. However, we cannot construct *any* architecture search space since we are limited to context-free languages (although our enhancements in Section 3.3 overcome some limitations), e.g., architecture search spaces of the type $\{a^n b^n c^n | n \in \mathbb{N}_{>0}\}$ cannot be generated by CFGs (this can be proven using Ogden's lemma [71]).

While more expressive search spaces facilitate the search over a wider spectrum of architectures, there is an *inherent trade-off* between the expressiveness and the search complexity. The mere existence of potentially better-performing architectures does not imply that we can actually find them with a limited search budget (Appendix J.3); although our experiments suggest that this may be more feasible than previously expected (Section 6). In addition, these potentially better-performing architectures may not work well with current training protocols and hyperparameters due to interaction effects between them and over-optimization for specific types of architectures. A joint optimization of neural architectures, training protocols, and hyperparameters could overcome this limitation, but further fuels the trade-off between expressiveness and search complexity.

Finally, our search strategy, BOHNAS, is sample-based and therefore *computationally intensive* (search costs for up to 40 GPU days for our longest search runs). While it would benefit from weight sharing, current weight sharing approaches are not directly applicable due to the exponential increase of architectures and the consequent memory requirements. We discuss this issue further in Appendix G.

# 8 Conclusion

We introduced a unifying search space design framework for hierarchical search spaces based on CFGs that allows us to search over *all (abstraction) levels* of neural architectures, foster *regularity*, and incorporate user-defined *constraints*. To efficiently search over the resulting huge search spaces, we proposed BOHNAS, an efficient BO strategy with a kernel leveraging the available hierarchical information. Our experiments show the *versatility* of our search space design framework and how it can be used to study the performance impact of architectural design choices beyond the micro-level. We also show that BOHNAS *can be superior* to existing NAS approaches. Our empirical findings motivate further steps into investigating the impact of other architectural design choices on performance as well as search on search spaces with even more atomic primitive computations. Next steps could include the improvement of search efficiency by means of multi-fidelity optimization or meta-learning, or simultaneously search for architectures and the search spaces themselves.

## Acknowledgments and Disclosure of Funding

This research was funded by the Deutsche Forschungsgemeinschaft (DFG, German Research Foundation) under grant number 417962828, and the Bundesministerium für Umwelt, Naturschutz, nukleare Sicherheit und Verbraucherschutz (BMUV, German Federal Ministry for the Environment, Nature Conservation, Nuclear Safety and Consumer Protection) based on a resolution of the German Bundestag (67KI2029A). Robert Bosch GmbH is acknowledged for financial support. We gratefully acknowledge support by the European Research Council (ERC) Consolidator Grant "Deep Learning 2.0" (grant no. 101045765). Funded by the European Union. This research was partially supported by TAILOR, a project funded by EU Horizon 2020 research and innovation programme under GA No 952215. Views and opinions expressed are however those of the author(s) only and do not necessarily reflect those of the European Union or the ERC. Neither the European Union nor the ERC can be held responsible for them.

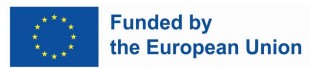

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

# A Vocabulary for primitive computations and topological operators

Table 3 and Figure 5 describe the primitive computations and topological operators used throughout our experiments, respectively. We defined the order of arguments of the topological operator as follows: we order arguments first by their sink nodes and then by the source node (following Dong and Yang [58]). Note that by adding more primitive computations and/or topological operators we could construct even more expressive search spaces.

Table 3: Primitive computations. "Name" corresponds to the string function symbols in our CFGs and "Function" is the associated implementation of the primitive computation in pseudocode. The subscripts $g$, $k$, $s$, and $p$ are abbreviations for groups, kernel size, strides, and padding, respectively. During assembly of neural architectures $\Phi(\omega)$, we replace string function symbols with the associated primitive computation.

| Name | Function |
|------|----------|
| avg_pool | $\mathtt{AvgPool}_{k=3,s=1,p=1}(\mathtt{x})$ |
| batch | $\mathtt{BN(x)}$ |
| conv1x1 | $\mathtt{Conv}_{k=1,s=1,p=0}(x)$ |
| conv3x3 | $\mathtt{Conv}_{k=3,s=1,p=1}(x)$ |
| dconv3x3 | $\mathtt{Conv}_{g=C,k=3,s=1,p=1}(x)$ |
| down | $\mathtt{conv3x3}_{s=1}(\mathtt{conv3x3}_{s=2}(\mathtt{x}))\ +\ \mathtt{Conv}_{k=1,s=1}(\mathtt{AvgPool}_{k=2,s=2}(\mathtt{x}))$ |
| hardswish | $\mathtt{Hardswish(x)}$ |
| id | $\mathtt{Identitiy(x)}$ |
| instance | $\mathtt{IN(x)}$ |
| layer | $\mathtt{LN(x)}$ |
| mish | $\mathtt{Mish(x)}$ |
| relu | $\mathtt{ReLU(x)}$ |
| zero | $\mathtt{Zeros(x)}$ |

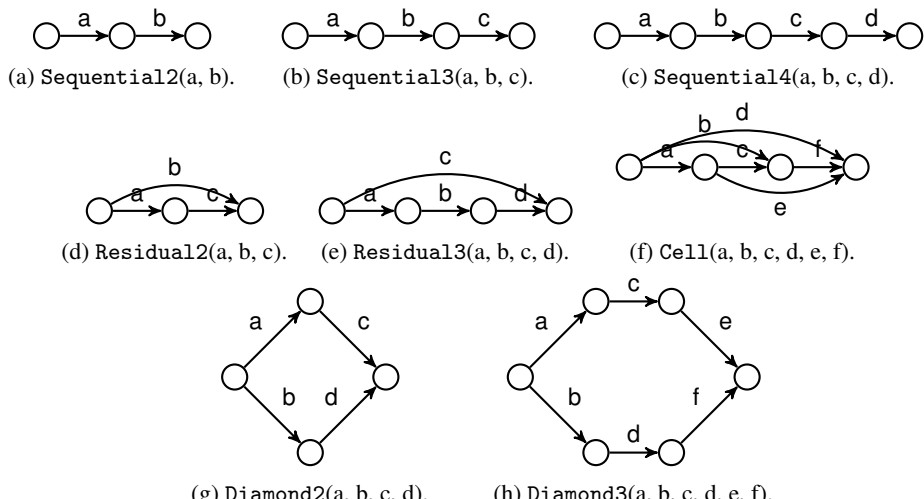

(a) Sequential2(a, b).  (b) Sequential3(a, b, c).  (c) Sequential4(a, b, c, d).

(d) Residual2(a, b, c).  (e) Residual3(a, b, c, d).  (f) Cell(a, b, c, d, e, f).

(g) Diamond2(a, b, c, d).  (h) Diamond3(a, b, c, d, e, f).

Figure 5: Topological operators. Each subfigure makes the connection between the topogolocial operator and associated computational graph explicit, i.e., the arguments of the topological operators (a, b, ...) are mapped to the respective edges of the corresponding the computational graph.

# B Extended related work

Complementary to Section 2, we provide a comparison of our proposed search space design to others from the literature in Table 4. Our search space design framework allows us to search over all aspects of the architecture and incorporate user-defined constraints as well as foster regularity.

Table 4: Comparison of search space designs.

| Design approach | Example | Search over/from | | | | |
| --- | --- | --- | --- | --- | --- | --- |
| | | Micro structure | Macro structure | Primitives | Constraints | Regularity |
| Global | Kandasamy et al. [18] | ✓ | ✓* | - | - | - |
| Hyperparameter-based | Tan and Le [1] | ✓ | ✓† | - | - | - |
| Chain-structured | Roberts et al. [19] | ✓ | - | ✓ | - | - |
| Cell-based | Zoph et al. [23] | ✓ | - | - | - | ✓° |
| n-level hierarchical assembly | Liu et al. [2] | ✓ | ✓† | - | - | ✓° |
| Graph generator hierarchy | Ru et al. [8] | ✓ | ✓* | - | - | - |
| CoDeepNEAT | Miikkulainen et al. [11] | ✓ | ✓* | - | - | ✓ |
| Formal systems | Negrinho et al. [42] | ✓ | ✓† | - | ✓ | - |
| Enhanced CFGs | Ours | ✓ | ✓ | ✓ | ✓ | ✓ |

†search only linear or confined macro topologies; *requires post-hoc validation scheme; °do not share architectural patterns hierarchically

While our work focuses exclusively on NAS, we will discuss below how it relates to the areas of optimizer search (as well as automated machine learning from scratch), and neural-symbolic programming.

Optimizer search is a closely related field to NAS, where we automatically search for an optimizer (i.e., an update function for the weights) instead of an architecture. Initial works used learnable parametric or non-parametric optimizers. While the former approaches [72–75] have poor scalability and generality, the latter works overcome those limitations. More specifically, Bello et al. [76] searched for an instantiation of hand-crafted patterns via reinforcement learning, while Wang et al. [77] proposed a tree-structured search space[1] and searched for optimizers via a modified Monte Carlo sampling approach. AutoML-Zero [78] took an even more general approach by searching over entire machine learning algorithms, including optimizers, from a generic search space built from basic mathematical operations with an evolutionary algorithm. Chen et al. [79] used RE to search for optimizers from a generic search space (inspired by AutoML-Zero) for the training of vision transformers [80].

Complementary to the above, there is also recent interest in automatically synthesizing programs from domain-specific languages. Gaunt et al. [81] proposed a hand-crafted program template and simultaneously optimized the parameters of the differentiable program with gradient descent. Valkov et al. [82] proposed a type-directed (top-down) enumeration and evolution approaches over differentiable functional programs. Shah et al. [83] hierarchically assembled differentiable programs and used neural networks for the approximation of missing expression in partial programs. Cui and Zhu [84] treated CFGs stochastically with trainable production rule sampling weights, which were optimized with a gradient descent [6]. However, naïvely (directly) applying gradient-based approaches (without any novel adaption) does not scale to hierarchical search spaces due to the exponential explosion of supernet weights, but still renders an interesting direction for future work.

Compared to these lines of work, we enhance CFGs to handle changes in spatial resolution, explicitly promote regularity in the search space design, and (compared to most of them) incorporate constraints. Note that the latter two could also be applied in those domains. We also proposed a BO search strategy to search efficiently with a tailored kernel design to handle the hierarchical nature of the architectural search space.

## C  Details on extended Backus-Naur form

The (extended) Backus-Naur form [57] is a meta-language to describe the syntax of CFGs. We use meta-rules of the form $S ::= \alpha$ where $S \in N$ is a nonterminal and $\alpha \in (N \cup \Sigma)^*$ is a string of nonterminals and/or terminals. We denote nonterminals in UPPER CASE, terminals corresponding to topological operators in `Initial upper case/teletype`, and terminals corresponding to primitive computations in `lower case/teletype`, e.g., $S ::=$ `Residual(S, id, S)`. To compactly express production rules with the same left-hand side nonterminal, we use the

---

[1]Note that the tree-structured search space can equivalently be described with a CFG (with a constraint on the number of maximum depth of the syntax trees).

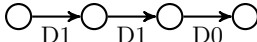

(a) Computational graph representation of current derivation in Equation 8.

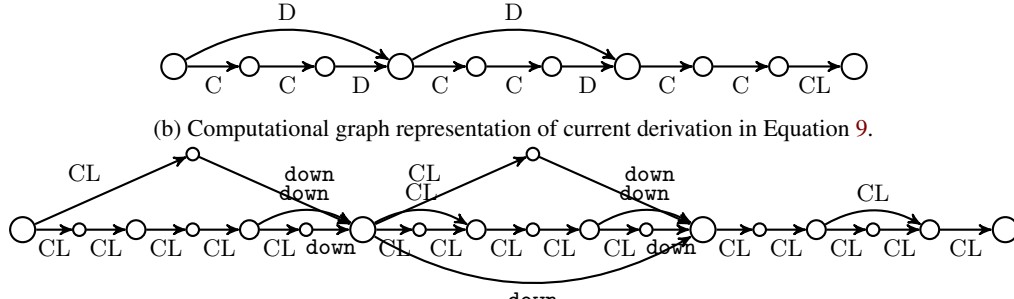

(b) Computational graph representation of current derivation in Equation 9.

(c) Computational graph representation of current derivation in Equation 10.

Figure 6: Comprehensive example for the construction of the macro topology in the hierarchical NAS-Bench-201 search space. We visualize the computational graphs of the derivation steps from Equations 8 to 10. Note that CLs (representing spatial resolution preserving cells) do not change the spatial resolution and the primitive computation `down` downsamples the features by a factor of two. Our proposed mechanism to overload the nonterminals (Section 3.3) effectively distributes the downsampling operations across the macro topology so that all incoming features at each node will have the same spatial resolution.

vertical bar | to indicate a choice of production rules with the same left-hand side, e.g., S ::= `Sequential(S, S, S) | Residual(S, id, S) | conv`.

## D  Comprehensive example for the construction of the macro topology

Consider the following derivation of the macro topology of an architecture in the hierarchical NAS-Bench-201 (see Section 4 for the search space definition):

$$\text{D2} \;\rightarrow\; \text{S3(D1, D1, D0)} \tag{8}$$

$$\rightarrow \text{S3(R3(C, C, D, D), R3(C, C, D, D), S3(C, C, CL))} \tag{9}$$

$$\begin{aligned}
\rightarrow\; &\text{S3(R3(S2(CL, CL), R2(CL, CL, CL), S2(CL, down), R2(CL, down, down)),} \\
&\text{R3(R2(CL, CL, CL), S2(CL, CL), R2(CL, down, down), R2(CL, down, down)),} \\
&\text{S3(S2(CL, CL), R2(CL, CL, CL), CL)} \qquad ,
\end{aligned} \tag{10}$$

where we abbreviate `Sequential` and `Residual` with S or R for sake of brevity. Figure 6 visualizes the corresponding computational graphs at each step and shows how our proposed mechanism of overloading nonterminals (Section 3.3) effectively distributes downsampling operations (`down`) across the macro topology of an architecture so that incoming features at each node will have the same spatial resolution.

## E  Comprehensive example for the implementation of constraints

In the design of a search space, one often wants to adhere to one or more constraints. For example, one wants to have exactly two incident edges per node [6] or wants to ensure that every architecture in the search space is valid, i.e., there needs to exist at least one path from the input to the output. CFGs (by design) cannot handle such constraints since it requires *context-sensitivity*. Thus, we extend CFGs with a $n$-step *lookahead* to guarantee compliance with the constraints. The $n$-step lookahead considers all $n$-step scenarios (e.g., all possible $n$-step derivations) and filters out all scenarios that would result in a violation with at least one constraint. In practice, we can leverage the recursive

nature of CFGs and find that one-step lookaheads are sufficient for the constraints considered in our work. In the following, we consider how the lookahead effectively ensures compliance for an exemplary constraint.

We consider the constraint "the input is associated with the output". To guarantee this constraint, it is sufficient to ensure that for each topological operator during sampling (or evolution) there is at least one path from the input to the output (due to the recursive nature of CFGs). To this end, we test whether the removal of an edge would result in a disassociation of the input from the output of the computational graph of the current topological operator (no path from input to output) with an one-step lookahead. More specifically, consider the following topological operator Cell(a, b, c, d,e, f):

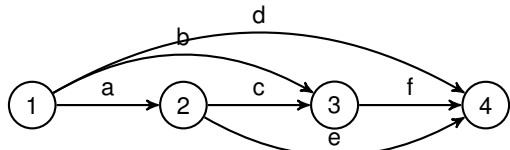

and the primitive computations zero and non-zero, where the former disassociates two nodes, i.e., removes their edge. The topological operator Cell has the following four paths from the input (1) to output (4): 1-2-3-4, 1-2-4, 1-3-4, and 1-4. Note that to *not* violate the constraint at least one of these paths needs to exist after sampling (or evolution). Below, we describe a potential sampling of primitive computations for the topological operator:

1. Substitute a with non-zero: valid since non-zero does not disassociate the input from the output.
2. Substitute b with zero: valid since the paths 1-2-3-4, 1-2-4, and 1-4 are still possible. Remove 1-3-4 from the paths.
3. Substitute c with zero: valid since the paths 1-2-4 and 1-4 are still possible. Remove 1-2-3-4 from the paths.
4. Substitute d with zero: valid since the path 1-2-4 is still possible. Remove 1-4 from the paths.
5. Substitute e with
   - zero: invalid since it would remove the last remaining possible path 1-2-4.
   - non-zero: valid since non-zero does not disassociate the input from the output.
6. Substitute f with zero: valid since the path 1-2-4 is still possible. Note that we also could substitute f with non-zero but this operation would be pruned away since the node 3 is not connected to the input.

Consequently, the sampling generates the following computational graph that adheres to the constraint:

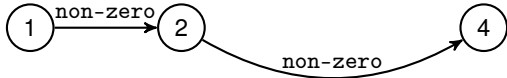

Note that this procedure only samples architectures which comply with the constraint. Further, we note that all architectures will be valid, i.e., there are no spatial resolution or channel mismatches due to our enhancements of CFGs (Section 3.3). We can similarly implement this procedure for evolution as well as for other constraints.

## F  Common search spaces from the literature

In Section 4, we demonstrated how to construct the hierarchical NAS-Bench-201 search space, that subsumes the cell-based NAS-Bench-201 search space [58], with our search space design framework. Below, we first provide a best practice guide to design search spaces and then show how to also define the following popular search spaces from the literature: NAS-Bench-101 search space [50], DARTS

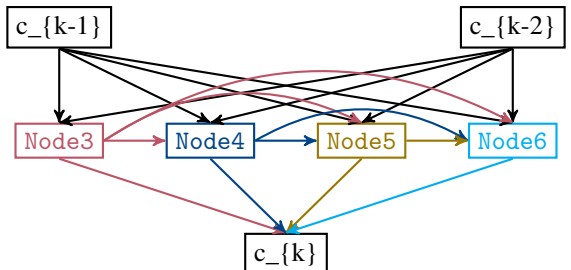

Figure 7: Visualization of the `Darts` topological operator.

search space [6], Auto-DeepLab search space [2], hierarchical cell search space [9], Mobile-net search space [55], and hierarchical random graph generator search space [8]. For more details on the search spaces, please refer to the respective works.

**Best practice to construct search space with context-free grammars**

We first note that the design largely depends on the target task. That is, an image-based task may require a different design than a text-based task. In general, when looking for high-performing architectures for a given task, we recommend reviewing the relevant literature and consulting domain experts if one is not particularly familiar with the application domain. In particular, we recommend distilling commonalities and differences among architectures that form the basis of the design for the CFG. Starting from this basis, we recommend expanding the search space by adding more and more production rules to the CFG to allow for the exploration of interesting novel architectures. This procedure ensures search can efficiently find well-performing architectures by exploiting prior knowledge in the search space design. We leave the integration of prior knowledge into BOHNAS for future work. If computational resources are virtually unlimited, one could also search for for architectures from scratch, similar to the AutoML-Zero [78].

**NAS-Bench-101 search space**

The NAS-Bench-101 search space [50] consists of a fixed macro architecture and a searchable cell, i.e., directed acyclic graph with maximally 7 nodes and 9 edges. We adopted the API of NAS-Bench-101 in our search space design and, thus, we can define the NAS-Bench-101 search space as follows

$$
\begin{aligned}
\text{NB101} \ &::= \ \texttt{ModelSpec}(\text{MATRIX, OPS}) \\
\text{MATRIX} \ &::= \ \texttt{Matrix}(\underbrace{\text{BINARY, ..., BINARY}}_{21x}) \\
\text{BINARY} \ &::= \ \texttt{0} \mid \texttt{1} \\
\text{OPS} \ &::= \ \texttt{List}(\text{OP, OP, OP, OP, OP}) \\
\text{OP} \ &::= \ \texttt{conv\_1x1} \mid \texttt{conv\_3x3} \mid \texttt{max\_pool}
\end{aligned}
\tag{11}
$$

and add a constraint to ensure that $|\text{BINARY} \rightarrow 1| \leq 9$.

**DARTS search space**

The DARTS search space [6] consists of a fixed macro architecture and a cell, i.e., a seven node directed acyclic graph (`Darts`; see Figure 7 for the topological operator). We omit the fixed macro architecture from our search space design for simplicity. Each cell receives the feature maps from the two preceding cells as input and outputs a single feature map. Each intermediate node (i.e., `Node3`, `Node4`, `Node5`, and `Node6`) is computed based on two of its predecessors. Thus, we can define the

DARTS search space as follows:

$$
\begin{aligned}
\text{DARTS} \ &::= \ \texttt{Darts}(\text{NODE3, NODE4, NODE5, NODE6}) \\
\text{NODE3} \ &::= \ \texttt{Node3}(\text{OP, OP}) \\
\text{NODE4} \ &::= \ \texttt{Node4}(\text{OP, OP, OP}) \\
\text{NODE5} \ &::= \ \texttt{Node5}(\text{OP, OP, OP, OP}) \\
\text{NODE6} \ &::= \ \texttt{Node6}(\text{OP, OP, OP, OP, OP}) \\
\text{OP} \ &::= \ \texttt{sep\_conv\_3x3} \mid \texttt{sep\_conv\_5x5} \mid \texttt{dil\_conv\_3x3} \mid \texttt{dil\_conv\_5x5} \\
&\quad \mid \texttt{max\_pool} \mid \texttt{avg\_pool} \mid \texttt{id} \mid \texttt{zero} \quad ,
\end{aligned}
\tag{12}
$$

where the topological operator `Node3` receives two inputs, applies the operations separately on them, and sums them up. Similarly, `Node4`, `Node5`, and `Node6` apply their operations separately to the given inputs and sum them up. The topological operator `Darts` feeds the corresponding feature maps into each of those topological operators and finally concatenates all intermediate feature maps. Since in the final discrete architecture each intermediate node only receives two non-zero inputs, we need to add a constraint to ensure that each intermediate node only has two incident edges. More specifically, we enforce that for each intermediate node only two OP primitive computations are non-`zero` and the others are `zero`. For example, one of the primitive computations of the topological operator `Node4` is enforced to be `zero`.

To avoid adding an explicit constraint, we can utilize that the topological operators can be an arbitrary (Python) function (with an associated computational graph which is only required for search strategies using the graph information in their performance modeling). Thus, we can equivalently express the DARTS search space as follows:

$$
\begin{aligned}
\text{DARTS} \ &::= \ \texttt{GENOTYPE}(\text{OP, IN1, OP, IN1, OP, IN2, OP, IN2, OP, IN3, OP, IN3, OP, IN4, OP, IN4}) \\
\text{OP} \ &::= \ \texttt{sep\_conv\_3x3} \mid \texttt{sep\_conv\_5x5} \mid \texttt{dil\_conv\_3x3} \mid \texttt{dil\_conv\_5x5} \\
&\quad \mid \texttt{max\_pool} \mid \texttt{avg\_pool} \mid \texttt{id} \\
\text{IN1} \ &::= \ \texttt{0} \mid \texttt{1} \\
\text{IN2} \ &::= \ \texttt{0} \mid \texttt{1} \mid \texttt{2} \\
\text{IN3} \ &::= \ \texttt{0} \mid \texttt{1} \mid \texttt{2} \mid \texttt{3} \\
\text{IN4} \ &::= \ \texttt{0} \mid \texttt{1} \mid \texttt{2} \mid \texttt{3} \mid \texttt{4} \quad ,
\end{aligned}
\tag{13}
$$

where `GENOTYPE` resembles the Python object (`namedtuple`), which can be plugged (almost) directly into standard DARTS implementations.

**Auto-DeepLab search space**

Auto-DeepLab [2] combines a cell-level with a network-level search space to search for segmentation networks, where the cell is shared across the searched macro architecture, i.e., a twelve step (linear) path across different spatial resolutional levels. The cell-level design is similar to the one of Liu et al. [6] and, thus, we can use a similar search space design as in Equation 12. For the network-level, we introduce a constraint that ensures that the path is of length twelve, i.e., we ensure exactly twelve derivations in our CFG at the network-level. Further, we overload the nonterminals so that they correspond to the respective spatial resolutional level, e.g., D4 indicates that the original input is downsampled by a factor of four (see Section 3.3 for details). For the sake of simplicity, we omit the first two layers and atrous spatial pyramid poolings as they are fixed, and hence define the network-level search space as follows:

$$
\begin{aligned}
\text{D4} \ &::= \ \texttt{Same}(\text{CELL, D4}) \mid \texttt{Down}(\text{CELL}, D8) \\
\text{D8} \ &::= \ \texttt{Up}(\text{CELL, D4}) \mid \texttt{Same}(\text{CELL, D8}) \mid \texttt{Down}(\text{CELL, D16}) \\
\text{D16} \ &::= \ \texttt{Up}(\text{CELL, D8}) \mid \texttt{Same}(\text{CELL, D16}) \mid \texttt{Down}(\text{CELL, D32}) \\
\text{D32} \ &::= \ \texttt{Up}(\text{CELL, D16}) \mid \texttt{Same}(\text{CELL, D32}) \quad ,
\end{aligned}
\tag{14}
$$

where the topological operators `Up`, `Same`, and `Down` upsample/halve, do not change/do not change, or downsample/double the spatial resolution/channels, respectively. The intermediate variable `CELL` maps to the shared cell.

**Hierarchical cell search space**

The hierarchical cell search space [9] consists of a fixed (linear) macro architecture and a hierarchically assembled cell with three levels which is shared across the macro architecture. Thus, we can omit

the fixed macro architecture from our search space design for simplicity. The first, second, and third hierarchical levels correspond to the primitive computations (i.e., `id`, `max_pool`, `avg_pool`, `sep_conv`, `depth_conv`, `conv`, `zero`), six densely connected four node directed acyclic graphs (`DAG4`), and a densely connected five node directed acyclic graph (`DAG5`), respectively. The `zero` operation could lead to directed acyclic graphs which have fewer nodes. Therefore, we introduce a constraint enforcing that there are always four (level 2) or five (level 3) nodes for every directed acyclic graph. More specifically, we only need to ensure each node has at least on incident and outgoing edge. Alternatively, we could write out all all variants explicitly. Further, since a densely connected five node directed acyclic graph graph can have (at most) ten edges, we need to introduce the intermediate variables (i.e., `M1`, ..., `M6`) to enforce that only six (possibly) different four node directed acyclic graphs are used. Consequently, we define a CFG for the third hierarchical level

$$\text{LEVEL3} ::= \text{DAG5}(\underbrace{\text{LEVEL2}, ..., \text{LEVEL2}}_{\times 10})$$
$$\text{LEVEL2} ::= \text{M1} \mid \text{M2} \mid \text{M3} \mid \text{M4} \mid \text{M5} \mid \text{M6} \mid \text{zero} \quad , \tag{15}$$

where we substitute the intermediate variables `M1`, ..., `M6` with the six lower-level motifs constructed by the first and second hierarchical level

$$\text{LEVEL2} ::= \text{DAG4}(\underbrace{\text{LEVEL1}, ..., \text{LEVEL1}}_{\times 6})$$
$$\text{LEVEL1} ::= \text{id} \mid \text{max\_pool} \mid \text{avg\_pool} \mid \text{sep\_conv} \mid \text{depth\_conv} \mid \text{conv} \mid \text{zero} \quad . \tag{16}$$

**Mobile-net search space**

Factorized hierarchical search spaces, e.g., the Mobile-net search space [55], factorize a (fixed) macro architecture – often based on an already well-performing reference architecture – into separate blocks (e.g., cells). For the sake of simplicity, we assume a three sequential blocks (`Block`) reference architecture (`Sequential`). In each of those blocks, we search for the convolution operations (CONV), kernel sizes (KSIZE), squeeze-and-excitation ratio (SERATIO) [85], skip connections (SKIP), number of output channels (FSIZE), and number of layers per block (#LAYERS), where the latter two are discretized using the reference architecture. Consequently, we can express this search space as follows:

$$
\begin{aligned}
\text{MACRO} &::= \text{Sequential}(\text{CONVBLOCK}, \text{CONVBLOCK}, \text{CONVBLOCK}) \\
\text{CONVBLOCK} &::= \text{Block}(\text{CONV}, \text{KSIZE}, \text{SERATIO}, \text{SKIP}, \text{FSIZE}, \text{\#LAYERS}) \\
\text{CONV} &::= \text{conv} \mid \text{dconv} \mid \text{mbconv} \\
\text{KSIZE} &::= \text{3} \mid \text{5} \\
\text{SERATIO} &::= \text{0} \mid \text{0.25} \\
\text{SKIP} &::= \text{pooling} \mid \text{id\_residual} \mid \text{no\_skip} \\
\text{FSIZE} &::= \text{0.75} \mid \text{1.0} \mid \text{1.25} \\
\text{\#LAYERS} &::= \text{-1} \mid \text{0} \mid \text{1} \quad ,
\end{aligned}
\tag{17}
$$

where `conv`, `donv` and `mbconv` correspond to convolution, depthwise convolution, and mobile inverted bottleneck convolution [86], respectively.

**Hierarchical random graph generator search space**

The hierarchical random graph generator search space [8] consists of three hierarchical levels of random graph generators (i.e., `Watts-Strogatz` [87] and `Erdõs-Rényi` [88]). We denote with `Watts-Strogatz_i` the random graph generated by the Watts-Strogatz model with i nodes. Thus,

we can represent the search space as follows:

$$
\begin{aligned}
\text{TOP} &::= \texttt{Watts-Strogatz\_3}(\text{K, Pt})(\text{MID, MID, MID}) \mid ... \\
&\mid \texttt{Watts-Strogatz\_10}(\text{K, Pt})\underbrace{(\text{MID, ..., MID})}_{\times 10} \\
\text{MID} &::= \texttt{Erdõs-Rényi\_1}(\text{Pm})(\text{BOT}) \mid ... \\
&\mid \texttt{Erdõs-Rényi\_10}(\text{Pm})\underbrace{(\text{BOT, ..., BOT})}_{\times 10} \\
\text{BOT} &::= \texttt{Watts-Strogatz\_3}(\text{K, Pb})(\text{NODE, NODE, NODE}) \mid ... \\
&\mid \texttt{Watts-Strogatz\_10}(\text{K, Pb})\underbrace{(\text{NODE ..., NODE})}_{\times 10} \\
\text{K} &::= \texttt{2} \mid \texttt{3} \mid \texttt{4} \mid \texttt{5} \quad ,
\end{aligned}
\tag{18}
$$

where each terminal Pt, Pm, and Pb map to a continuous number in $[0.1, \ 0.9]$[2] and the placeholder variable NODE maps to a primitive computation, e.g., separable convolution. Note that we omit other hyperparameters, such as stage ratio, channel ratio etc., for simplicity.

## G  Computation of search space and supernet size

In this section, we show how to efficiently compute the size of search spaces and the size of a (hypothetical) supernet.

**Search space size**  There are two cases to consider: (i) a CFG contains cycles (i.e., part of the derivation can be repeated infinitely many times), yielding an open-ended, infinite search space; and (ii) a CFG contains no cycles, yielding in a finite search space whose size we can compute.

Consider a production $\text{A} \rightarrow \texttt{Residual}(\text{B, B, B})$ where $\texttt{Residual}$ is a terminal, and A and B are nonterminals with $\text{B} \rightarrow \texttt{conv} \mid \texttt{id}$. Consequently, there are $2^3 = 8$ possible instances of the residual block. If we add another production rule for the nonterminal A, e.g., $\text{A} \rightarrow \texttt{Sequential}(\text{B, B, B})$, we would have $2^3 + 2^3 = 16$ possible instances. Further, adding a production rule $\text{C} \rightarrow \texttt{Sequential}(\text{A, A, A})$ would yield a search space size of $(2^3 + 2^3)^3 = 4096$.

More generally, we introduce the function $P_\text{A}$ that returns the set of productions for nonterminal $\text{A} \in N$, and the function $\mu : P \rightarrow N$ that returns all the nonterminals for a production $p \in P$. We can then recursively, starting from the start symbol $\text{A} \in N$, compute the size of the search space as follows:

$$
f(\text{A}) := \sum_{p \in P_\text{A}} \begin{cases} 1 & , \ \mu(p) = \emptyset, \\ \prod_{\text{A}' \in \mu(p)} f(\text{A}') & , \ \text{otherwise} \end{cases} .
\tag{19}
$$

When a CFG contains some constraint, we ensure to only account for valid architectures (i.e., compliant with the constraints) by ignoring productions which would lead to invalid architectures.

**Supernet size**  Stochastic supernets are a key component of recently popular fast search strategies, e.g., gradient-based methods. A supernet is composed of *all* possible architectures of a search space. We outline how one could naïvely adopt these gradient-based methods to hierarchical search spaces and discuss why these supernets (without any novel adaption) do not scale to large hierarchical search spaces.

DARTS [6] and its variants (e.g., Chen et al. [45], Dong and Yang [58], Xiao et al. [89]) relax the categorical (discrete) choice of primitive computations $\mathcal{O}$, e.g., $\bar{o}^{(i,j)}(x) = \sum_{o \in \mathcal{O}} m(x)o(x)$, where $m(\cdot)$ is a mixing function, e.g., DARTS uses the softmax over all possible primitive computations $m(x) = \frac{\exp(\alpha_o^{(i,j)})}{\sum_{o' \in \mathcal{O}} \exp(\alpha_{o'}^{(i,j)})}$, and $o(\cdot)$ is a primitive computation, e.g., convolution.

---

[2]While CFGs do not support continuous production choices, we extend the notion of substitution by substituting a string representation of a Python (float) variable for the intermediate variables Pt, Pm, and Pb.

For hierarchical search spaces, we could extend DARTS by also relaxing the categorical choices of the topological operators $t_l \in \mathcal{T}_l$ (e.g., Liu et al. [2]), i.e., $\bar{t}_l^{(i,j)}(x) = \sum_{t_l \in \mathcal{T}_l} m_l(x) t_l(x)$ with, e.g.,

$m_l(x) = \frac{\exp(\alpha_{t_l}^{(i,j)})}{\sum_{t_l' \in \mathcal{T}_l} \exp(\alpha_{t_l'}^{(i,j)})}$, where $l$ indicates the hierarchical level, $i, j$ denote the edge $e = (i, j)$ of the topological operator $t_l$, and $\mathcal{T}_1 = \mathcal{O}$. Note that this corresponds to a *stochastic* treatment of CFGs with learnable sampling probabilities $\alpha_{t_l}^{(i,j)}$ for their productions. Further, note that we (implicitly) assume independence of the current derivation step from the previous ones, even though it actually is conditionally dependent on the previous derivation steps.

To search with gradient descent, we need to create a supernet that contains all (exponentially many) discrete architectures. To this end, we define densely connected $n$-node directed acyclic graphs $G_l$ that entails all possible topological operators at each hierarchical level $l \in [1, ..., L]$, in which each topological operator $t_l \in \mathcal{T}_l$ can be identified by the indication vector $z_{t_l} \in \{0, 1\}^{n(n-1)/2}$, where $n(n-1)/2 = |E_{G_l}|$ is the number of edges. Consequently, we can recursively compute the number of possible architectures represented by the supernet as follows

$$g(l) := \begin{cases} C_{G_1} & , l = 1, \\ \prod_{e \in \{1, ..., |E_{G_l}|\}} g(l-1) & , otherwise \end{cases} , \tag{20}$$

where $G_1$ corresponds directed acyclic graph for the cell-level search space of size $C_{G_1}$, e.g., $10^9$ for the DARTS cell. We can similarly compute the number of searchable cells as follows

$$g'(l) := \begin{cases} |E_{G_2}| & , l = 2, \\ \prod_{e \in \{1, ..., |E_{G_l}|\}} g'(l-1) & , \text{otherwise} \end{cases} . \tag{21}$$

Consequently, the supernet is of size of $g'(L) \cdot S_{G_1}$, where $S_{G_1}$ is the size of a searchable cell. It is apparent that the supernet size scales linearly in the number of cells $g'(L)$, which, however, scales exponentially in the number of hierarchical levels. This makes the application of methods using supernets, e.g., gradient-based ones, (generally) challenging in hierarchical search spaces with many hierarchical levels.

## H   More details on the proposed search strategy

In this section, we provide more details and comprehensive examples for our search strategy Bayesian Optimization for Hierarchical Neural Architecture Search (BOHNAS) presented in Section 5.

### H.1   Bayesian Optimization

Bayesian Optimization (BO) is a powerful family of search techniques for finding the global optimum of a black-box objective problem. It is particularly suited when the objective is expensive to evaluate and, thus, sample efficiency is highly important [90].

To minimize a black-box objective problem with BO, we first need to build a probabilistic surrogate to model the objective based on the observed data so far. Based on the surrogate model, we design an acquisition function to evaluate the utility of potential candidate points by trading off exploitation (where the posterior mean of the surrogate model is low) and exploration (where the posterior variance of the surrogate model is high). The next candidate points to evaluate is then selected by maximizing the acquisition function [91]. The general procedure of BO is summarized in Algorithm 1.

We used the widely used acquisition function, expected improvement (EI) [92], in our BO strategy. EI evaluates the expected amount of improvement of a candidate point $\mathbf{x}$ over the minimal value $f'$ observed so far. Specifically, denote the improvement function as $I(\mathbf{x}) = \max(0, f' - f(\mathbf{x}))$, the EI acquisition function has the form

$$\alpha_{EI}(\mathbf{x}|\mathcal{D}_t) = \mathbb{E}[I(\mathbf{x})|\mathcal{D}_t] = \int_{-\infty}^{f'} (f' - f)\mathcal{N}\left(f; \mu(\mathbf{x}|\mathcal{D}_t), \sigma^2(\mathbf{x}|\mathcal{D}_t)\right) df$$
$$= (f' - f)\Phi\left(f'; \mu(\mathbf{x}|\mathcal{D}_t), \sigma^2(\mathbf{x}|\mathcal{D}_t)\right) + \sigma^2(\mathbf{x}|\mathcal{D}_t)\phi(f'; \mu(\mathbf{x}|\mathcal{D}_t), \sigma^2(\mathbf{x}|\mathcal{D}_t)) \quad ,$$

**Algorithm 1** Bayesian Optimization algorithm [90].

---

**Input:** Initial observed data $\mathcal{D}_t$, a black-box objective function $f$, total number of BO iterations $T$
**Output:** The best recommendation about the global optimizer $\mathbf{x}^*$
**for** $t = 1, \ldots, T$ **do**
    Select the next $\mathbf{x}_{t+1}$ by maximizing acquisition function $\alpha(\mathbf{x}|\mathcal{D}_t)$
    Evaluate the objective function at $f_{t+1} = f(\mathbf{x}_{t+1})$
    $\mathcal{D}_{t+1} \leftarrow \mathcal{D}_t \cup (\mathbf{x}_{t+1}, f_{t+1})$
    Update the surrogate model with $\mathcal{D}_{t+1}$
**end for**

---

**Algorithm 2** Kriging Believer algorithm to select one batch of points.

---

**Input:** Observation data $\mathcal{D}_t$, batch size $b$
**Output:** The batch points $\mathcal{B}_{t+1} = \{\mathbf{x}_{t+1}^{(1)}, \ldots, \mathbf{x}_{t+1}^{(b)}\}$
$\tilde{\mathcal{D}}_t = \mathcal{D}_t \cup \tilde{\mathcal{D}}_p$
**for** $j = 1, \ldots, b$ **do**
    Select the next $\mathbf{x}_{t+1}^{(j)}$ by maximizing acquisition function $\alpha(\mathbf{x}|\tilde{\mathcal{D}}_t)$
    Compute the predictive posterior mean $\mu(\mathbf{x}_{t+1}^{(j)}|\tilde{\mathcal{D}}_t)$
    $\tilde{\mathcal{D}}_t \leftarrow \tilde{\mathcal{D}}_t \cup (\mathbf{x}_{t+1}, \mu(\mathbf{x}_{t+1}^{(j)}|\tilde{\mathcal{D}}_t))$
**end for**

---

where $\mu(\mathbf{x}|\mathcal{D}_t)$ and $\sigma^2(\mathbf{x}|\mathcal{D}_t)$ are the mean and variance of the predictive posterior distribution at a candidate point $\mathbf{x}$, and $\phi(\cdot)$ and $\Phi(\cdot)$ denote the PDF and CDF of the standard normal distribution, respectively.

To make use of ample distributed computing resource, we adopted Kriging Believer [63] which uses the predictive posterior of the surrogate model to assign hallucinated function values $\{\tilde{f}_p\}_{p \in \{1, \ldots, P\}}$ to the $P$ candidate points with pending evaluations $\{\tilde{\mathbf{x}}_p\}_{p \in \{1, \ldots, P\}}$ and perform next BO recommendation in the batch by pseudo-augmenting the observation data with $\tilde{\mathcal{D}}_p = \{(\tilde{\mathbf{x}}_p, \tilde{f}_p)\}_{p \in \{1, \ldots, P\}}$, namely $\tilde{\mathcal{D}}_t = \mathcal{D}_t \cup \tilde{\mathcal{D}}_p$. The algorithm of Kriging Believer at one BO iteration to select a batch of recommended candidate points is summarized in Algorithm 2.

### H.2 Hierarchical Weisfeiler-Lehman kernel

Inspired by Ru et al. [48], we adopted the Weisfeiler-Lehman (WL) graph kernel [62] in the GP surrogate model to handle the graph nature of neural architectures. The basic idea of the WL kernel is to first compare node labels, and then iteratively aggregate labels of neighboring nodes, compress them into a new label and compare them. Algorithm 3 summarizes the WL kernel procedure.

Ru et al. [48] identified three reasons for using the WL kernel: (1) it is able to compare labeled and directed graphs of different sizes, (2) it is expressive, and (3) it is relatively efficient and scalable. Hierarchical search spaces can express a diverse spectrum of neural architectures with very heterogeneous topological structure. Therefore, reason (1) is a very important property of the WL kernel to account for the diversity of neural architectures and makes other kernels (e.g., adjacency [50] or path encoding [47]) not applicable. Moreover, if we allow for many hierarchical levels, we can construct very large neural architectures. Therefore, reasons (2) and (3) are essential for accurate and fast modeling.

However, modeling solely based on the (standard) WL graph kernel neglects the useful hierarchical information from our construction process. Moreover, the large size of neural architectures makes it still challenging to capture the more global topological patterns. We therefore propose to use hierarchical information through a hierarchy of WL graph kernels that take into account the different granularities of the architectures and combine them in a weighted sum. To obtain the different granularities, we use the fold operators $F_l$ that removes parts beyond the $l$-th hierarchical level. Thereby, we obtain the folds

$$F_3(\omega) = \omega = \texttt{Sequential(Residual(conv, id, conv), Residual(conv, id, conv), linear)},$$
$$F_2(\omega) = \texttt{Sequential(Residual, Residual, linear)} \quad, \quad F_1(\omega) = \texttt{Sequential} \quad, \qquad .$$

**Algorithm 3** Weisfeiler-Lehman subtree kernel computation [62].

**Input:** Graphs $G_1$, $G_2$, maximum iterations $H$
**Output:** Kernel function value between the graphs
Initialize the feature vectors $\phi(G_1) = \phi_0(G_1), \phi(G_2) = \phi_0(G_2)$ with the respective counts of original node labels (i.e., the $h = 0$ WL features)
**for** $h = 1, \ldots H$ **do**
    Assign a multiset $M_h(v) = \{l_{h-1}(u) | u \in \mathcal{N}(v)\}$ to each node $v \in G$, where $l_{h-1}$ is the node label function of the $h-1$-th WL iteration and $\mathcal{N}$ is the node neighbor function
    Sort elements in multiset $M_h(v)$ and concatenate them to string $s_h(v)$
    Compress each string $s_h(v)$ using the hash function $f$ s.t. $f(s_h(v)) = f(s_h(w)) \iff s_h(v) = s_h(u)$
    Add $l_{h-1}$ as prefix for $s_h(v)$
    Concatenate the WL features $\phi_h(G_1), \phi_h(G_2)$ with the counts of the *new* labels: $\phi(G_1) = [\phi(G_1), \phi_h(G_1)], \phi(G_2) = [\phi(G_2), \phi_h(G_2)]$
    Set $l_h(v) := f(s_h(v)) \ \forall v \in G$
**end for**
Compute inner product $k = \langle \phi_h(G_1), \phi_h(G_2) \rangle$ between WL features $\phi_h(G_1), \phi_h(G_2)$ in RKHS $\mathcal{H}$

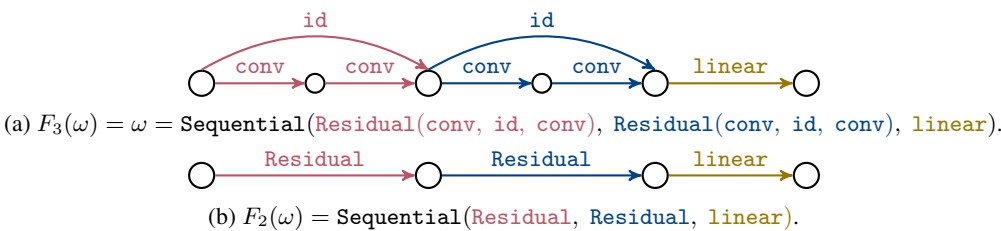

(a) $F_3(\omega) = \omega =$ `Sequential(Residual(conv, id, conv), Residual(conv, id, conv), linear)`.

(b) $F_2(\omega) =$ `Sequential(Residual, Residual, linear)`.

Figure 8: Labeled computational graphs $\Phi(F_2)$ and $\Phi(F_3)$ of the folds $F_2$ and $F_3$, respectively.

for the string $\omega$ from Equation 1. Figure 8 visualizes the labeled computational graphs $\Phi(F_2)$ and $\Phi(F_3)$ of the folds $F_2$ or $F_3$, respectively. Note that we ignore the first fold since it does not represent a labeled DAG. Also, note that we additionally add the previous nonterminals to provide additional information about the construction history. These graphs can be fed into (standard) WL graph kernels. Therefore, we can construct a hierarchy of WL graph kernels $k_{WL}$ as follows:

$$k_{hWL}(\omega_i, \omega_j) = \sum_{l=2}^{L} \lambda_l \cdot k_{WL}(\Phi(F_l(\omega_i)), \Phi(F_l(\omega_j))) \quad , \tag{22}$$

where $\omega_i$ and $\omega_j$ are two strings. Note that $\lambda_l$ governs the importance of the learned graph information across the hierarchical levels and can be optimized through the marginal likelihood. Note that while the hierarchical kernel structure seems superfluous for this example, for search spaces with many hierarchical levels it greatly improves the ability of the kernel to capture the more topological patterns (Section 6).

### H.3 Examples for the evolutionary operations

For the evolutionary operations, we adopted ideas from grammar-based genetic programming [60, 61]. In the following, we will show how these evolutionary operations manipulate the function composition of an architecture, e.g.,

`Sequential(Residual(conv, id, conv), Residual(conv, id, conv), linear)` , (23)

from the search space

S ::= `Sequential(S, S, S)` | `Residual(S, S, S)` | `conv` | `id` | `linear` . (24)

Figure 1 shows how we can derive the function composition in Equation 23 from the search space in Equation 24. For mutation operations, we first randomly pick a substring (i.e., part of the composition), e.g., `Residual(conv, id, conv)`. Then, we randomly sample a new function composition with the

same nonterminal symbol S as start symbol, e.g., `Sequential(conv, id, linear)`, and replace the previous substring, yielding

$$\texttt{Sequential(Sequential(conv, id, linear), Residual(conv, id, conv), linear)} \quad . \tag{25}$$

For (self-)crossover, we swap the two substrings, e.g., `Residual(conv, id, conv)` and `Residual(conv, id, conv)` with the same nonterminal S as start symbol, yielding

$$\texttt{Sequential(Residual(conv, id, conv), Residual(conv, id, conv), linear)} \quad . \tag{26}$$

Note that unlike the commonly used crossover operation, which uses two parents, self-crossover has only one parent. Future works could also consider a *self-copy* operation that copies a part of the function composition to another part of the (total) function composition of an architecture to explicitly regularizing diversity.

# I   Implementation details of the search strategies and dataset details

**BOHNAS, NASBOWL, and NASBOT**   We used the same hyperparameters for the three BO strategies. We ran the BO approaches asynchronously in parallel throughout our experiments with a batch size of $B = 1$, i.e., at each BO iteration a single architecture is proposed for evaluation. We used 10 or 50 random samples to initialize the BO strategies for the architecture (hierarchical NAS-Bench-201, DARTS, transformer) or activation function search spaces, respectively. For the acquisition function optimization, we used a pool size of $P = 200$, where the initial population consisted of the current ten best-performing architectures and the remainder were randomly sampled architectures to encourage exploration in the huge search spaces. During evolution, the mutation probability was set to $p_{mut} = 0.5$ and crossover probability was set to $p_{cross} = 0.5$. From the crossovers, half of them were self-crossovers of one parent and the other half were common crossovers between two parents. The tournament selection probability was set to $p_{tour} = 0.2$. We evolved the population at least for ten iterations and a maximum of 50 iterations using a early stopping criterion based on the fitness value improvements over the last five iterations.

**Regularized Evolution (RE) and AREA**   RE [9, 12] and AREA [65] iteratively mutate the best architectures out of a sample of the population. We reduced the population size from 50 to 30 to account for fewer evaluations in the architecture search experiments, and used a sample size of 10. For AREA, we randomly sampled 60 candidates and selected the initial population (30) based on its zero-cost proxy score (`nwot`). We also ran RE and AREA asynchronously for better comparability.

# J   Searching the hierarchical NAS-Bench-201 search space

In this section, we provide training and dataset details (Appendix J.1), provide search space definitions of variants of the hierarchical NAS-Bench-201 search space (Appendix J.2) and provide supplementary results to Section 6 and conduct extensive analyses (Appendix J.3).

## J.1   Training details

**Training protocol**   We evaluated all search strategies on CIFAR-10/100 [93], ImageNet-16-120 [94], CIFARTile, and AddNIST [95]. Note that CIFARTile and AddNIST are novel datasets and therefore have not yet been optimized by the research community. We provide further dataset details below. For training of architectures on CIFAR-10/100 and ImageNet-16-120, we followed the training protocol of Dong and Yang [58]. We trained architectures with SGD with learning rate of 0.1, Nesterov momentum of 0.9, weight decay of 0.0005 with cosine annealing [96], and batch size of 256 for 200 epochs. The initial channels were set to 16. For both CIFAR-10 and CIFAR-100, we used random flip with probability 0.5 followed by a random crop (32x32 with 4 pixel padding) and normalization. For ImageNet-16-120, we used a 16x16 random crop with 2 pixel padding instead.

For training of architectures on AddNIST and CIFARTile, we followed the training protocol from the CVPR-NAS 2021 competition [95]: We trained architectures with SGD with learning rate of 0.01, momentum of 0.9, and weight decay of 0.0003 with cosine annealing, and batch size of 64 for 64 epochs. We set the initial channels to 16 and did not apply any further data augmentation.

Table 5: Licenses for the datasets we used in our experiments.

| Dataset | License | URL |
|---------|---------|-----|
| CIFAR-10 [93] | MIT | https://www.cs.toronto.edu/~kriz/cifar.html |
| CIFAR-100 [93] | MIT | https://www.cs.toronto.edu/~kriz/cifar.html |
| ImageNet-16-120 [94] | MIT | https://patrykchrabaszcz.github.io/Imagenet32/ |
| CIFARTile [95] | GNU | https://github.com/RobGeada/cvpr-nas-datasets |
| AddNIST [95] | GNU | https://github.com/RobGeada/cvpr-nas-datasets |
| ImageNet [97] | Custom | https://www.image-net.org/index.php |
| Tiny Shakespeare [98] | MIT | https://github.com/karpathy/char-rnn |
| IMDb [99] | Custom | https://ai.stanford.edu/~amaas/data/sentiment/ |

**Dataset details** In Table 5, we provide the licenses for the datasets used in our experiments. For training of architectures on CIFAR-10, CIFAR-100 [93], and ImageNet-16-120 [94], we followed the dataset splits and training protocol of NAS-Bench-201 [58]. For CIFAR-10, we split the original training set into a new training set with 25k images and validation set with 25k images for search following [58]. The test set remained unchanged. For evaluation, we trained architectures on both the training and validation set. For CIFAR-100, the training set remained unchanged, but the test set was partitioned in a validation set and new test set each with 5K images. For ImageNet-16-120, all splits remained unchanged. For AddNIST and CIFARTile, we used the training, validation, and test splits as defined in the CVPR-NAS 2021 competition [95].

## J.2 Derivatives of the hierarchical NAS-Bench-201 search space

We can derive several derivatives of the hierarchical NAS-Bench-201 search space (Section 4). This derivative let us study the impact of different parts of the architecture (e.g., branching of the macro architecture or repetition of the same architectural blocks) in Section 6. Below, we provide their formal definitions.

**Derivative: `fixed+shared` (cell-based)**

This derivative uses a fixed macro topology with a single, shared cell. Formally, we first define the fixed macro search space:

$$
\begin{aligned}
\text{D2} &::= \text{Sequential3(D1, D1, D0)} \\
\text{D1} &::= \text{Sequential3(C, C, D)} \\
\text{D0} &::= \text{Sequential3(C, C, SHARED)} \\
\text{D} &::= \text{Sequential2(SHARED, down)} \\
\text{C} &::= \text{Sequential2(SHARED, SHARED)} \quad,
\end{aligned}
\tag{27}
$$

where the intermediate variable SHARED shares the cell across the fixed macro architecture and maps to a single instance of the cell-level space:

$$
\begin{aligned}
\text{CL} &::= \text{Cell(OP, OP, OP, OP, OP, OP)} \\
\text{OP} &::= \text{zero} \mid \text{id} \mid \text{CONVBLOCK} \mid \text{avg\_pool} \\
\text{CONVBLOCK} &::= \text{Sequential3(ACT, CONV, NORM)} \\
\text{ACT} &::= \text{relu} \\
\text{CONV} &::= \text{conv1x1} \mid \text{conv3x3} \\
\text{NORM} &::= \text{batch} \quad.
\end{aligned}
\tag{28}
$$

Note that the cell-level space is equivalent to the original NAS-Bench-201 search space [58], and since the macro level search space is fixed to the same macro architecture used in the original NAS-Bench-201 search space, this derivative is consequently equivalent to the original NAS-Bench-201 search space.

**Derivative:** `hierarchical+shared`

For this derivative, we searched over the macro search space but limited the cell-level part to a single instance (a shared cell). Formally, we define macro-level of this derivative as follows:

$$
\begin{aligned}
\text{D2} \;&::=\; \texttt{Sequential3}(\text{D1, D1, D0}) \;|\; \texttt{Sequential3}(\text{D0, D1, D1}) \;|\; \texttt{Sequential4}(\text{D1, D1, D0, D0}) \\
\text{D1} \;&::=\; \texttt{Sequential3}(\text{C, C, D}) \;|\; \texttt{Sequential4}(\text{C, C, C, D}) \;|\; \texttt{Residual3}(\text{C, C, D, D}) \\
\text{D0} \;&::=\; \texttt{Sequential3}(\text{C, C, SHARED}) \;|\; \texttt{Sequential4}(\text{C, C, C, SHARED}) \\
&\quad\; |\; \texttt{Residual3}(\text{C, C, SHARED, SHARED}) \\
\text{D} \;&::=\; \texttt{Sequential2}(\text{SHARED, down}) \;|\; \texttt{Sequential3}(\text{SHARED, SHARED, down}) \\
&\quad\; |\; \texttt{Residual2}(\text{SHARED, down, down}) \\
\text{C} \;&::=\; \texttt{Sequential2}(\text{SHARED, SHARED}) \;|\; \texttt{Sequential3}(\text{SHARED, SHARED, SHARED}) \\
&\quad\; |\; \texttt{Residual2}(\text{SHARED, SHARED, SHARED}) \quad ,
\end{aligned}
\tag{29}
$$

where the intermediate variable SHARED shares the cell across the macro architecture and maps to a single instance of the cell-level space:

$$
\begin{aligned}
\text{CL} \;&::=\; \texttt{Cell}(\text{OP, OP, OP, OP, OP, OP}) \\
\text{OP} \;&::=\; \texttt{zero} \;|\; \texttt{id} \;|\; \text{CONVBLOCK} \;|\; \texttt{avg\_pool} \\
\text{CONVBLOCK} \;&::=\; \texttt{Sequential3}(\text{ACT, CONV, NORM}) \\
\text{ACT} \;&::=\; \texttt{relu} \;|\; \texttt{hardswish} \;|\; \texttt{mish} \\
\text{CONV} \;&::=\; \texttt{conv1x1} \;|\; \texttt{conv3x3} \;|\; \texttt{dconv3x3} \\
\text{NORM} \;&::=\; \texttt{batch} \;|\; \texttt{instance} \;|\; \texttt{layer} \quad .
\end{aligned}
\tag{30}
$$

**Derivative:** `hierarchical+non-linear`

For this derivative, we replaced some of the linear macro topologies (`Sequential`) with non-linear macro topologies (`Diamond`); see Figure 5 for the topological operator `Diamond`. This results in architectures with more branching at the macro-level. Formally, we define this derivative as follows:

$$
\begin{aligned}
\text{D2} \;&::=\; \texttt{Sequential3}(\text{D1, D1, D0}) \;|\; \texttt{Sequential3}(\text{D0, D1, D1}) \;|\; \texttt{Sequential4}(\text{D1, D1, D0, D0}) \\
\text{D1} \;&::=\; \texttt{Sequential3}(\text{C, C, D}) \;|\; \texttt{Residual3}(\text{C, C, D, D}) \\
&\quad\; |\; \texttt{Diamond3}(\text{C, C, C, C, D, D}) \\
\text{D0} \;&::=\; \texttt{Sequential3}(\text{C, C, CL}) \;|\; \texttt{Residual3}(\text{C, C, CL, CL}) \\
&\quad\; |\; \texttt{Diamond3}(\text{C, C, C, C, CL, CL}) \\
\text{D} \;&::=\; \texttt{Sequential2}(\text{CL, down}) \;|\; \texttt{Residual2}(\text{CL, down, down}) \\
&\quad\; |\; \texttt{Diamond2}(\text{CL, CL, down, down}) \\
\text{C} \;&::=\; \texttt{Sequential2}(\text{CL, CL}) \;|\; \texttt{Residual2}(\text{CL, CL, CL}) \\
&\quad\; |\; \texttt{Diamond2}(\text{CL, CL, CL, CL}) \\
\text{CL} \;&::=\; \texttt{Cell}(\text{OP, OP, OP, OP, OP, OP}) \\
\text{OP} \;&::=\; \texttt{zero} \;|\; \texttt{id} \;|\; \text{CONVBLOCK} \;|\; \texttt{avg\_pool} \\
\text{CONVBLOCK} \;&::=\; \texttt{Sequential3}(\text{ACT, CONV, NORM}) \\
\text{ACT} \;&::=\; \texttt{relu} \;|\; \texttt{hardswish} \;|\; \texttt{mish} \\
\text{CONV} \;&::=\; \texttt{conv1x1} \;|\; \texttt{conv3x3} \;|\; \texttt{dconv3x3} \\
\text{NORM} \;&::=\; \texttt{batch} \;|\; \texttt{instance} \;|\; \texttt{layer} \quad .
\end{aligned}
\tag{31}
$$

**Derivative:** `hierarchical+shared+non-linear`

For this derivative, we replaced some of the linear macro topologies (`Sequential`) with non-linear macro topologies (`Diamond`). This results in architectures with more branching at the macro-level. Different to the derivative `hierarchical+non-linear` we used a single, shared cell. Formally, we

define this derivative as follows:

$$
\begin{aligned}
\text{D2} &::= \texttt{Sequential3(D1, D1, D0)} \mid \texttt{Sequential3(D0, D1, D1)} \mid \texttt{Sequential4(D1, D1, D0, D0)} \\
\text{D1} &::= \texttt{Sequential3(C, C, D)} \mid \texttt{Residual3(C, C, D, D)} \mid \texttt{Diamond3(C, C, C, C, D, D)} \\
\text{D0} &::= \texttt{Sequential3(C, C, SHARED)} \mid \texttt{Residual3(C, C, SHARED, SHARED)} \\
&\quad\mid \texttt{Diamond3(C, C, C, C, SHARED, SHARED)} \\
\text{D} &::= \texttt{Sequential2(SHARED, down)} \mid \texttt{Residual2(SHARED, down, down)} \\
&\quad\mid \texttt{Diamond2(SHARED, SHARED, down, down)} \\
\text{C} &::= \texttt{Sequential2(SHARED, SHARED)} \mid \texttt{Residual2(SHARED, SHARED, SHARED)} \\
&\quad\mid \texttt{Diamond2(SHARED, SHARED, SHARED, SHARED)} \qquad,
\end{aligned}
\tag{32}
$$

where the intermediate variable SHARED shares the cell across the macro architecture and maps to a single instance of the cell-level space:

$$
\begin{aligned}
\text{CL} &::= \texttt{Cell(OP, OP, OP, OP, OP, OP)} \\
\text{OP} &::= \texttt{zero} \mid \texttt{id} \mid \text{CONVBLOCK} \mid \texttt{avg\_pool} \\
\text{CONVBLOCK} &::= \texttt{Sequential3(ACT, CONV, NORM)} \\
\text{ACT} &::= \texttt{relu} \mid \texttt{hardswish} \mid \texttt{mish} \\
\text{CONV} &::= \texttt{conv1x1} \mid \texttt{conv3x3} \mid \texttt{dconv3x3} \\
\text{NORM} &::= \texttt{batch} \mid \texttt{instance} \mid \texttt{layer} \qquad.
\end{aligned}
\tag{33}
$$

### J.3 Supplementary results and analyses

**Search costs** Search time varied across datasets from ca. 0.5 days (CIFAR-10) to ca. 1.8 days (ImageNet-16-120) using eight asynchronous workers, each with an NVIDIA RTX 2080 Ti GPU, for ca. 4 to ca. 14.4 GPU days in total.

**Test errors** We report the test errors of our best found architectures in Table 6. We observe that our search strategy BOHNAS finds the strongest performing architectures across all dataset on the hierarchical as well as cell-based search space. Note that we found, e.g., on ImageNet-16-120, an architecture which is on par with the *optimal* architecture of the cell-based NAS-Bench-201 search space ($52.78\,\%$ vs. $52.69\,\%$). When using BOHNAS in the hierarchical search space with more non-linear macro topologies and a single, shared cell (`hierarchical+shared+non-linear`) we can even better architectures, which improved the test error by $0.51\,\%$, $1.1\,\%$, and $4.99\,\%$ on CIFAR-10, CIFAR-100, or ImageNet-16-120, respectively.

Figure 9 shows the test error vs. the number of parameters or FLOPs. Our best found architectures on hierarchical NAS-Bench-201 (`hierarchical`) fall well within the parameter and FLOPs ranges of the cell-based NAS-Bench-201 search space across all datasets, except for the parameters on CIFAR-10. Note that our best found architecture on ImageNet-16-120 on hierarchical NAS-Bench-201 is Pareto-optimal for test error vs. number of parameters and test error vs. number of FLOPs.

**Best architectures** Below we report the best found architecture per dataset on the hierarchical NAS-Bench-201 search space (Section 4) for each dataset. Figure 10 visualizes the novel and diverse design of the architectures (including stem and classifier head).

CIFAR-10 (mean test error $5.65\,\%$, #params $2.204\,\text{MB}$, FLOPs $127.673\,\text{M}$):

> Sequential4(Residual3(Residual2(Cell(id, zero, Sequential1(Sequential3(hardswish, conv1x1, layer)), Sequential1(Sequential3(hardswish, conv3x3, layer)), zero, Sequential1(Sequential3(mish, conv3x3, instance))), Cell(Sequential1(Sequential3(relu, dconv3x3, layer)), id, avg_pool, Sequential1(Sequential3(relu, dconv3x3, layer)), id, zero), Cell(zero, Sequential1(Sequential3(relu, conv1x1, layer)), id, Sequential1(Sequential3(hardswish, conv1x1, instance)), Sequential1(Sequential3(hardswish, conv3x3, layer)), Sequential1(Sequential3(hardswish, dconv3x3, layer)))), Residual2(Cell(id, zero, Sequential1(Sequential3(relu, conv1x1, layer)), Sequential1(Sequential3(mish, conv1x1, layer)), Sequential1(Sequential3(hardswish, conv3x3, layer)), zero), Cell(id, zero, id, Sequential1(Sequential3(relu, conv3x3, batch)), id, id), Cell(Sequential1(Sequential3(hardswish, conv3x3, layer)), Sequential1(Sequential3(hardswish, conv1x1, layer)), Sequential1(Sequential3(relu, conv1x1, layer)), Sequential1(Sequential3(relu, conv3x3, layer)),

Table 6: Test errors (and ±1 standard error) of popular baseline architectures (e.g., ResNet [100] and EfficientNet [1] variants), and our best found architectures on the cell-based and hierarchical NAS-Bench-201 search space. Note that we picked the ResNet and EfficientNet variant based on the test error, consequently giving an overestimate of their test performance. We also compared to gradient-based approaches which cannot be applied to hierarchical search spaces without any novel adoption.

† reported numbers.

* optimal numbers as reported in Dong and Yang [58].

(best) test error (and ±1 standard error) across three seeds {777, 888, 999} of the best architecture of the three search runs with lowest validation error.

° test errors on the hierarchical search space variant with more non-linear macro topologies and a single, shared cell (hierarchical+shared+non-linear, refer to Appendix J.2 for its definition).

| Method | CIFAR-10 cell-based | CIFAR-10 hierarchical | CIFAR-100 cell-based | CIFAR-100 hierarchical | ImageNet-16-120 cell-based | ImageNet-16-120 hierarchical | CIFARTile cell-based | CIFARTile hierarchical | AddNIST cell-based | AddNIST hierarchical |
|---|---|---|---|---|---|---|---|---|---|---|
| Best ResNet [100] | 6.49 ± 0.24 (32) | - | 27.1 ± 0.67 (110) | - | 53.67 ± 0.18 (56) | - | 57.8 ± 0.57 (18) | - | 7.78 ± 0.05 (34) | - |
| Best EfficientNet [1] | 11.73 ± 0.1 (B0) | - | 35.17 ± 0.42 (B6) | - | 77.73 ± 0.29 (B0) | - | 61.01 ± 0.62 (B0) | - | 13.24 ± 0.58 (B1) | - |
| DARTS [6] | 45.7 ± 0.00† | - | 61.03 ± 0.00† | - | 81.59 ± 0.00† | - | 76.1 ± 0.07 | - | 73.21 ± 0.13 | - |
| DrNAS [45] | 5.64 ± 0.00† | - | 26.49 ± 0.00† | - | 53.66 ± 0.00† | - | 54.77 ± 0.68 | - | 10.18 ± 0.74 | - |
| DrNAS (progressive) [45] | - | - | - | - | - | - | 55.15 ± 2.3 | - | 9.92 ± 0.63 | - |
| NAS-Bench-201 oracle* | 5.63 | - | 26.49 | - | 52.69 | - | - | - | - | - |
| RS | 6.39 ± 0.18 | 6.77 ± 0.1 | 28.75 ± 0.18 | 29.49 ± 0.57 | 54.83 ± 0.78 | 54.7 ± 0.82 | **52.72 ± 0.45** | 40.93 ± 0.81 | 7.82 ± 0.36 | 8.05 ± 0.29 |
| RE [9, 12] | **5.76 ± 0.17** | 6.88 ± 0.24 | 27.68 ± 0.55 | 30.0 ± 0.32 | 53.92 ± 0.6 | 55.39 ± 0.54 | 52.79 ± 0.59 | 40.99 ± 2.89 | **7.69 ± 0.35** | 7.56 ± 0.69 |
| AREA [65] | - | 6.37 ± 0.03 | - | 29.35 ± 0.79 | - | 54.24 ± 0.66 | - | 38.11 ± 2.39 | - | 7.4 ± 0.23 |
| NASWOT (N=10) [65] | 6.55 ± 0.1 | 8.18 ± 0.46 | 29.35 ± 0.53 | 31.73 ± 0.96 | 56.8 ± 1.35 | 58.66 ± 0.29 | 41.83 ± 2.29 | 49.46 ± 2.95 | 10.11 ± 0.69 | 11.81 ± 1.55 |
| NASWOT (N=100) [65] | 6.59 ± 0.17 | 8.56 ± 0.87 | 28.91 ± 0.25 | 31.65 ± 1.95 | 55.99 ± 1.3 | 58.47 ± 2.74 | 41.63 ± 1.02 | 43.31 ± 2.0 | 10.75 ± 0.23 | 14.47 ± 1.44 |
| NASWOT (N=1000) [65] | 6.68 ± 0.12 | 8.26 ± 0.38 | 29.37 ± 0.17 | 31.66 ± 0.72 | 58.93 ± 2.92 | 58.33 ± 0.91 | 39.61 ± 1.12 | 45.66 ± 1.29 | 10.68 ± 0.27 | 13.57 ± 1.89 |
| NASWOT (N=10000) [65] | 6.98 ± 0.43 | 8.4 ± 0.52 | 29.95 ± 0.42 | 32.09 ± 1.61 | 54.2 ± 0.49 | 57.58 ± 1.53 | 39.9 ± 1.2 | 42.45 ± 0.67 | 10.72 ± 0.53 | 14.82 ± 0.66 |
| NASBOT [18] | - | 6.68 ± 0.15 | - | 29.25 ± 0.55 | - | 54.61 ± 0.28 | - | 36.47 ± 2.25 | - | 7.45 ± 0.23 |
| NASBOWL [48] | **5.68 ± 0.11** | 6.98 ± 0.5 | **27.66 ± 0.18** | 28.7 ± 0.64 | **53.67 ± 0.39** | 53.47 ± 0.86 | 52.81 ± 0.27 | 35.75 ± 1.58 | 7.86 ± 0.41 | 8.2 ± 0.37 |
| BOHNAS | **5.68 ± 0.11** | **6.0 ± 0.16** | **27.66 ± 0.18** | **27.57 ± 0.46** | **53.67 ± 0.39** | **53.43 ± 0.61** | 52.81 ± 0.27 | **32.28 ± 2.39** | 7.86 ± 0.41 | **6.09 ± 0.34** |
| BOHNAS (best) | 5.64 ± 0.14 | 5.65 ± 0.09 | 27.03 ± 0.23 | 27.63 ± 0.2 | 53.54 ± 0.43 | 52.78 ± 0.23 | 53.18 ± 0.91 | 30.33 ± 0.77 | 8.04 ± 0.45 | 6.33 ± 0.59 |
| BOHNAS° | 5.37 ± 0.12 | - | 25.28 ± 0.19 | - | 49.47 ± 0.73 | - | 37.44 ± 1.62 | - | 4.27 ± 0.23 | - |
| BOHNAS° (best) | 5.02 ± 0.1 | - | 25.41 ± 0.28 | - | 48.16 ± 0.17 | - | 39.2 ± 2.24 | - | 4.57 ± 0.38 | - |

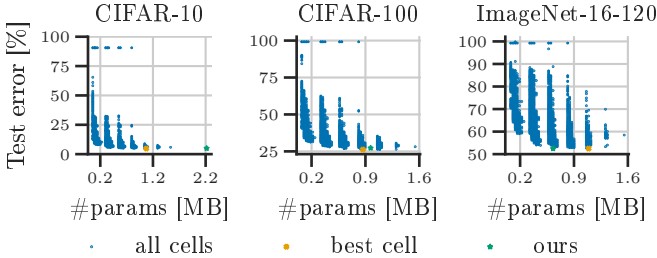

(a) Test error vs. number of parameters (#params (MB)).

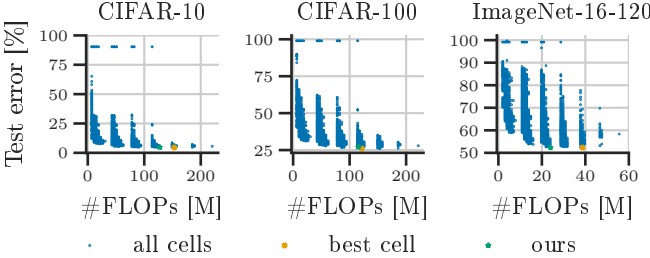

(b) Test error vs. number of FLOPs (#FLOPs (M)).

Figure 9: Test error vs. number of parameters (a) and FLOPs (b) for each architecture candidate in the cell-based search space (blue dots), the best cell (orange cross), and our best found architecture in the hierarchical NAS-Bench-201 search space (green star).

zero, id)), Residual2(Cell(Sequential1(Sequential3(hardswish, conv1x1, instance)), Sequential1(Sequential3(hardswish, dconv3x3, batch)), Sequential1(Sequential3(mish, dconv3x3, instance)), Sequential1(Sequential3(relu, conv1x1, batch)), id, id), down, down), Residual2(Cell(Sequential1(Sequential3(hardswish, conv1x1, layer)), Sequential1(Sequential3(hardswish, dconv3x3, batch)), Sequential1(Sequential3(relu, conv1x1, batch)), Sequential1(Sequential3(hardswish, conv3x3, layer)), id, avg_pool), down, down)), Residual3(Residual2(Cell(id, zero, Sequential1(Sequential3(hardswish, conv1x1, layer)), Sequential1(Sequential3(hardswish, conv3x3, layer)), id, Sequential1(Sequential3(mish, conv3x3, instance))), Cell(Sequential1(Sequential3(relu, dconv3x3, layer)), id, avg_pool, Sequential1(Sequential3(relu, dconv3x3, layer)), id, zero), Cell(zero, Sequential1(Sequential3(relu, conv1x1, layer)), id, Sequential1(Sequential3(hardswish, conv1x1, instance)), Sequential1(Sequential3(hardswish, conv1x1, layer)), Sequential1(Sequential3(hardswish, dconv3x3, layer)))), Residual2(Cell(id, zero, Sequential1(Sequential3(mish, conv1x1, layer)), Sequential1(Sequential3(mish, conv3x3, layer)), Sequential1(Sequential3(hardswish, dconv3x3, batch)), zero), Cell(id, zero, id, Sequential1(Sequential3(relu, conv3x3, batch)), id, id), Cell(Sequential1(Sequential3(hardswish, conv3x3, layer)), Sequential1(Sequential3(hardswish, conv1x1, layer)), Sequential1(Sequential3(mish, conv1x1, batch)), Sequential1(Sequential3(relu, conv3x3, instance)), zero, id)), Residual2(Cell(Sequential1(Sequential3(relu, conv1x1, batch)), Sequential1(Sequential3(hardswish, dconv3x3, batch)), id, Sequential1(Sequential3(relu, conv1x1, batch)), id, id), down, down), Residual2(Cell(Sequential1(Sequential3(hardswish, conv1x1, layer)), Sequential1(Sequential3(hardswish, dconv3x3, batch)), Sequential1(Sequential3(relu, conv1x1, batch)), Sequential1(Sequential3(hardswish, conv3x3, layer)), id, avg_pool), down, down)), Sequential3(Residual2(Cell(Sequential1(Sequential3(hardswish, conv3x3, batch)), Sequential1(Sequential3(relu, conv1x1, instance)), Sequential1(Sequential3(relu, conv1x1, layer)), Sequential1(Sequential3(relu, conv1x1, layer)), Sequential1(Sequential3(relu, conv1x1, layer)), id), Cell(Sequential1(Sequential3(relu, conv1x1, batch)), id, id, Sequential1(Sequential3(relu, conv1x1, layer)), id, id), Cell(id, Sequential1(Sequential3(relu, conv1x1, instance)), Sequential1(Sequential3(relu, conv1x1, instance)), Sequential1(Sequential3(relu, conv1x1, layer)), zero, Sequential1(Sequential3(hardswish, conv3x3, batch)))), Residual2(Cell(Sequential1(Sequential3(hardswish, conv3x3, layer)), Sequential1(Sequential3(relu, conv3x3, instance)), Sequential1(Sequential3(mish, conv1x1,

layer)), Sequential1(Sequential3(relu, conv1x1, layer)), Sequential1(Sequential3(relu, conv3x3, layer)), id), Cell(Sequential1(Sequential3(relu, conv1x1, batch)), id, id, Sequential1(Sequential3(relu, conv3x3, batch)), id, id), Cell(id, Sequential1(Sequential3(relu, conv1x1, instance)), Sequential1(Sequential3(relu, conv1x1, instance)), Sequential1(Sequential3(relu, dconv3x3, layer)), zero, Sequential1(Sequential3(hardswish, dconv3x3, layer)))), Cell(Sequential1(Sequential3(relu, dconv3x3, instance)), zero, zero, id, zero, id)), Sequential4(Residual2(Cell(Sequential1(Sequential3(hardswish, conv3x3, layer)), Sequential1(Sequential3(relu, conv3x3, layer)), Sequential1(Sequential3(relu, conv1x1, layer)), Sequential1(Sequential3(relu, conv1x1, layer)), Sequential1(Sequential3(relu, conv3x3, layer)), id), Cell(Sequential1(Sequential3(relu, conv1x1, batch)), id, id, Sequential1(Sequential3(relu, conv3x3, batch)), id, id), Cell(id, Sequential1(Sequential3(relu, conv1x1, instance)), Sequential1(Sequential3(relu, conv1x1, instance)), Sequential1(Sequential3(relu, conv1x1, layer)), zero, Sequential1(Sequential3(hardswish, conv3x3, batch)))), Residual2(Cell(Sequential1(Sequential3(hardswish, conv3x3, layer)), Sequential1(Sequential3(relu, conv3x3, layer)), Sequential1(Sequential3(relu, conv1x1, layer)), Sequential1(Sequential3(relu, conv1x1, layer)), Sequential1(Sequential3(relu, conv3x3, layer)), id), Cell(Sequential1(Sequential3(relu, conv1x1, batch)), id, id, Sequential1(Sequential3(relu, conv3x3, batch)), id, id), Cell(id, Sequential1(Sequential3(relu, conv1x1, instance)), Sequential1(Sequential3(relu, conv1x1, instance)), Sequential1(Sequential3(relu, conv1x1, layer)), zero, Sequential1(Sequential3(hardswish, conv3x3, layer)))), Residual2(Cell(Sequential1(Sequential3(hardswish, conv3x3, layer)), Sequential1(Sequential3(relu, conv3x3, layer)), Sequential1(Sequential3(relu, conv1x1, layer)), Sequential1(Sequential3(relu, conv1x1, layer)), Sequential1(Sequential3(relu, conv3x3, layer)), id), Cell(Sequential1(Sequential3(relu, conv1x1, batch)), id, id, Sequential1(Sequential3(relu, conv3x3, batch)), id, id), Cell(id, Sequential1(Sequential3(relu, conv1x1, instance)), Sequential1(Sequential3(relu, conv1x1, instance)), Sequential1(Sequential3(relu, conv1x1, layer)), zero, Sequential1(Sequential3(hardswish, conv3x3, layer)))), Cell(id, Sequential1(Sequential3(hardswish, conv1x1, layer)), Sequential1(Sequential3(mish, conv1x1, batch)), id, zero, id)))      .

CIFAR-100 (mean test error $27.63\,\%$, #params $0.962\,\mathrm{MB}$, FLOPs $115.243\,\mathrm{M}$):

Sequential3(Residual3(Sequential3(Cell(Sequential3(mish, conv3x3, layer), avg_pool, Sequential3(hardswish, conv1x1, instance), zero, Sequential3(mish, conv3x3, batch), zero), Cell(Sequential3(hardswish, dconv3x3, batch), zero, Sequential3(hardswish, dconv3x3, batch), Sequential3(relu, dconv3x3, batch), id, id), Cell(Sequential3(mish, conv3x3, batch), zero, id, zero, Sequential3(hardswish, dconv3x3, batch), id)), Sequential2(Cell(id, zero, Sequential3(mish, conv3x3, batch), zero, zero, Sequential3(mish, conv1x1, batch)), Cell(zero, zero, zero, id, zero, avg_pool)), Cell(Sequential3(relu, conv3x3, batch), zero, Sequential3(hardswish, conv3x3, instance), id, id, avg_pool), Cell(id, id, zero, zero, id, id)), Residual3(Sequential3(Cell(Sequential3(mish, conv3x3, layer), id, Sequential3(hardswish, dconv3x3, layer), Sequential3(hardswish, dconv3x3, batch), Sequential3(mish, conv3x3, instance), Sequential3(mish, conv3x3, batch)), Cell(Sequential3(hardswish, conv1x1, layer), id, Sequential3(hardswish, dconv3x3, batch), Sequential3(relu, conv3x3, layer), id, id), Cell(Sequential3(relu, conv3x3, instance), zero, id, zero, Sequential3(mish, conv3x3, batch), avg_pool)), Sequential3(Cell(zero, id, Sequential3(hardswish, conv1x1, layer), Sequential3(mish, conv3x3, instance), Sequential3(mish, conv3x3, instance), zero), Cell(Sequential3(hardswish, conv1x1, layer), id, Sequential3(hardswish, dconv3x3, batch), Sequential3(relu, conv3x3, batch), id, id), Cell(Sequential3(relu, conv3x3, instance), zero, id, zero, Sequential3(mish, conv3x3, layer), avg_pool)), Residual2(Cell(zero, id, zero, Sequential3(mish, conv3x3, layer), avg_pool, Sequential3(mish, conv3x3, layer)), down, down), Residual2(Cell(zero, id, zero, Sequential3(mish, conv3x3, batch), avg_pool, Sequential3(mish, conv3x3, layer)), down, down)), Residual3(Sequential3(Cell(Sequential3(mish, conv3x3, layer), id, Sequential3(hardswish, dconv3x3, layer), Sequential3(hardswish, dconv3x3, batch), Sequential3(mish, conv3x3, instance), Sequential3(mish, conv3x3, batch)), Cell(Sequential3(hardswish, conv1x1, layer), id, Sequential3(hardswish, dconv3x3, batch), Sequential3(relu, conv3x3, layer), id, id), Cell(Sequential3(relu, conv3x3, instance), zero, id, zero, Sequential3(mish, conv3x3, batch), avg_pool)), Sequential3(Cell(Sequential3(mish, conv3x3, batch), id, Sequential3(hardswish, conv1x1, batch), Sequential3(mish, conv3x3, instance), Sequential3(mish, conv3x3, instance), zero), Cell(Sequential3(hardswish, conv1x1, layer), id, Sequential3(hardswish, dconv3x3, batch), Sequential3(hardswish, dconv3x3, batch), id, id), Cell(Sequential3(relu, conv3x3, instance), zero, id, zero, Sequential3(mish, conv3x3, layer), avg_pool)), Residual2(Cell(zero, id, zero, Sequential3(mish, conv3x3, layer), avg_pool, Sequential3(mish, conv3x3, layer)),

down, down), Residual2(Cell(zero, id, zero, Sequential3(mish, conv3x3, batch), avg_pool, Sequential3(mish, conv3x3, layer)), down, down)))   .

ImageNet-16-120 (mean test error $52.78\,\%$, #params $0.626\,\mathrm{MB}$, FLOPs $23.771\,\mathrm{M}$):

Sequential3(Sequential4(Residual2(Cell(id, avg_pool, id, id, Sequential3(relu, dconv3x3, layer), zero), Cell(Sequential3(hardswish, conv1x1, batch), zero, zero, Sequential3(mish, dconv3x3, layer), zero, zero), Cell(Sequential3(relu, dconv3x3, layer), Sequential3(mish, dconv3x3, layer), zero, Sequential3(hardswish, conv3x3, layer), Sequential3(relu, dconv3x3, instance), Sequential3(hardswish, conv3x3, instance))), Sequential2(Cell(zero, Sequential3(relu, conv3x3, layer), Sequential3(mish, conv1x1, batch), Sequential3(mish, conv1x1, batch), avg_pool, Sequential3(relu, conv3x3, layer)), Cell(id, id, Sequential3(mish, conv3x3, layer), Sequential3(relu, conv3x3, instance), id, id)), Residual2(Cell(zero, avg_pool, Sequential3(mish, conv1x1, batch), Sequential3(mish, conv1x1, layer), zero, zero), Cell(id, Sequential3(relu, dconv3x3, layer), zero, zero, Sequential3(relu, dconv3x3, instance), zero), Cell(id, Sequential3(relu, conv3x3, layer), id, zero, zero, id)), Cell(zero, Sequential3(hardswish, conv3x3, layer), avg_pool, zero, Sequential3(hardswish, conv1x1, layer), id)), Residual3(Residual2(Cell(Sequential3(relu, conv1x1, instance), Sequential3(mish, conv1x1, layer), Sequential3(mish, conv1x1, instance), zero, Sequential3(hardswish, dconv3x3, layer), id), Cell(id, avg_pool, avg_pool, Sequential3(relu, conv1x1, instance), id, zero), Cell(avg_pool, Sequential3(mish, conv3x3, instance), Sequential3(mish, conv1x1, instance), Sequential3(relu, dconv3x3, batch), id, Sequential3(hardswish, conv3x3, instance))), Sequential2(Cell(zero, Sequential3(relu, conv3x3, layer), Sequential3(mish, conv1x1, batch), Sequential3(mish, conv1x1, batch), avg_pool, Sequential3(relu, conv3x3, instance)), Cell(id, zero, Sequential3(mish, conv3x3, layer), Sequential3(relu, conv3x3, instance), id, id)), Residual2(Cell(Sequential3(mish, conv3x3, layer), Sequential3(mish, conv1x1, batch), id, Sequential3(mish, conv1x1, layer), zero, id), down, down), Residual2(Cell(Sequential3(relu, conv3x3, layer), zero, Sequential3(relu, dconv3x3, layer), Sequential3(mish, conv1x1, layer), zero, id), down, down)), Residual3(Residual2(Cell(Sequential3(mish, conv1x1, instance), Sequential3(mish, conv1x1, layer), Sequential3(mish, conv1x1, instance), avg_pool, Sequential3(hardswish, dconv3x3, layer), id), Cell(id, avg_pool, avg_pool, Sequential3(relu, conv1x1, instance), id, zero), Cell(avg_pool, Sequential3(mish, conv3x3, instance), Sequential3(mish, conv1x1, instance), Sequential3(relu, dconv3x3, batch), id, Sequential3(hardswish, conv3x3, instance))), Sequential2(Cell(zero, Sequential3(relu, conv3x3, layer), Sequential3(mish, conv1x1, batch), Sequential3(mish, conv1x1, batch), avg_pool, Sequential3(relu, conv3x3, layer)), Cell(id, zero, Sequential3(mish, conv3x3, layer), Sequential3(relu, conv3x3, instance), id, id)), Residual2(Cell(Sequential3(relu, conv3x3, layer), avg_pool, id, Sequential3(mish, conv3x3, instance), zero, id), down, down), Residual2(Cell(Sequential3(relu, conv3x3, layer), zero, Sequential3(relu, dconv3x3, instance), Sequential3(mish, conv1x1, layer), zero, id), down, down)))   .

CIFARTile (mean test error $30.33\,\%$, #params $2.356\,\mathrm{MB}$, FLOPs $372.114\,\mathrm{M}$):

Sequential4(Residual3(Residual2(Cell(Sequential3(hardswish, conv3x3, instance), id, zero, Sequential3(relu, dconv3x3, instance), Sequential3(mish, conv1x1, instance), avg_pool), Cell(avg_pool, avg_pool, id, zero, Sequential3(hardswish, conv3x3, batch), avg_pool), Cell(Sequential3(relu, dconv3x3, instance), zero, id, Sequential3(relu, dconv3x3, layer), id, id)), Residual2(Cell(zero, zero, Sequential3(mish, conv1x1, instance), Sequential3(mish, conv3x3, batch), zero, id), Cell(Sequential3(mish, conv3x3, instance), zero, Sequential3(relu, dconv3x3, batch), id, Sequential3(mish, conv3x3, batch), id), Cell(Sequential3(hardswish, dconv3x3, batch), Sequential3(relu, conv3x3, batch), Sequential3(relu, conv1x1, batch), zero, Sequential3(relu, conv3x3, batch), id)), Sequential2(Cell(Sequential3(relu, dconv3x3, layer), Sequential3(mish, conv1x1, layer), id, zero, Sequential3(mish, conv3x3, batch), Sequential3(relu, dconv3x3, layer)), down), Sequential2(Cell(id, Sequential3(hardswish, conv1x1, layer), id, Sequential3(relu, conv1x1, instance), avg_pool, Sequential3(relu, conv1x1, layer)), down)), Residual3(Residual2(Cell(id, avg_pool, avg_pool, Sequential3(hardswish, dconv3x3, instance), Sequential3(mish, conv1x1, layer), Sequential3(hardswish, dconv3x3, instance)), Cell(id, id, Sequential3(relu, dconv3x3, layer), id, id, zero), Cell(Sequential3(relu, conv3x3, layer), id, avg_pool, Sequential3(mish, dconv3x3, instance), Sequential3(relu, conv1x1, layer), zero)), Residual2(Cell(Sequential3(mish, conv3x3, batch), Sequential3(mish, conv3x3, instance), zero, avg_pool, avg_pool, Sequential3(mish, conv1x1, batch)), Cell(Sequential3(mish, conv1x1, batch), Sequential3(relu, dconv3x3, layer), zero, id, avg_pool, avg_pool), Cell(avg_pool, Sequential3(hardswish, conv1x1, instance), id, avg_pool, avg_pool, Sequential3(hardswish, conv1x1, instance))), Residual2(Cell(Sequential3(relu, dconv3x3, batch), avg_pool, id,

avg_pool, id, zero), down, down), Residual2(Cell(zero, zero, Sequential3(relu, dconv3x3, batch), avg_pool, Sequential3(hardswish, conv1x1, instance), avg_pool), down, down)), Sequential4(Sequential3(Cell(Sequential3(hardswish, conv3x3, batch), Sequential3(hardswish, conv3x3, batch), Sequential3(relu, conv1x1, instance), id, Sequential3(relu, conv1x1, layer), Sequential3(relu, conv3x3, layer)), Cell(id, Sequential3(relu, conv3x3, instance), Sequential3(hardswish, conv1x1, instance), Sequential3(relu, conv3x3, layer), avg_pool, Sequential3(mish, conv1x1, layer)), Cell(zero, zero, id, Sequential3(relu, conv3x3, batch), id, Sequential3(relu, conv1x1, layer))), Sequential3(Cell(Sequential3(hardswish, conv3x3, batch), Sequential3(hardswish, conv3x3, batch), Sequential3(relu, conv1x1, instance), Sequential3(relu, dconv3x3, layer), Sequential3(mish, conv1x1, layer), Sequential3(relu, conv3x3, batch)), Cell(id, Sequential3(relu, conv3x3, instance), Sequential3(hardswish, conv1x1, instance), Sequential3(relu, dconv3x3, instance), avg_pool, Sequential3(mish, conv1x1, layer)), Cell(zero, zero, id, Sequential3(relu, conv3x3, batch), id, avg_pool)), Sequential3(Cell(id, id, avg_pool, Sequential3(mish, conv1x1, layer), Sequential3(mish, conv3x3, batch), zero), Cell(id, Sequential3(relu, conv1x1, batch), avg_pool, Sequential3(relu, conv1x1, layer), avg_pool, zero), Cell(zero, Sequential3(relu, conv1x1, batch), Sequential3(mish, dconv3x3, batch), Sequential3(mish, conv1x1, batch), id, id)), Cell(id, Sequential3(hardswish, conv1x1, layer), zero, id, zero, id)), Sequential3(Sequential2(Cell(id, zero, Sequential3(mish, dconv3x3, instance), Sequential3(mish, conv3x3, batch), Sequential3(mish, dconv3x3, instance), Sequential3(relu, conv1x1, instance)), Cell(Sequential3(relu, dconv3x3, instance), avg_pool, Sequential3(mish, conv1x1, instance), Sequential3(hardswish, dconv3x3, instance), id, Sequential3(hardswish, conv1x1, layer))), Sequential2(Cell(zero, zero, Sequential3(mish, dconv3x3, instance), Sequential3(relu, conv3x3, instance), Sequential3(hardswish, conv3x3, batch), avg_pool), Cell(id, id, Sequential3(hardswish, conv1x1, instance), avg_pool, zero, Sequential3(hardswish, conv3x3, batch))), Cell(avg_pool, Sequential3(mish, dconv3x3, layer), zero, avg_pool, avg_pool, zero)))     .

AddNIST (mean test error $6.33\%$, #params $2.853\,\mathrm{MB}$, FLOPs $593.856\,\mathrm{M}$):

Sequential4(Residual3(Sequential3(Cell(id, Sequential3(hardswish, dconv3x3, batch), Sequential3(relu, conv1x1, layer), Sequential3(mish, conv3x3, batch), avg_pool, zero), Cell(zero, zero, avg_pool, id, avg_pool, Sequential3(hardswish, conv1x1, instance)), Cell(Sequential3(relu, conv3x3, layer), id, zero, Sequential3(mish, conv3x3, instance), id, avg_pool)), Sequential2(Cell(id, Sequential3(relu, conv3x3, layer), Sequential3(relu, conv3x3, layer), Sequential3(hardswish, conv3x3, batch), id, Sequential3(relu, conv3x3, layer)), Cell(Sequential3(mish, conv3x3, instance), id, Sequential3(mish, conv3x3, batch), id, avg_pool, id)), Sequential3(Cell(zero, id, Sequential3(relu, dconv3x3, instance), Sequential3(relu, dconv3x3, layer), Sequential3(relu, dconv3x3, instance), Sequential3(mish, conv3x3, batch)), Cell(Sequential3(mish, conv1x1, instance), zero, Sequential3(relu, conv3x3, instance), id, zero, Sequential3(relu, conv3x3, batch)), down), Sequential3(Cell(zero, avg_pool, Sequential3(hardswish, dconv3x3, layer), Sequential3(relu, conv3x3, layer), Sequential3(hardswish, conv1x1, instance), Sequential3(hardswish, conv3x3, batch)), Cell(Sequential3(hardswish, conv3x3, batch), Sequential3(hardswish, conv1x1, layer), Sequential3(mish, conv1x1, batch), id, Sequential3(hardswish, conv3x3, batch), zero), down)), Residual3(Sequential2(Cell(Sequential3(mish, conv1x1, layer), avg_pool, Sequential3(hardswish, dconv3x3, batch), Sequential3(mish, dconv3x3, batch), id, Sequential3(mish, conv3x3, layer)), Cell(zero, Sequential3(relu, dconv3x3, layer), Sequential3(hardswish, conv3x3, instance), avg_pool, avg_pool, zero)), Sequential3(Cell(Sequential3(relu, conv3x3, batch), id, Sequential3(relu, conv3x3, layer), Sequential3(mish, conv1x1, instance), id, Sequential3(relu, dconv3x3, batch)), Cell(Sequential3(mish, conv3x3, batch), Sequential3(mish, conv1x1, instance), Sequential3(mish, conv3x3, instance), zero, Sequential3(mish, dconv3x3, layer), Sequential3(relu, conv3x3, batch)), Cell(avg_pool, Sequential3(mish, conv1x1, instance), Sequential3(relu, conv3x3, batch), avg_pool, id, Sequential3(mish, dconv3x3, batch))), Sequential3(Cell(zero, avg_pool, Sequential3(hardswish, dconv3x3, layer), Sequential3(relu, conv3x3, batch), Sequential3(hardswish, conv1x1, batch), Sequential3(hardswish, conv3x3, batch)), Cell(avg_pool, Sequential3(hardswish, dconv3x3, layer), Sequential3(mish, conv1x1, batch), id, Sequential3(hardswish, conv3x3, batch), zero), down), Residual2(Cell(zero, Sequential3(mish, conv1x1, instance), Sequential3(hardswish, conv1x1, instance), avg_pool, Sequential3(relu, conv1x1, layer), Sequential3(hardswish, dconv3x3, batch)), down, down)), Sequential4(Sequential2(Cell(Sequential3(relu, conv3x3, instance), id, Sequential3(relu, conv3x3, batch), avg_pool, zero, id), Cell(avg_pool, Sequential3(hardswish, conv3x3, layer), avg_pool, Sequential3(mish, conv3x3, batch), Sequential3(relu, conv3x3, batch), id)), Sequential2(Cell(Sequential3(mish, conv1x1, layer), avg_pool, Sequential3(hardswish,

dconv3x3, batch), Sequential3(mish, dconv3x3, batch), id, Sequential3(mish, conv3x3, layer)), Cell(zero, Sequential3(relu, dconv3x3, layer), Sequential3(hardswish, conv3x3, instance), avg_pool, avg_pool, zero)), Sequential2(Cell(id, Sequential3(relu, conv3x3, instance), Sequential3(relu, conv3x3, layer), Sequential3(hardswish, dconv3x3, batch), id, Sequential3(relu, conv3x3, layer)), Cell(Sequential3(mish, conv1x1, batch), id, avg_pool, id, avg_pool, id)), Cell(id, Sequential3(relu, conv3x3, layer), Sequential3(mish, conv1x1, instance), Sequential3(hardswish, conv3x3, batch), Sequential3(mish, dconv3x3, instance), Sequential3(hardswish, conv1x1, instance))), Sequential4(Sequential2(Cell(Sequential3(relu, conv3x3, instance), id, Sequential3(relu, conv3x3, batch), avg_pool, zero, id), Cell(zero, Sequential3(relu, conv3x3, batch), avg_pool, Sequential3(mish, conv3x3, batch), Sequential3(relu, dconv3x3, instance), id)), Sequential3(Cell(Sequential3(relu, conv3x3, batch), id, Sequential3(relu, conv3x3, layer), Sequential3(mish, conv1x1, layer), id, Sequential3(relu, dconv3x3, instance)), Cell(Sequential3(mish, conv3x3, batch), Sequential3(mish, conv1x1, instance), Sequential3(hardswish, dconv3x3, instance), zero, Sequential3(mish, dconv3x3, layer), Sequential3(relu, conv3x3, batch)), Cell(avg_pool, Sequential3(mish, conv1x1, instance), Sequential3(relu, conv3x3, batch), avg_pool, id, Sequential3(mish, dconv3x3, batch))), Sequential2(Cell(id, Sequential3(relu, conv3x3, layer), Sequential3(hardswish, conv3x3, layer), Sequential3(hardswish, dconv3x3, batch), id, Sequential3(relu, conv3x3, layer)), Cell(Sequential3(mish, conv3x3, batch), id, avg_pool, id, avg_pool, id)), Cell(id, Sequential3(relu, conv3x3, layer), Sequential3(mish, conv1x1, instance), Sequential3(hardswish, conv3x3, batch), Sequential3(mish, dconv3x3, instance), Sequential3(mish, conv3x3, instance)))) .

Below we also report the best found architecture per dataset on the hierarchical search space with more non-linear macro topologies (instead of linear macro topologies) and a single, shared cell (`hierarchical+shared+non-linear`). For sake of readability we report macro topology and cell separately. Different to the best architectures found in the hierarchical NAS-Bench-201 search space (`hierarchical`), the found architectures on this variant tend to have higher computational complexity (parameters and FLOPs).

CIFAR-10 (mean test error $5.02\,\%$, #params $4.82\,\text{MB}$, FLOPs $439.796\,\text{M}$):

Sequential3(Residual3(Residual2(Shared, Shared, Shared), Diamond2(Shared, Shared, Shared, Shared), Residual2(Shared, down, down), Residual2(Shared, down, down)), Residual3(Residual2(Shared, Shared, Shared), Diamond2(Shared, Shared, Shared, Shared), Diamond2(Shared, Shared, down, down), Diamond2(Shared, Shared, down, down)), Diamond3(Residual2(Shared, Shared, Shared), Residual2(Shared, Shared, Shared), Residual2(Shared, Shared, Shared), Residual2(Shared, Shared, Shared), Shared, Shared))

Cell(Sequential3(hardswish, conv3x3, layer), Sequential3(hardswish, conv3x3, layer), Sequential3(mish, conv3x3, layer), Sequential3(relu, conv3x3, batch), Sequential3(mish, conv3x3, instance), Sequential3(relu, conv1x1, batch)) .

CIFAR-100 (mean test error $25.41\,\%$, #params $2.68\,\text{MB}$, FLOPs $307.87\,\text{M}$):

Sequential4(Residual3(Residual2(Shared, Shared, Shared), Residual2(Shared, Shared, Shared), Residual2(Shared, down, down), Sequential2(Shared, down)), Residual3(Residual2(Shared, Shared, Shared), Residual2(Shared, Shared, Shared), Residual2(Shared, down, down), Sequential2(Shared, down)), Residual3(Residual2(Shared, Shared, Shared), Sequential2(Shared, Shared), Shared, Shared), Sequential3(Diamond2(Shared, Shared, Shared, Shared), Sequential2(Shared, Shared), Shared))

Cell(Sequential3(mish, conv3x3, batch), Sequential3(relu, conv3x3, batch), Sequential3(relu, conv3x3, batch), Sequential3(mish, conv3x3, batch), Sequential3(mish, dconv3x3, instance), id) .

ImageNet-16-120 (mean test error $48.16\,\%$, #params $0.852\,\text{MB}$, FLOPs $89.908\,\text{M}$):

Sequential3(Sequential3(Diamond2(Shared, Shared, Shared, Shared), Residual2(Shared, Shared, Shared), Shared), Diamond3(Residual2(Shared, Shared, Shared), Diamond2(Shared, Shared, Shared, Shared), Residual2(Shared, Shared, Shared), Diamond2(Shared, Shared, Shared, Shared), Diamond2(Shared, Shared, down, down), Diamond2(Shared, Shared, down, down)), Residual3(Residual2(Shared, Shared, Shared), Residual2(Shared, Shared, Shared), Residual2(Shared, down, down), Residual2(Shared, down, down)))

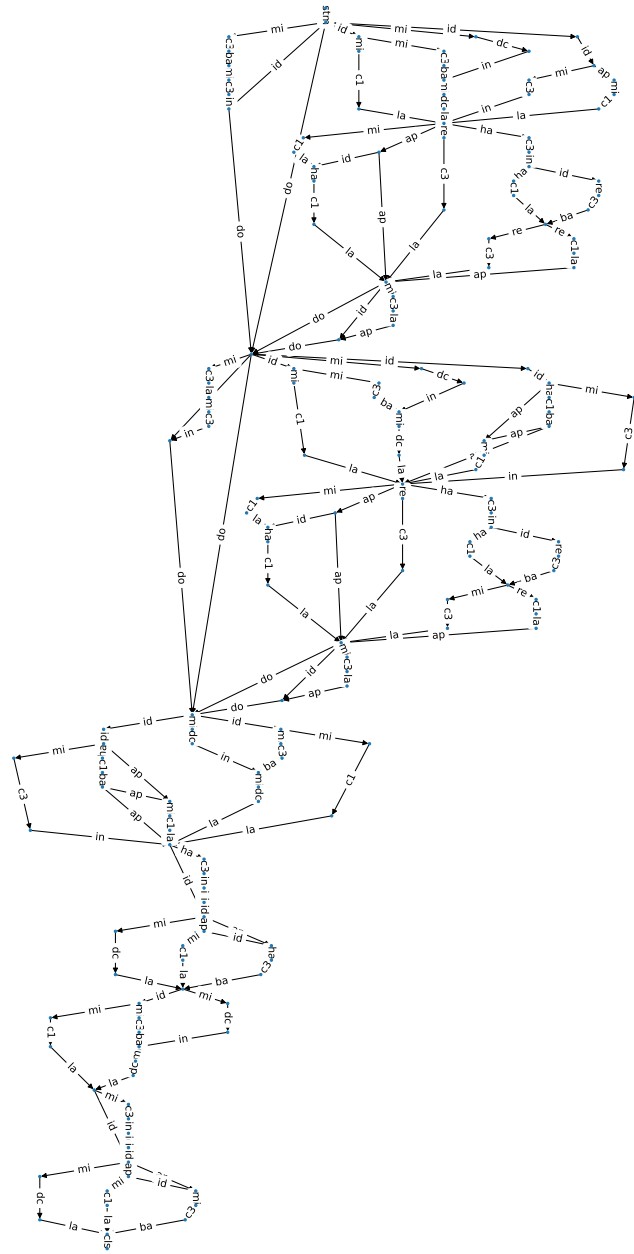

(a) CIFAR-10.

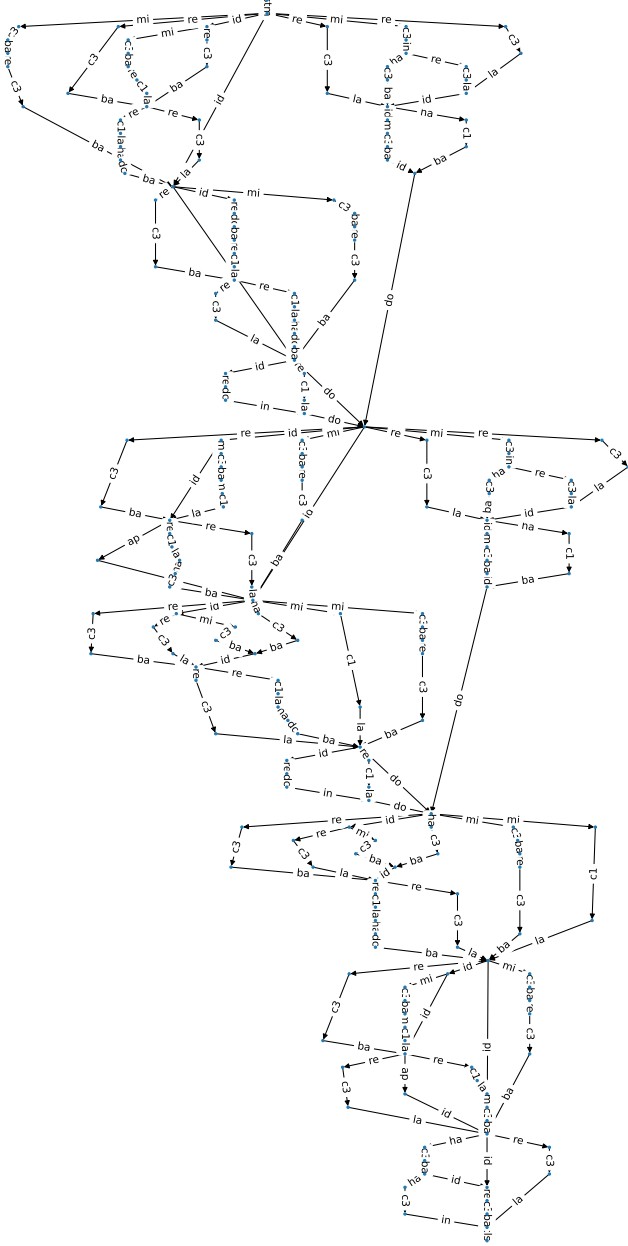

(b) CIFAR-100.

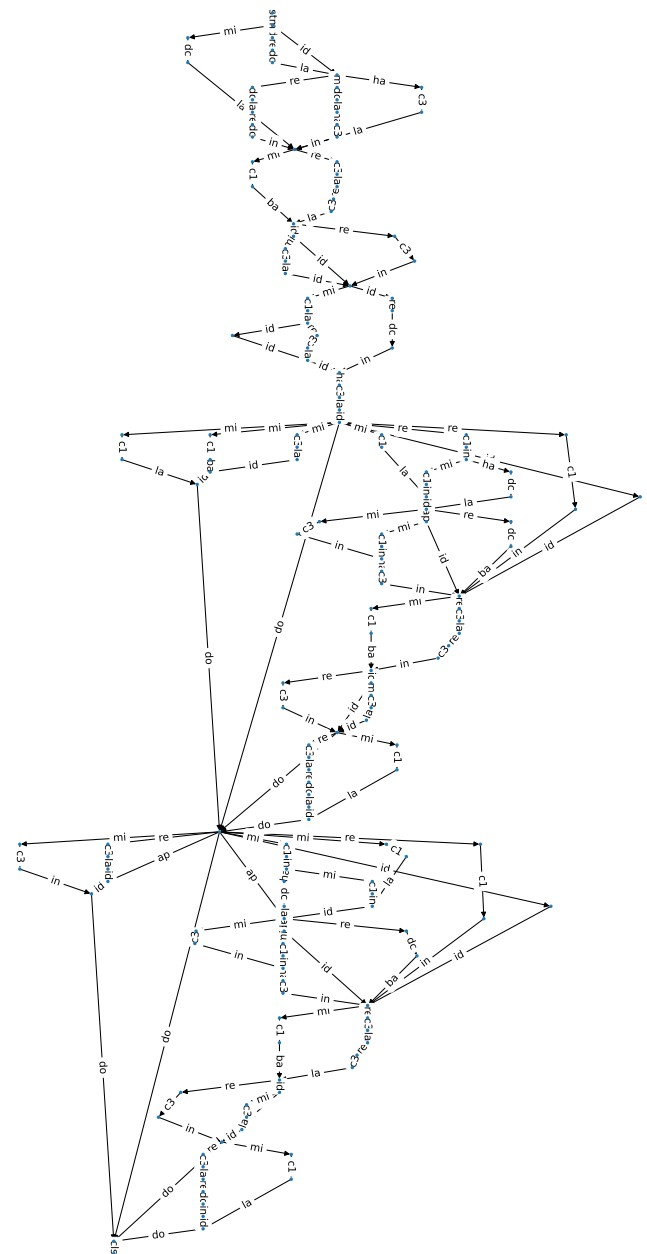

(c) ImageNet-16-120.

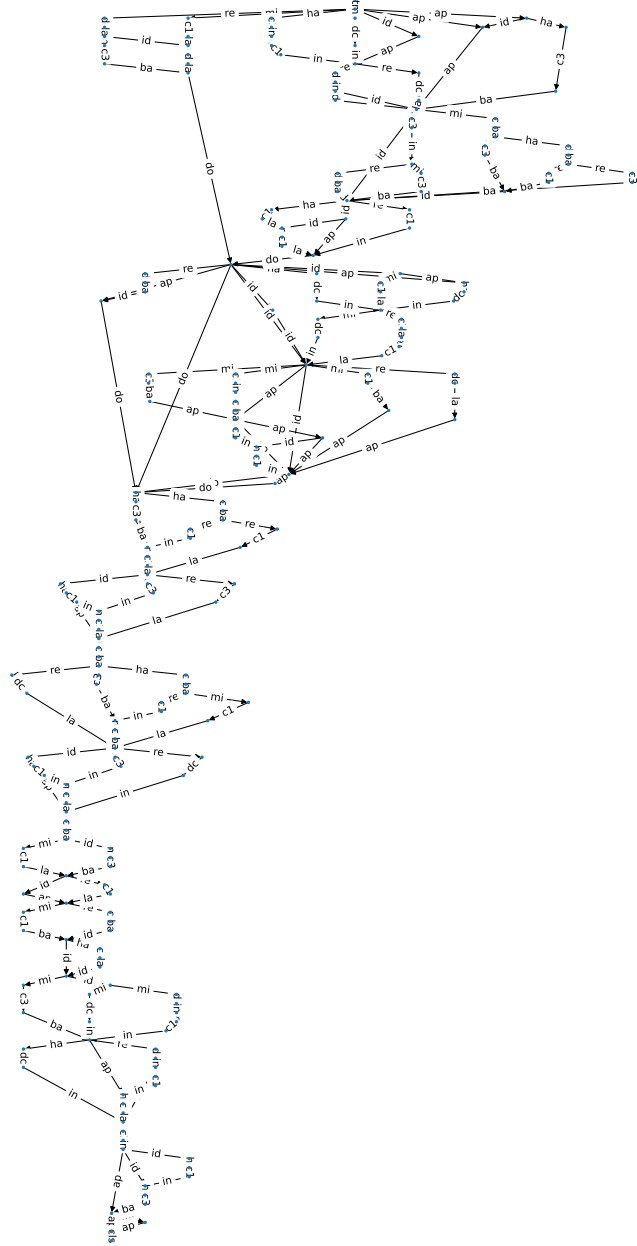

(d) CIFARTile.

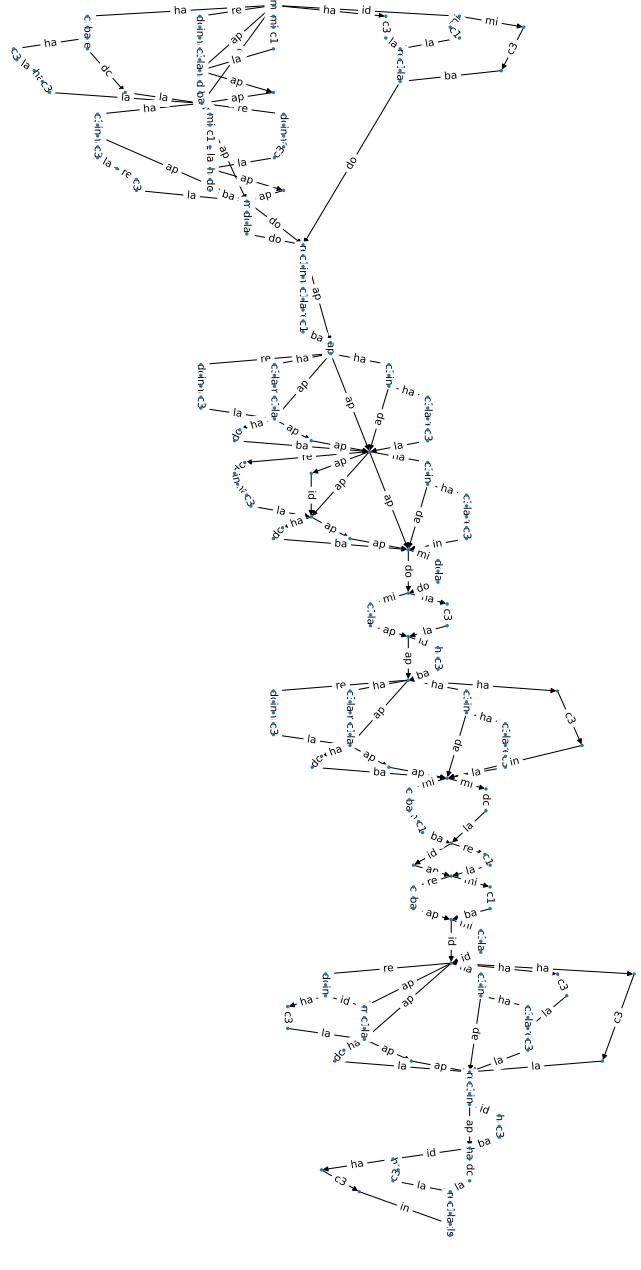

(e) AddNIST.

Figure 10: Visualization of the best found architectures in our hierarchical NAS-Bench-201 search space. Abbreviations are defined as follows: ap=avg_pool, ba=batch, c1=conv1x1, c3=conv3x3, cls=classifier, dc=dconv3x3, ha=hardswish, in=instance, la=layer, mi=mish, re=relu, and stm=stem. Best viewed with zoom.

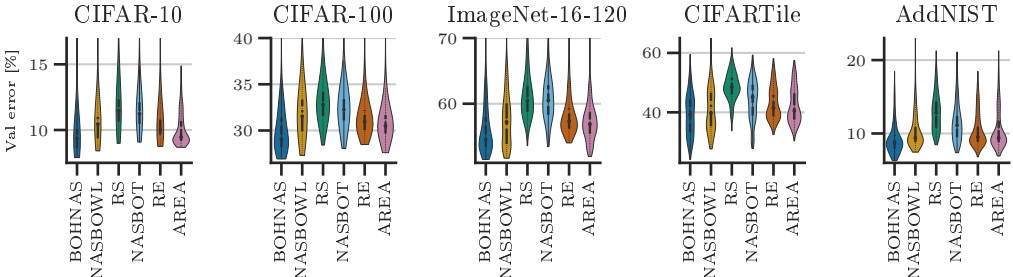

Figure 11: Density estimates of the validation errors of all architecture candidates proposed by the search strategies and across all datasets from our experiments in Section 6.

> Cell(Sequential3(mish, conv3x3, batch), Sequential3(mish, conv1x1, batch), Sequential3(relu, conv3x3, instance), Sequential3(mish, conv3x3, instance), Sequential3(relu, conv3x3, batch), id)    .

CIFARTile (mean test error $39.2\,\%$, #params $10.712\,\mathrm{MB}$, FLOPs $1519.944\,\mathrm{M}$):

> Sequential4(Sequential3(Residual2(Shared, Shared, Shared), Residual2(Shared, Shared, Shared), Diamond2(Shared, Shared, down, down)), Residual3(Sequential2(Shared, Shared), Residual2(Shared, Shared, Shared), Sequential2(Shared, down), Diamond2(Shared, Shared, down, down)), Diamond3(Sequential2(Shared, Shared), Sequential2(Shared, Shared), Diamond2(Shared, Shared, Shared, Shared), Sequential2(Shared, Shared), Shared, Shared), Diamond3(Sequential2(Shared, Shared), Sequential2(Shared, Shared), Sequential2(Shared, Shared), Sequential2(Shared, Shared), Shared, Shared))
>
> Cell(Sequential3(mish, conv3x3, batch), Sequential3(relu, conv3x3, instance), avg_pool, Sequential3(mish, conv3x3, layer), Sequential3(relu, dconv3x3, layer), Sequential3(mish, conv3x3, layer))    .

AddNIST (mean test error $4.57\,\%$, #params $3.345\,\mathrm{MB}$, FLOPs $347.959\,\mathrm{M}$):

> Sequential4(Diamond3(Diamond2(Shared, Shared, Shared, Shared), Diamond2(Shared, Shared, Shared, Shared), Diamond2(Shared, Shared, Shared, Shared), Diamond2(Shared, Shared, Shared, Shared), Residual2(Shared, down, down), Residual2(Shared, down, down)), Residual3(Diamond2(Shared, Shared, Shared, Shared), Diamond2(Shared, Shared, Shared, Shared), Residual2(Shared, down, down), Residual2(Shared, down, down)), Residual3(Diamond2(Shared, Shared, Shared, Shared), Diamond2(Shared, Shared, Shared, Shared), Shared, Shared), Sequential3(Sequential2(Shared, Shared), Diamond2(Shared, Shared, Shared), Shared))
>
> Cell(Sequential3(relu, conv3x3, batch), zero, avg_pool, Sequential3(hardswish, conv3x3, instance), Sequential3(hardswish, conv3x3, instance), Sequential3(hardswish, conv3x3, instance))    .

**Is BOHNAS exploring well-performing architectures during search?** To investigate this question, we visualized density estimates of the validation error of proposed candidates for all search strategies across our experiments on the hierarchical NAS-Bench-201 search space from Section 6. Figure 11 shows that BOHNAS explored better architecture candidates across all the datasets, i.e., it has smaller median validation errors and the distributions are further shifted towards smaller validation errors than for the other search strategies.

**Does incorporating hierarchical information improve performance modeling?** To further assess the modeling performance of our surrogate, we compared regression performance of GPs with different kernels, i.e., our hierarchical WL kernel (hWL), WL kernel [48], and NASBOT's kernel [18]. We also tried the GCN encoding [52] but it could not capture the mapping from the complex graph space to performance, resulting in constant performance predictions. Further, note that the adjacency encoding [50] and path encoding [47] cannot be used in our hierarchical search spaces since the former requires a fixed number of nodes and the latter scales exponentially in the number of nodes. We re-used the data from the search runs and ran 20 trials over the seeds {0, ..., 19}. In every trial, we sampled a training of varying number and test set of 500 architecture and validation

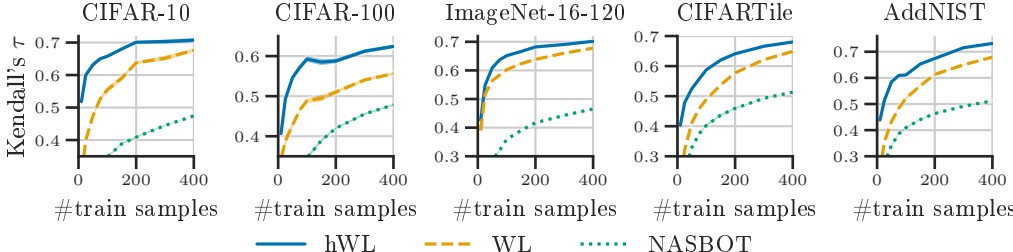

Figure 12: Mean Kendall's $\tau$ rank correlation with $\pm 1$ standard error achieved by a GP with various kernels.

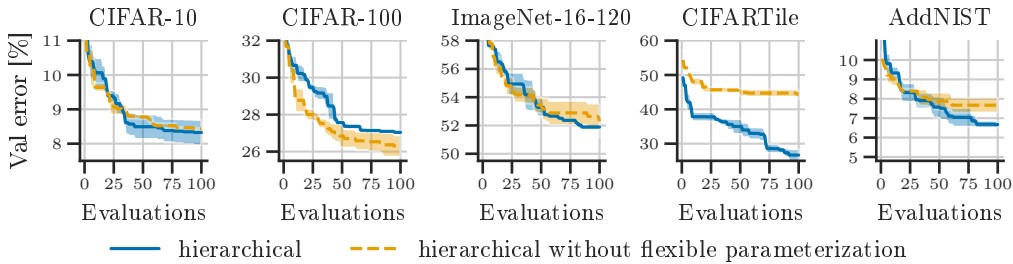

Figure 13: Impact of flexible parameterization of convolutional blocks in the hierarchical NAS-Bench-201 search space.

error pairs, respectively. We recorded Kendall's $\tau$ rank correlation between the predicted and true validation error.

Figure 12 clearly shows that the incorporation of the hierarchical information in the kernel design improves performance modeling. This is especially pronounced for smaller amounts of seen architectures.

**What is the impact of flexible parameterization of convolutional blocks?** To investigate the impact of the flexible parameterization of the convolutional blocks (i.e., activation functions, normalizations, and type of convolution), we removed the flexible parameterization and allowed only the same primitive computations as in the cell-based NAS-Bench-201 search space, while still searching over the macro architecture. More explicitly, we only allow ReLU non-linearity as the activation function, batch normalization as the normalization, and $1 \times 1$ or $3 \times 3$ convolutions. Figure 13 shows that for all datasets except CIFAR-100, flexible parameterization of the convolutional blocks improves performance of the found architectures. Interestingly, we find an architecture on CIFAR-100, which achieves $26.24\%$ test error with $1.307\,\mathrm{MB}$ and $167.172\,\mathrm{M}$ number of parameters or FLOPs, respectively. This architecture is superior to the optimal architecture in the cell-based NAS-Bench-201 search space ($26.49\%$). Note that this architecture is also Pareto-optimal for test error vs. number of parameters and test error vs. number of FLOPs.

**Do zero-cost proxies also work in vast hierarchical search spaces?** To assess zero-cost proxies, we re-used the data from the search runs and recorded Kendall's $\tau$ rank correlation. Table 7 shows that `nwot` is surprisingly the best-performing non-baseline zero-cost proxy, even though it is specifically crafted for architectures with ReLU non-linearities. Interestingly, the baseline zero-cost proxies often yield better results than other non-baseline zero-cost proxies. Thus, expressive hierarchical search spaces bring new challenges for the design of zero-cost proxies.

**Search space comparisons** Supplementary to Figure 3, Figure 14 compares the search space variants `hierarchical` vs. `fixed+shared` (`cell-based`) vs. `hierarchical+shared+nonlinear` using RS, RE, and NASBOWL. The cell-based search space design shows on par or superior performance on all datasets except for CIFARTile for the three search strategies. This is in stark contrast to the results using BOHNAS (Figure 3). It shows that more expressive search spaces do not necessitate

Table 7: Kendall's $\tau$ rank correlation of zero-cost-proxies on our hierarchical NAS-Bench-201 space. Note that `zen` and `nwot` may not be suitable for our hierarchical search space, as they are specifically designed for architectures with batch normalization and ReLU non-linearity, respectively. We report exemplary search results for NASWOT [65] in Table 6.

| Zero-cost proxy | Type | CIFAR-10 | CIFAR-100 | ImageNet-16-120 | CIFARTile | AddNIST |
|---|---|---|---|---|---|---|
| plain [66] | Baseline | -0.01 | 0.0 | -0.08 | 0.02 | -0.04 |
| params [67] | Baseline | 0.05 | 0.24 | 0.37 | 0.09 | 0.14 |
| flops [67] | Baseline | 0.16 | 0.38 | 0.38 | 0.36 | 0.33 |
| l2-norm [66] | Baseline | **0.58** | 0.42 | 0.29 | 0.44 | **0.63** |
| zen-score [101] | Piece. Lin. | 0.54 | 0.5 | 0.42 | 0.27 | 0.45 |
| fisher [102] | Pruning-at-init | -0.24 | -0.21 | -0.13 | -0.12 | 0.01 |
| grad-norm [66] | Pruning-at-init | -0.17 | -0.14 | -0.08 | -0.01 | 0.06 |
| grasp [103] | Pruning-at-init | -0.02 | 0.07 | 0.05 | -0.01 | -0.0 |
| snip [104] | Pruning-at-init | -0.1 | -0.15 | -0.11 | 0.05 | 0.13 |
| synflow [105] | Pruning-at-init | 0.09 | 0.35 | 0.42 | -0.21 | -0.11 |
| epe-nas [106] | Jacobian | 0.11 | 0.17 | 0.26 | 0.15 | 0.03 |
| jacov [65] | Jacobian | 0.3 | 0.29 | 0.2 | 0.21 | 0.17 |
| nwot [65] | Jacobian | 0.36 | **0.58** | **0.73** | **0.46** | 0.21 |

Table 8: Vocabulary for the binary and unary operations in the activation function search space. The symbol $\beta$ indicates a per-channel trainable parameter and $\sigma(\cdot)$ is the sigmoid function.

| | | | | | | | | | |
|---|---|---|---|---|---|---|---|---|---|
| add | $x_1 + x_2$ | bgaussian_sq | $\exp(-\beta(x_1-x_2)^2)$ | cubic | $x^3$ | cos | $\cos x$ | umax | $\max(x,0)$ |
| multi | $x_1 \cdot x_2$ | bgaussian_abs | $\exp(-\beta|x_1-x_2|)$ | square_root | $\sqrt{x}$ | sinh | $\sinh x$ | umin | $\min(x,0)$ |
| sub | $x_1 - x_2$ | wavg | $\beta x_1 + (1-\beta)x_2$ | mconst | $\beta$ | cosh | $\cosh(x)$ | sigmoid | $\sigma(x)$ |
| div | $\frac{x_1}{x_2+\epsilon}$ | id | $x$ | aconst | $x+\beta$ | tanh | $\tanh(x)$ | logexp | $\log(1+\exp(x))$ |
| bmax | $\max(x_1,x_2)$ | neg | $-x$ | log | $\log(|x|+\epsilon)$ | asinh | $\sinh^{-1}(x)$ | gaussian | $\exp(-x^2)$ |
| bmin | $\min(x_1,x_2)$ | abs | $|x|$ | exp | $\exp(x)$ | atanh | $\tanh^{-1}(x)$ | erf | $\mathrm{erf}(x)$ |
| bsigmoid | $\sigma(x_1) \cdot x_2$ | square | $x^2$ | sin | $\sin(x)$ | sinc | $\mathrm{sinc}(x)$ | const | $\beta$ |

an improvement in performance since a search strategy also needs to be able to such well-performing architectures. It also shows the superiority of BOHNAS over the other NAS approaches (**Q1**) and provides further evidence that the incorporation of hierarchical information is a key contributor for the search efficiency (**Q2**). Based on this, we believe that future work using, e.g., graph neural networks as a surrogate, may benefit from also incorporating hierarchical information.

For the hierarchical search space with more non-linear macro topologies (instead of linear macro topologies) and a single, shared cell (`hierarchical+shared+non-linear`), we find a similar pattern as in Figure 3. We believe that one reason for this is that the architectural regularization (sharing of a single cell) reduces search complexity.

# K    Searching for activation functions

## K.1    Search space

To search for activation functions, we adopted the search space from Ramachandran et al. [64]. More specifically, we define the search space as follows:

$$
\begin{aligned}
\text{L2} \quad &::= \quad \text{BinTopo(L1, L1, BINOP)} \mid \text{UnTopo(L1)}\\
\text{L1} \quad &::= \quad \text{BinTopo(UNOP, UNOP, BINOP)} \mid \text{UnTopo(UNOP)}\\
\text{BINOP} \quad &::= \quad \text{add} \mid \text{multi} \mid \text{sub} \mid \text{div} \mid \text{bmax} \mid \text{bmin} \mid \text{bsigmoid} \mid \text{bgaussian\_sq}\\
&\qquad \mid \text{bgaussian\_abs} \mid \text{wavg}\\
\text{UNOP} \quad &::= \quad \text{id} \mid \text{neg} \mid \text{abs} \mid \text{square} \mid \text{cubic} \mid \text{square\_root} \mid \text{mconst} \mid \text{aconst}\\
&\qquad \mid \text{log} \mid \text{exp} \mid \text{sin} \mid \text{cos} \mid \text{sinh} \mid \text{cosh} \mid \text{tanh} \mid \text{asinh} \mid \text{atanh} \mid \text{sinc}\\
&\qquad \mid \text{umax} \mid \text{umin} \mid \text{sigmoid} \mid \text{logexp} \mid \text{gaussian} \mid \text{erf} \mid \text{const} \quad,
\end{aligned} \tag{34}
$$

where the vocabulary for the binary operations (BINOP) and unary operations (UNOP) is provided in Table 8 and `BinTopo` and `UnTopo` define the topological operators of binary or unary operations, respectively. The search space comprises ca. $10^{8.6}$ potential activation functions. Note that the search space size can be varied by adding more levels or can be made open-ended by using a recursive formulation, i.e., $\text{L} ::= \text{BinOp(L, L)} \mid \text{UnOp(L)} \mid \text{id}$.

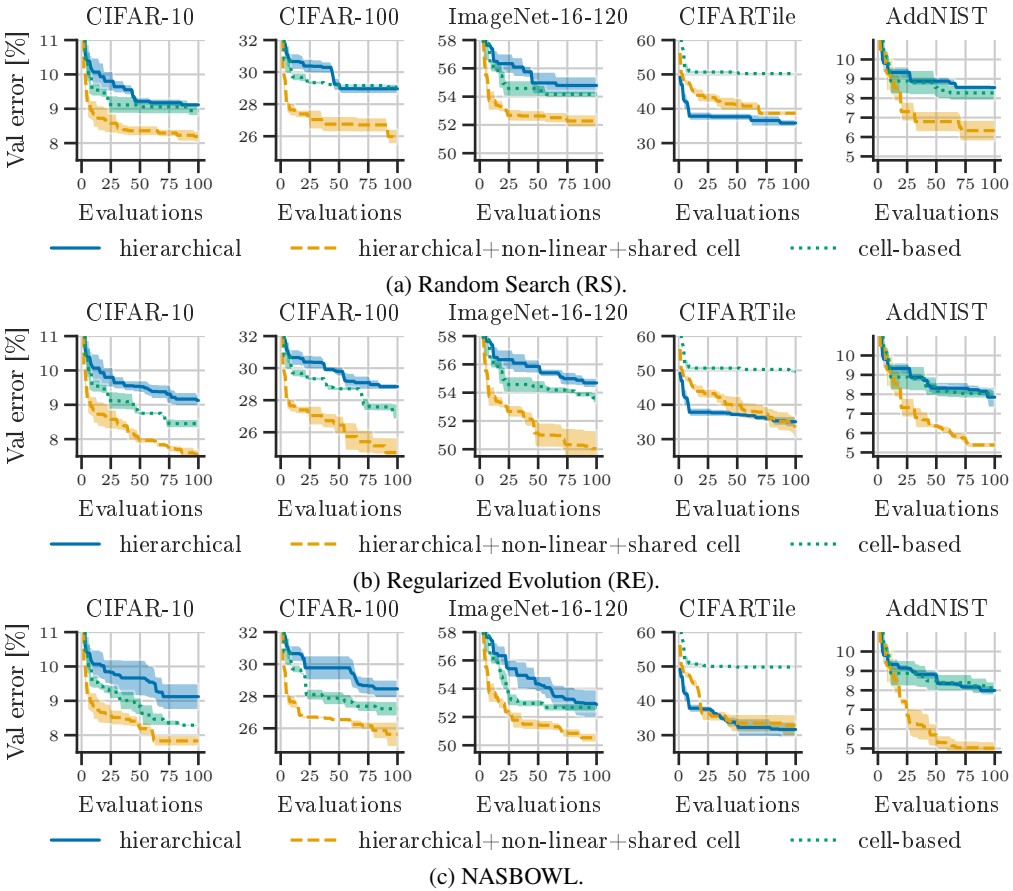

Figure 14: Comparison of mean and $\pm 1$ validation error of hierarchical, hierarchical with shared cell, and cell-based search space designs using Random Search (RS) (a), Regularized Evolution (RE) (b), and NASBOWL (c) for search.

## K.2 Training details

Similar to Ramachandran et al. [64], we trained a ResNet-20 [100] on CIFAR-10 [93]. Specifically, we trained the network with SGD with learning rate of 0.1, momentum of 0.9, and weight decay of 0.0001, with a batch size of 128 for 64K steps. The learning rate is divided by 10 after 32K and 48K steps. We used random flip with probability 0.5 followed by a random crop (32x32 with 4 pixel padding) and normalization. We split the training data into 45K training or 5K validation samples, respectively, and report the final test error on the unseen test set (10K samples).

## K.3 Supplementary search results

**Choice of search strategies** We only ran experiments using BOHNAS, RS, RE, and NASBOWL. We did not ran experiments using AREA since its zero-cost proxy specifically uses the binary codes of the ReLU non-linearity. We did not ran NASBOT since it already performed poorly on the hierarchical NAS-Bench-201 search space.

**Search costs** The search cost was ca. 5 GPU days using eight asynchronous workers, each with an NVIDIA RTX 2080 Ti GPU, for ca. 40 GPU days in total.

**Best found activation function** Our best found activation function for ResNet-20 is as follows:

$$\beta(\sigma(-x) \cdot \min(x, 0)) + (1 - \beta)(\min(\max(x, 0), \operatorname{erf} x)) \quad . \tag{35}$$

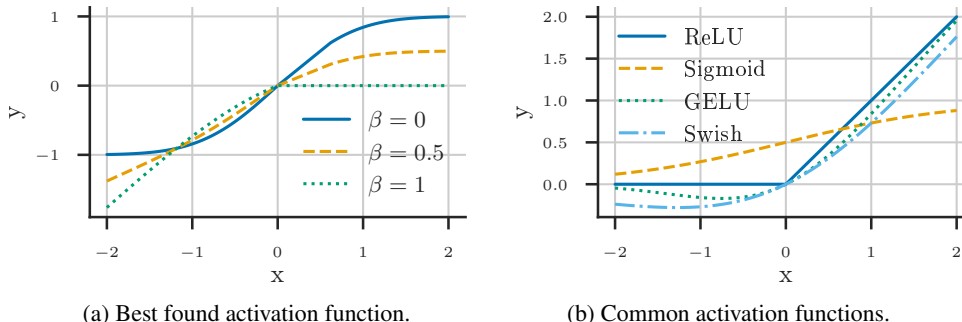

(a) Best found activation function.      (b) Common activation functions.

Figure 15: Visualization of our best found activation function (see Equation 35) for different values of the trainable parameter $\beta$ (left) and common activation function from the literature (right).

Figure 15 visualizes and compares the found activation function with common activation functions from the literature. The found activation function is a weighted mixture of the sigmoid function and ReLU non-linearity.

## L Searching for convolutional cells in the DARTS search space

While our main focus is to go beyond cell-based search spaces in this work, we also demonstrate that our proposed search strategy BOHNAS works in popular cell-based search spaces, e.g., cell-based NAS-Bench-201 [58] in Section 6. In addition, we also evaluated BOHNAS on the popular DARTS search space [6]. The DARTS search space definition is provided in Equation 13 (Appendix F). Following Ru et al. [48], we used the ENAS-style node-attributed DAG representation to apply the WL graph kernel on the normal or reduction cells, respectively.

### L.1 Implementation details

In contrast to all other experiments, we set the crossover probability to $p_{cross} = 0.0$ (and consequently $p_{mut} = 1.0$), as Schneider et al. [107] showed the importance of mutation[3] for the performance of BANANAS [47] on NAS-Bench-301 (surrogate benchmark of the DARTS search space, Zela et al. [108]).

### L.2 Training details

**Search settings**    Following Liu et al. [6], we split the CIFAR-10 training dataset into a train and validation set with 25k samples each. We trained a small network with 8 cells for 50 epochs, with a batch size of 64 and set the initial number of channels to 16. We used SGD with learning rate of 0.025 with cosine annealing Loshchilov and Hutter [96], momentum of 0.9, and weight decay of 0.0003. Since the validation performance on CIFAR-10 is very volatile, we averaged the final 5 validation errors, following Ru et al. [48]. In contrast to previous works, we picked the best normal and reduction cells based on the validation performance from the different search runs with random seeds $\{666, 777, 888, 999\}$, and did not pick the cell-based on its validation performance by training it from scratch, e.g., for a short period of 100 epochs [6].

**Retrain settings**    Following Liu et al. [6], we retrained the a large network with 20 cells for 600 epochs with batch size 96. The initial channels were set to 36. Other hyperparameters are same as used during search. We also applied cutout and auxiliary towers with weight 0.4. Following Chen et al. [45], we set the path dropout probability to 0.3 (instead of 0.2). We used the entire CIFAR-10 training set (50k samples) for training and report the final accuracy on the test set (10k samples). We evaluated the architecture with our discovered normal and reduction cells for the same random seeds used for search. We additionally also retrained the reported architectures of NASBOWL [48],

---

[3]In their implementation only the incumbent is mutated, whereas we still ran evolution over multiple steps to remain consistent across our experiments.

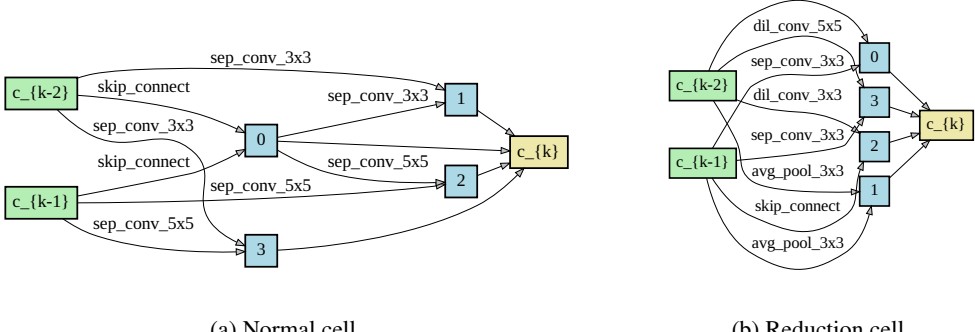

(a) Normal cell.                    (b) Reduction cell.

Figure 16: Normal and reduction cells discovered by BOHNAS on the DARTS search space on CIFAR-10.

DrNAS [45], and Shapley-NAS [89] to reduce effects of possible confounding factors such as choice of random seeds, software versions, hardware, etc.

To assess the transferability of our found normal and reduction cell on CIFAR-10, we trained the best cell on ImageNet following the mobile setting by Liu et al. [6], where the input image size is $224 \times 224$ and the number of multiply-add operations is restricted to be less than $600\,\mathrm{M}$. For training, we used the same random seeds used for search on CIFAR-10. Due to the high training cost, we omitted retraining of reported cells of prior works.

### L.3  Search results

**Search costs**  We ran search with eight asynchronous workers, each with an NVIDIA RTX 2080 Ti GPU, for exactly 1 GPU day, resulting in a total search cost of 8 GPU days. Note that by using the extensive parallel computing power available in today's computing clusters, the wallclock time for search (1 day) is comparable to that of state-of-the-art gradient-based methods. Retraining on CIFAR-10/ImageNet required ca. 1.5/5 GPU days on a single/eight NVIDIA RTX 2080 Ti GPU(s).

**CIFAR-10 Results**  Figure 16 shows the best found normal and reduction cell. Table 9 shows that BOHNAS can find well-performing normal and reduction cells that are on-par or (mostly) superior to other sample-based search strategies. The discovered normal and reduction cells are also competitive with state-of-the-art gradient-based methods, which are (typically) optimized for this particular search space.

**ImageNet results**  Table 10 shows that BOHNAS achieves competitive (or superior) results with $24.55\,\%$ and $7.4\,\%$ average top-1 or top-5 test error, respectively. This shows the transferability of the found normal and reduction cells by BOHNAS. The discovered normal and reduction cells are also superior to state-of-the-art gradient-based methods in the transfer setting[4], which are actually optimized for this particular search space since successive works target the DARTS search space. BOHNAS is only inferior to gradient-based methods that searched for the cells on ImageNet.

## M  Searching for transformers for generative language modeling and sentiment analysis

To demonstrate that our approach is also applicable to diverse architecture types and tasks, we searched for transformers for language modeling (nanoGPT experiment) as well as sentiment analysis.

---

[4]Note that for fair comparison, we compare only reproduced numbers to avoid confounding factors.

Table 9: Comparison with state-of-the-art image classifiers on CIFAR-10 on the DARTS search space.

| Search strategy | Type | Avg. val error [%] | Avg. test error [%] | Best test error [%] | Params [M] | Search cost [GPU days] |
|---|---|---|---|---|---|---|
| ASHA [109] | RS with early stopping | - | $3.03 \pm 0.13$ | 2.85 | 4.6 | 9 |
| Random-WS [7] | RS | - | $2.85 \pm 0.08$ | 2.71 | 4.3 | 10 |
| ENAS* [43] | RL | - | - | 2.89 | 4.6 | 6 |
| AmoebaNet-A [12] | evolution | - | $3.34 \pm 0.06$ | - | 3.2 | 3150 |
| AmoebaNet-B [12] | evolution | - | $2.55 \pm 0.05$ | - | 2.8 | 3150 |
| GP-NAS [110] | BO | - | - | 3.79 | 3.9 | 1 |
| BANANAS [47] | BO | - | 2.64 | 2.57 | - | 12 |
| BONAS-C [52] | BO | - | - | 2.46 | 3.48 | 7.5 |
| BONAS-D [52] | BO | - | - | 2.43 | 3.3 | 10 |
| NASBOWL [48] | BO | - | $2.61 \pm 0.08$ | 2.50 | 3.7 | 3 |
| NASBOWL$^\dagger$ [48] | BO | $11.84 \pm 0.15$ | $2.93 \pm 0.1$ | 2.83 | 3.7 | N/A |
| DARTS (v2) [6] | gradient | - | $2.76 \pm 0.09$ | - | 3.3 | 4 |
| GDAS [58] | gradient | - | - | 2.93 | 3.4 | 0.3 |
| DrNAS [45] | gradient | - | $2.46 \pm 0.03$ | - | 4.1 | 0.6 |
| DrNAS$^\dagger$ [45] | gradient | $12.19 \pm 0.06$ | $2.65 \pm 0.09$ | 2.51 | 4.1 | N/A |
| Shapley-NAS [89] | gradient | - | $2.47 \pm 0.04$ | 2.43 | 3.4/3.6 | 0.3 |
| Shapley-NAS$^\dagger$ [89] | gradient | $12.35 \pm 0.22$ | $3.09 \pm 0.07$ | 2.97 | 3.6 | N/A |
| BOHNAS | BO | $11.27 \pm 0.16$ | $2.68 \pm 0.12$ | 2.55 | 3.9 | 8 |

$^*$ expanded search space from DARTS. $^\dagger$ reproduced results using the reported genotypes. - not reported.

## M.1 Training details

**nanoGPT**  We followed the training protocol for Shakespearean works [98] of Karpathy [68] to train a character-based language model. Specifically, we trained the networks for 5000 iterations with batch size 64 with 100 warmup iterations and AdamW optimizer [96] with learning rate of 0.001, weight decay of 0.1, $\beta_1$ of 0.9, and $\beta_2$ of 0.95 with decaying learning rate schedule. We equally split the data into training and validation data. The vocabulary is of size of 65. We set the block size to 256 and used a dropout rate of 0.2.

**Sentiment analysis**  For sentiment analysis on IMDb [99], we used a frozen BERT encoder [117] and searched for a classifier head (see Appendix M.2 for details). We trained the classifier head for five epochs with a batch size of 128 with the Adam optimizer with learning rate of 0.001, $\beta_1$ of 0.9, and $\beta_2$ of 0.999. We used a dropout rate of 0.25 and hidden dimensionality of 256.

## M.2 Search space definitions

**nanoGPT**  We constructed the search space around the default transformer architecture of Karpathy [68], excluding the embedding layers and the final couple of layers (layer normalization followed by a linear layer) for simplicity. The search space is defined as follows:

$$
\begin{aligned}
\text{S} &::= \texttt{Sequential3(L, L, L)} \mid \texttt{Sequential2(L, L)} \mid \texttt{Residual2(L, L, L)} \\
\text{L} &::= \texttt{Sequential2(T, T)} \mid \texttt{Sequential3(T, T, T)} \mid \texttt{Residual2(T, T, T)} \\
\text{T} &::= \texttt{TransformerBlock(NHEADS, MLPRATIO, ACT)} \\
\text{NHEADS} &::= \texttt{6} \mid \texttt{4} \mid \texttt{8} \\
\text{MLPRATIO} &::= \texttt{4} \mid \texttt{2} \mid \texttt{3} \\
\text{ACT} &::= \texttt{gelu} \mid \texttt{relu} \mid \texttt{silu} \quad .
\end{aligned}
\tag{36}
$$

Additionally, we searched for the embedding dimensionality $\{96, 192, 384\}$. However, since the embedding dimensionality acts globally on the architecture, we treated it as a categorical hyperparameter. To this end, we combined our hierarchical graph kernel (hWL) with a Hamming kernel in our search strategy, BOHNAS. This shows that both our search space design framework as well as search strategy, BOHNAS, can also be used for joint NAS and hyperparameter optimization.

Table 10: Comparison with state-of-the-art image classifiers on ImageNet in the mobile setting on the DARTS search space. We reproduced results for DrNAS [45] and ShapleyNAS [89] using the reported genotype on one seed (777).

| Search strategy | Type | Top-1 test error [%] | Top-5 test error [%] | Params [M] | Search cost [GPU days] |
|---|---|---|---|---|---|
| Inception-v1 [111] | manual | 30.1 | 10.1 | 6.6 | N/A |
| MobileNet [112] | manual | 29.4 | 10.5 | 4.2 | N/A |
| ShuffleNet $2\times$ (v1) [113] | manual | 26.4 | 10.2 | $\sim 5$ | N/A |
| ShuffleNet $2\times$ (v1) [114] | manual | 25.1 | - | $\sim 5$ | N/A |
| NASNet-A [23] | RL | 26.0 | 8.4 | 5.3 | 2000 |
| AmoebaNet-C [12] | evolution | 24.3 | 7.6 | 6.4 | 3150 |
| PNAS [115] | BO | 25.8 | 8.1 | 5.1 | 255 |
| BONAS-C [52] | BO | 24.6 | 7.5 | 5.1 | 7.5 |
| BONAS-D [52] | BO | 25.4 | 8.0 | 4.8 | 10 |
| MnasNet-92 [55] | RL | 25.2 | 8.0 | 4.4 | - |
| Random-WS† [7] | RS | $25.3 \pm 0.25$ | $92.2 \pm 0.15$ | $5.6 \pm 0.1$ | - |
| DARTS (v2) [6] | gradient | 26.7 | 8.7 | 4.7 | 1 |
| GDAS [58] | gradient | 26.0 | 8.5 | 5.3 | 0.3 |
| PC-DARTS [116] | gradient | 25.1 | 7.8 | 3.6 | 0.1 |
| P-DARTS† [116] | gradient | 24.6 | 7.5 | 4.9 | N/A |
| P-DARTS [116] | gradient | 24.4 | 7.4 | 4.9 | 0.3 |
| DrNAS*† [45] | gradient | 24.3 | 7.4 | 5.7 | N/A |
| DrNAS* [45] | gradient | 23.7 | 7.1 | 5.7 | 4.6 |
| Shapley-NAS† [89] | gradient | 24.9 | 7.6 | 5.1 | N/A |
| Shapley-NAS [89] | gradient | 24.3 | 7.6 | 5.1 | 0.3 |
| Shapley-NAS*† [89] | gradient | 25.0 | 7.6 | 5.4 | N/A |
| Shapley-NAS* [89] | gradient | 23.9 | - | 5.4 | 4.2 |
| BOHNAS | BO | $24.55 \pm 0.03$ | $7.44 \pm 0.04$ | 5.4 | 8 |
| BOHNAS (best) | BO | 24.48 | 7.3 | 5.4 | 8 |

* search run on ImageNet, otherwise on CIFAR-10. † reproduced results using the reported genotypes.  - not reported.

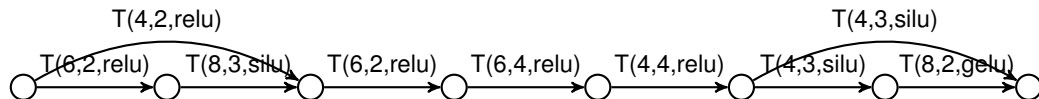

Figure 17: Best found transformer architecture (excluding embedding layers and final few layers) with an embedding dimensionality of 192 for the nanoGPT experiment.

**Sentiment analysis** We searched for a classifier head on top of the BERT encoder [117] followed by a linear projection. Specifically, we define the search space as follows:

$$\text{CELL} ::= \text{Cell}(\text{OPS, OPS, OPS, OPS, OPS, OPS})$$
$$\text{OPS} ::= \text{id} \mid \text{zero} \mid \text{transformer} \mid \text{gru} \mid \text{mlp} \quad , \tag{37}$$

where the topological operator (CELL) corresponds to a densely-connected four node DAG (similar to the NAS-Bench-201 cell-based search space [58]), and `transformer`, `gru`, and `mlp` are a transformer block, GRU unit, or two-layer MLP with GeLU non-linearity, respectively. We feed the output of the searched classifier cell into a GRU unit followed by dropout (with a rate of 0.25) and final linear layer.

## M.3 Search results

**Search cost** We searched with eight asynchronous workers, each with an NVIDIA RTX 2080 Ti GPU, for exactly 1 GPU day, resulting in a total search cost of 8 GPU days.

**Found transformer and classifier head** We depict the found transformer for the nanoGPT experiment in Figure 17. Figure 18 depicts the found classifier head for sentiment analysis.

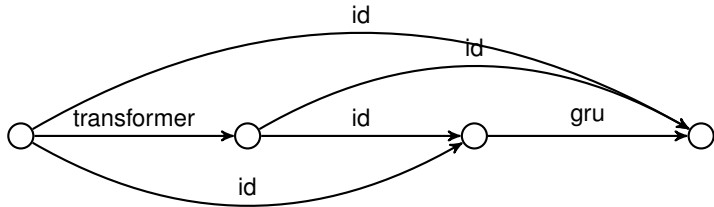

Figure 18: Found classifier head for the sentiment analysis experiment.

# N   Best practices checklist for NAS research

NAS research has faced challenges with reproducibility and fair empirical comparisons for a long time [109, 118]. In order to promote fair and reproducible NAS research, Lindauer and Hutter [13] created a best practices checklist for NAS research. Below, we address all points on the checklist.

1. **Best Practices for Releasing Code**
   (a) *Did you release code for the training pipeline used to evaluate the final architectures?, Did you release code for the search space?, Did you release code for your NAS method?* [Yes] All code can be found at https://github.com/automl/hierarchical_nas_construction.
   (b) *Did you release the hyperparameters used for the final evaluation pipeline, as well as random seeds?* [Yes] We provide experimental details (including hyperparameters as well as random seeds) in Section 6, Appendices J.1, K.2, L.2 and M.1.
   (c) *Did you release hyperparameters for your NAS method, as well as random seeds?* [Yes] We discuss implementation details (including hyperparameters as well as random seeds) of our NAS method in Section 6, Appendices I, L.1 and M.2.

2. **Best practices for comparing NAS methods**
   (a) *For all NAS methods you compare, did you use exactly the same NAS benchmark, including the same dataset (with the same training-test split), search space and code for training the architectures and hyperparameters for that code?* [Yes] We always used the same training pipelines during search and final evaluation.
   (b) *Did you control for confounding factors (different hardware, versions of deep learning libraries, different runtimes for the different methods)?* [Yes] We kept hardware, software versions, and runtimes fixed across our experiments to reduce effects of confounding factors.
   (c) *Did you run ablation studies?* [Yes] We compared different kernels (hWL, WL, NASBOT, and GCN) and investigated the surrogate performance in Section 6. We also analyzed the distribution of architectures during search, impact of the flexible parameterization of the convolutional blocks, test error vs. number of parameters and FLOPs in Appendix J.3.
   (d) *Did you use the same evaluation protocol for the methods being compared?* [Yes]
   (e) *Did you compare performance over time?* [Yes] We plotted results over the number of iterations, as typically done for black-box optimizers, and report the total search cost in Appendices J.3, K.3, L.3 and M.3. For the activation function search, DARTS cell search, and transformer-based searches, we did not compare performance over time, as this is typically not done for these experiments.
   (f) *Did you compare to random search?* [Yes]
   (g) *Did you perform multiple runs of your experiments and report seeds?* [Yes] We ran all search runs on the hierarchical or cell-based NAS-Bench-201 over three seeds, the search run for an activation function for one seed, the search on the DARTS search space on four seeds, transformer searches for one seed, and surrogate regression experiments over 20 seeds. All seeds are reported.
   (h) *Did you use tabular or surrogate benchmarks for in-depth evaluations?* [N/A] At the time of this work, there existed no surrogate benchmark for hierarchical NAS.

3. **Best practices for reporting important details**

   (a) *Did you report how you tuned hyperparameters, and what time and resources this required?* [N/A] We have not tuned any hyperparameter.

   (b) *Did you report the time for the entire end-to-end NAS method (rather than, e.g., only for the search phase?* [Yes] We report search times in Appendices J.3, K.3, L.3 and M.3. For the hierarchical NAS-Bench-201 experiments, the training protocols are exactly the same for the search and final evaluation (except for CIFAR-10), and, thus, there are no additional costs for final evaluation. For the activation function search experiment, we used the same training protocol for training and evaluation. For the DARTS experiment, we re-trained the discretized architecture on CIFAR-10/ImageNet which required ca. 1.5/5 GPU days on a single/eight NVIDIA RTX 2080 Ti GPU(s). For the nanoGPT experiment, we report the final error on the given validation set. For the sentiment analysis experiment, we used the same training and evaluation protocol.

   (c) *Did you report all the details of your experimental setup?* [Yes] We report all experimental details in Section 6, Appendices J.1, K.2, L.1, L.2 and M.1.

