# OpenReview forum: "Construction of Hierarchical Neural Architecture Search Spaces based on Context-free Grammars"
_NeurIPS.cc/2023/Conference — NeurIPS 2023 poster_

### Official Review · Reviewer_mjYa · 2023-07-05

**Soundness:** 3 good
**Presentation:** 3 good
**Contribution:** 3 good
**Rating:** 8
**Confidence:** 4

**Summary:**

In this paper, the authors focus on the search space issue in neural architecture search and propose a unifying search space design framework that models all candidate neural structures with context-free grammars. This approach enlarges the search space, which is more likely to find high-performance architectures, and meanwhile does not bring much search cost with the proposed Bayesian Optimization for hierarchical NAS. Experimental results show that systems with hierarchical search spaces outperform the previous NAS systems.

**Strengths:**

1. The issue that this article focuses on is precious. The search space is an important issue for NAS, and somewhat determines the upper bound of performance of the resulting system. The limited search space will make the system difficult to find a proper neural structure. The authors notice this problem and propose a new method to enlarge the space without much more search cost, which makes the approach more useful.
2. The proposed method is novel. There have already been some studies on NAS trying to introduce larger search space, but most are model specific. However, in this paper, the authors apply context-free grammars to model the search space, which is interesting and new. It’s easy for users to compose a considerable search space by allowing further derivations of the function composition.
3. The hierarchical search space is compatible with previous NAS methods.
4. The paper is well-written. The figures and descriptions are clear. It’s easy for readers, even researchers not in this area, to understand the main idea of this paper.


**Weaknesses:**

1. In the proposed hierarchical search space, it’s more complicated to design a search space because the space must be described in context-free grammars, which is a little bit difficult. Does it take a lot of experience to design a search space based on context-free grammars for tasks? If one wants to apply this method in some other tasks, such as natural language processing or speech processing, can this method be easily used? Are there any suggestions for designing grammar? I suggest the authors clarify these issues in the next version.

2. A related problem, which I think is important, is that the authors restrict the discussion to fixed-length problems. It is not obvious how the method is applied to variable-length problems, such as those in NLP. I guess the problem would be much different if we deal with a sequence of variable length, but it is often the case in recent LLMs.

3. The use of CFG often leads to a compact representation of exponentially many structures. In this sense, the method discussed here is similar to methods that make use of compact data structures, such as DARTS. The advantage of using CFGs is that the method is itself well explained and training CFG-based systems has long been studied. On the other hand, this makes the work have overlap with previous work.

4. Pruning is generally used in CFG-based systems. But it is not covered in this work. This is not a concern but a possible improvement to current work.



**Questions:**

Context-free grammars can well organize the search space and use clear rules to define a large search space, but what are the advantages of this approach for NAS? Although its search space has enlarged, a larger search space still requires more efficient search strategies to balance the search cost. What are the differences between your method and other previous work to expand the search space? For example, what is the main property of context-free grammars you think can help in NAS?

**Limitations:**

The authors have adequately addressed the limitations and discussed them in Section 7.

---

> ### Author Rebuttal · Authors · 2023-08-09
>
> We want to thank the reviewer for the encouraging comments and thoughtful questions. We are very happy that the reviewer finds our work “precious” and “easy for readers, even researchers not in this area, to understand the main idea of this paper.”
>
> By following the reviewer’s suggestion, we were able to demonstrate that our search space design framework can also easily be used also for “variable-length problems, such as those in NLP” by conducting experiments on two transformer-based search spaces on text (generative language modeling & sentiment analysis). Please refer to our general response for more details.
>
> We address the remaining questions and concerns below.
>
> > Differences to previous works on compact search space representations
>
> We acknowledge that there is some degree of overlap with previous methods that proposed compact search space representations. Nevertheless, we would like to point out some key differences from our work to previous works. Most importantly, our search space design allows searching over all parts of the architecture, from micro-level choices, such as which activation function to use for each layer, to macro-level choices. Previous works either considered only one of these, searched only for a confined set of macro topologies (e.g., only linear macro topologies), or required post-hoc validation procedures (e.g., checking for shape mismatches and fixing them). In contrast, our approach always generates valid entire architectures (according to the context-free grammar (CFG)). In addition, our approach supports the incorporation of user-defined architectural constraints as well as can effectively foster regularity. We provide a comparison of various search space design techniques in Table 4 in the supplement.
>
> > “What are the advantages of this approach for NAS?”; “What is the main property of context-free grammars you think can help in NAS?”
>
> Adding to the points mentioned above, we believe that CFGs provide a very formal way of treating search spaces and the NAS community could benefit from integrating ideas from the thoroughly studied field of grammars. Since CFGs generate and manipulate architectures in the text space, it also opens avenues of treating architectures not only as graphs but as strings; thereby enabling, e.g., the application of string kernels or potentially even providing an interface to leverage recent generative LLMs for architecture search.
>
> > “Does it take a lot of experience to design a search space based on context-free grammars [...]?”
>
> There will certainly be some familiarization time. Nevertheless, we are confident that researchers as well as practitioners alike will quickly get used to the search space design based on CFGs. To facilitate adaptation, we additionally provide a "Python" interface for designing CFGs without the requirement to get into the notational details of the extended Backus-Naur form [1].
>
> > “Are there any suggestions for designing grammar[s]?”
>
> The design of a CFG depends on the target task. That is, an image-based task may require a different design than a text-based task. The CFGs shown in our work can be a first step to get a better idea/inspiration for the own design of such a CFG.
>
> In general, when looking for high-performing architectures for a given task, we would always recommend reviewing the relevant literature and consulting domain experts if one is not particularly familiar with the application domain. In particular, we recommend distilling commonalities and differences among architectures that form the basis of the design for the CFG. Starting from this basis, we recommend expanding the search space by adding more and more production rules to the CFG to allow for the exploration of interesting novel architectures. This process of designing a CFG, especially in the beginning, may take some trial and error to get "right" but from our experience, it becomes easier with time. We will add to the revised manuscript a best practice guide for CFG design for NAS search spaces for researchers and practitioners alike.
>
> > “Pruning is generally used in CFG-based systems.”
>
> Thanks, this is indeed an interesting idea and could be a fruitful direction for future work!
>
> [1] JW, Backus. "The syntax and semantics of the proposed international algebraic language of the Zurich ACM-GAMM Conference." International Conference on Information Processing 1959.

---

> > ### Comment · Reviewer_mjYa · 2023-08-19
> > **Good work**
> >
> > Thank you for your response. My concerns have been addressed. As stated in my first review, i believe that this work is of good quality and should be accepted for publication of NeurIPS.

---

> > > ### Author Response · Authors · 2023-08-19
> > > **Re: Good work**
> > >
> > > Thank you for your reply! We are pleased that we were able to fully address all your concerns and appreciate that you believe that our work is of "good quality".

---

### Official Review · Reviewer_Q2bD · 2023-07-06

**Soundness:** 4 excellent
**Presentation:** 2 fair
**Contribution:** 3 good
**Rating:** 7
**Confidence:** 3

**Summary:**

The authors propose improvements to several parts of the neural architecture search (NAS) pipeline. First, they introduce a methodology for specifying search spaces based on a combinator grammar that is both more flexible than previous structured search spaces and encourages more regularity and hierarchy than previous unstructured ones. They show that this methodology allows them to represent all search spaces they’re aware of from the literature, and extend them in ways that are likely to be helpful. Then they describe a method for searching these spaces using Bayesian optimization with a multiscale graph kernel and evolutionary operations, and show that this method achieves better results on existing search spaces (e.g., NAS-Bench-201 and DARTS) and even better results on hierarchical extensions of these search spaces (hierarchical NAS-Bench-201).

**Strengths:**

The authors introduce good ideas and report strong results. They explicate and carefully follow NAS best practices, and use relatively little compute (only consumer GPUs!) while fostering reproducibility. Relative to a somewhat crowded field, the paper stands out as both careful and “obvious in retrospect”.

**Weaknesses:**

The work is somewhat sprawling and can’t go into enough depth on some of the most interesting parts; e.g. the constraints section that explains that the CFG isn’t actually context-free is one paragraph long in the main paper, and all full network examples are in the appendix. Overall I have to put a lot of work in to understand where your advances fit in the big picture, and exactly what parts contribute to improved results. It also feels a bit like the last hurrah of a somewhat fading subfield (both convnet architecture optimization and convnet-focused NAS are perhaps a bit tapped out).
A notational nit I have is that Residual(a, b, c) is a confusing way to write a structure where b is the residual for a and c. Is there any particular reason for this choice?

**Questions:**

How far are we from “architectures from the book”? Was anything surprising about the architectural structures your method discovered? You mention some architecture search spaces that can’t be covered with a CFG (can you give an example?); are any important architecture classes locked out because of these search spaces?

**Limitations:**

The authors adequately cover the limitations of their techniques. Two that particularly stood out to me are that the work assumes a computer vision/convnet context (e.g. special case handling of downsampling, no non-image-classification benchmarks) and lacks support for weight sharing.

---

> ### Author Rebuttal · Authors · 2023-08-09
>
> We want to thank the reviewer for the thorough review and thoughtful raised questions. We are glad that the reviewer finds that our “paper stands out as both careful and ‘obvious in retrospect’”. We are very happy that the reviewer acknowledges our “strong results”, while using “relatively little compute (only consumer GPUs!)”. We are pleased that the reviewer appreciates our efforts to foster reproducibility by “explicat[ing] and carefully follow[ing] NAS best practices”.
>
> By following the reviewer’s suggestions, we were able to demonstrate the generality beyond the, arguably “fading subfield” of conv-based NAS as well as vision tasks (“works assumes [...] vision/convnet context”) by conducting experiments on two transformer-based search spaces on text (generative language modeling and sentiment analysis). Please refer to our general response for more details.
>
> Below, we address the reviewer’s remaining questions and concerns.
>
> > “How far are we from ‘architectures from [scratch]’?”
>
> This is indeed a very interesting question! We believe that our work is one step closer toward searching for architectures from scratch by opening up search spaces to a broader spectrum of architectures. Nevertheless, there are several challenges for the NAS community working towards this goal. For example, the sheer size makes the search significantly more complex. Thus, we need to improve our search strategies in terms of efficiency or try to incorporate user priors to help kick-start the search process. A fruitful direction for future works could be to incorporate ideas from, e.g., multi-fidelity optimization or meta-learning. Besides the increased search complexity, search space design also is more challenging, e.g., how do we ensure that there are no shape mismatches. While our work proposes some techniques to remedy such difficulties in the search space design, there may be several more such challenges left to be solved in future works.
>
> > “Was anything surprising about the architectural structures your method discovered?”
>
> A surprising observation was that architectures with various branches at the macro-level often performed as well as or better than architectures with the common design of chaining similar layers or blocks (e.g., transformer or residual blocks) one after another. This prompts us (at least the authors of this work) to rethink the macro architectural design choices (for NAS) that we all adopted over the last decade; except for a couple of notable recent exceptions [1,2]. We were also genuinely surprised that different activation functions or normalizations throughout the architecture could work together and not lead to training instabilities or the like.
>
> > “You mention some architecture search spaces that can’t be covered with a CFG (can you give an example?)”
>
> In principle, all search spaces could be covered by *some* CFG, but there is *not one* CFG that can cover all architectures. To give an example, consider architectures of the form $a^n b^n c^n$, where a, b, and c are some layers (e.g., a is a convolution, b is a transformer block, c is an MLP), each repeated n times. For example, for n=2 we would have aabbcc. However, through the pumping lemma for context-free languages [3], we can show that no single context-free grammar can construct all architectures of the form $a^n b^n c^n$.
>
> > “Lacks support for weight sharing”
>
> Yes, we acknowledge that our current implementation does not support weight sharing. The main reason is that current weight-sharing approaches do not scale to large hierarchical search spaces, i.e., their supernet would require an exponential number of weights. Please refer to Appendix C for a more extensive discussion.
>
> > “The work is somewhat sprawling and can’t go into enough depth”
>
> We make the points raised by the reviewer clearer and more clearly emphasize how our advances fit into the bigger picture in the revised manuscript.
>
> Regarding the notation of topological operators (e.g., Residual(a, b, c)): we associate edges with the arguments and order the arguments first by the sink nodes of each edge and then by the source node (if multiple edges have the same sink node). For Residual, we have the edges (1,2), (1,3), & (2,3), and thus b is the residual connection in our case. However, we are willing to change this scheme if the reviewer thinks there is some other more intuitive scheme.
>
> [1] Goyal, Ankit, et al. "Non-deep networks." NeurIPS 2022.
>
> [2] Touvron, Hugo, et al. "Three things everyone should know about vision transformers." ECCV 2022.
>
> [3] Bar-Hillel, Y.; Micha Perles; Eli Shamir (1961). "On formal properties of simple phrase-structure grammars".

---

> > ### Comment · Reviewer_Q2bD · 2023-08-19
> >
> > Thank you for your response! I'd encourage incorporating a few of the points you make into the paper, in particular the "surprising observation" and "topological operator notation" responses.
> >
> > The transformer experiment is an exciting and significant addition to the paper as well.

---

> > > ### Author Response · Authors · 2023-08-19
> > > **Re: Official Comment by Reviewer Q2bD**
> > >
> > > Thank you for your reply! We are happy that you find the transformer experiments an "exciting and significant addition" and will include them, as well as the other points you mentioned, in the revised paper.

---

### Official Review · Reviewer_gqtA · 2023-07-07

**Soundness:** 3 good
**Presentation:** 3 good
**Contribution:** 3 good
**Rating:** 7
**Confidence:** 2

**Summary:**

This paper uses context-free grammar to improve the efficiency of neural architecture search. Experiments show that the proposed approach outperforms existing methods.

**Strengths:**

- Simple, well-motivated and efficient approach to improve neural architecture search.
- Clear presentation.

**Weaknesses:**

No major weakness found, though the improvement presented in Table 2 seemed marginal.
In addition, as the authors discussed in Section 7, the expressiveness of CFGs are limited to context-free languages, which may exclude the optimal architecture.

**Questions:**

N/A

**Limitations:**

The authors have discussed the limitation of this paper: while adopting CFGs may limit the search space to context-free languages, empirical results have demonstrated effectiveness of the proposed approach.

---

> ### Author Rebuttal · Authors · 2023-08-09
>
> We thank the reviewer for taking the time to review our work. We are glad that the reviewer finds our approach “simple, well-motivated and efficient”. Below, we address an interesting point raised by the reviewer in more detail.
>
> > Exclusion of the optimal architecture from the search space (as discussed started discussing in Section 7 of the main paper)
>
> This is indeed a great point, since the search space always defines the upper bound of the best-performing architecture we can find for a certain task. This is true for all types of search spaces, including our hierarchical search spaces based on context-free grammars, except for global search spaces, where we could indeed search over all architectures. However, search costs may be much higher, e.g., there are many architectures with poor performance, while the best-performing architectures may be only found in tiny subsets of the entire search space. Adding to the aforementioned, we would like to note that even if we ensure that the optimal architecture could be found hypothetically, there is no guarantee that any search strategy is able to actually find it with a limited computational budget.

---

### Official Review · Reviewer_sktw · 2023-07-11

**Soundness:** 3 good
**Presentation:** 3 good
**Contribution:** 3 good
**Rating:** 6
**Confidence:** 3

**Summary:**

The authors propose an approach to unify the search space for Neural Architecture Search (NAS) using context-free grammars. They further develop a Bayesian Optimization based search strategy to effectively explore a huge search space. They experiment over (proposed) hierarchical extension of NAS-Bench 201 benchmark to demonstrate sound improvements.



**Strengths:**

Hierarchical search space design allows exploration of more controlled and viable architectures. While prior works mostly explored cell-based search, or require post-hoc checking, the authors develop a principled solution for macro to micro search using context-free grammars (CFG).

In order to demonstrate their search granularity, the authors propose a hierarchical extension to NAS-Bench 201, and subsequently perform most of the experiments in the modified benchmark. The paper would benefit from also highlighting these results on the original NAS-Bench 201 benchmark to understand how the granularity of the search space impacts the different search methods in the traditional benchmark.

In the modified hierarchical benchmark, the authors demonstrate strong performance improvements against several baseline methods. In Table 1, the authors contrast their search strategy against existing search strategies on multiple dimensions.

**Weaknesses:**

1. While BOHNAS in general works well across multiple settings and outperforms several baselines, they do have limited gains when compared against DARTS and NASBOWL.
2. All the experiments and benchmarks in the paper focus on Conv based search spaces. Given that a lot of recent works, primarily on language models, focus on Transformer search spaces, a related discussion would be useful for the readers.
3. It will be nice to have results on larger datasets including the full ImageNet.

**Questions:**

See above

---

> ### Author Rebuttal · Authors · 2023-08-09
>
> We thank the reviewer for the review. We are happy that the reviewer views our approach as a “principled solution for macro to micro search” and that it “allows exploration of more controlled and viable architectures”.
>
> By following the reviewer’s suggestion, we additionally conducted experiments on transformer-based search spaces to showcase the general applicability and versatility of our proposed search space design. Please refer to our general response for more details.
>
> Below, we address the remaining questions and criticisms of the reviewer.
>
> > Comparisons across various search space granularities, including the original cell-based NAS-Bench-201 benchmark
>
> This is indeed an interesting question that we investigated in Figure 3 of the paper. In this experiment, we compared various search spaces (including the original NAS-Bench-201 benchmark (fixed+shared)) across various datasets using BOHNAS. The results show that we can indeed find better-performing architectures in hierarchical search spaces compared to, e.g., cell-based search spaces (such as the original NAS-Bench-201 search space). Figure 13 in the supplement complements this experiment by using other search methods (RS, RE, & NASBOWL) with similar results. While this experiment is an initial step toward better understanding the effects of search space granularities, we believe that this is a promising direction for future works to explore further.
>
> > Performance gains of BOHNAS beyond cell-based search spaces
>
> We would like to highlight that our proposed search strategy, BOHNAS, clearly outperforms NASBOWL on hierarchical search spaces (e.g., see Figure 2 of the main paper). DARTS is not (yet?) even applicable to such vast hierarchical search spaces since the expressive hierarchical search spaces would yield an exponential number of weights (please refer to Appendix C for an extensive discussion). However, we acknowledge that in search spaces with fewer or no hierarchical levels (e.g., the DARTS search space), BOHNAS only yields architectures with smaller performance gains, but importantly not inferior ones (see Table 1 of the main paper).
>
> > Results on larger datasets including the full ImageNet
>
> We acknowledge that we currently do not have ImageNet results besides on the DARTS search space (please refer to Table 10 in the supplement). We would like to note that our approach is independent of the target task or data modality (also please refer to our general response). Given our presented results, we expect similar results on larger datasets but we do not have access to sufficiently large compute resources to train multiple neural networks that are competitive for ImageNet in the non-mobile setting.

---

### Author Rebuttal · Authors · 2023-08-09

We thank all the reviewers for their thorough review, thoughtful questions, and constructive criticisms. We are very glad to hear that reviewers find that our work is “precious” (mjYa), “principled” (sktw), “simple, well-motivated and efficient” (gqtA), as well as “stands out as both careful and ‘obvious in retrospect’” (Q2bD).

Two shared questions and concerns among reviewers (sktw, Q2bD, mjYa) are whether our search space design can also be applied to (i) transformer-based search spaces and (ii) non-vision tasks.

To answer these in the affirmative, we conducted two NLP experiments to search for (1.) a GPT-like transformer for generative language modeling and (2.) a classifier head on top of a frozen BERT encoder for sentiment analysis. These experiments further show the generality and versatility of our proposed search space design framework. Additionally, the nanoGPT experiment (1.) shows that both our search space design framework as well as search strategy, BOHNAS, can be used for joint NAS & HPO. We will include these experiments in our revised manuscript.

## 1. Searching for a transformer for generative language modeling

For this experiment, we searched for a transformer (excluding the embedding layers and the final few layers for simplicity) based on Andrej Karpathy’s nanoGPT for Shakespearean works [1]. Similar to our other experiments, we used 8 asynchronous workers (each with an NVIDIA RTX 2080 Ti GPU) for one day. We define the context-free grammar for this search space as follows:

```
S::= Sequential3(L, L, L)  |  Sequential2(L, L)  |  Residual2(L, L, L)
L ::= Sequential2(T, T)  |  Sequential3(T, T, T)  |  Residual2(T, T, T)
T ::= TransformerBlock(NHEADS, MLPRATIO, ACT)
HEADS ::= 6  |  4  |  8
MLPRATIO ::= 4  |  2  |  3
ACT ::= gelu  |  relu  |  silu     ,
```
where the topological operators (Sequential3, Sequential2, Residual2) are similarly defined as in our other experiments from our work. The operator TransformerBlock corresponds to the basic transformer block commonly used in transformers and is parameterized by the number of heads (NHEADS), MLP ratio (MLPRATIO), and activation function (ACT).

We additionally searched for the embedding dimensionality {96, 192, 384}. However, since the embedding dimensionality acts globally on the architecture, we treated it as a categorical hyperparameter. To this end, we combined our hierarchical graph kernel (hWL) with a Hamming kernel in our search strategy, BOHNAS.

Our search yielded a transformer with the best validation loss of 1.4386 with only 3.33M parameters. For comparison, Andrej Karpathy reported a best validation loss of 1.4697 with 10.65M parameters for his transformer. We depict the found transformer in Figure 1 of the attached pdf.

Future work could scale up the search (in terms of the search space, training data, model scale, training compute, and/or search compute) to find even more powerful, general-purpose generative LLMs.

## 2. Searching for the classifier head of a frozen BERT encoder for sentiment analysis

For this experiment, we used a frozen BERT encoder and searched for a classifier head for sentiment analysis of IMDb movie reviews. We ran the search using 8 asynchronous workers (again, each equipped with an NVIDIA RTX 2080 Ti GPU) for one day. We define the context-free grammar for this search space as follows:

```
S ::= Cell(OPS, OPS, OPS, OPS, OPS, OPS)
OPS ::= id  |  zero  |  transformer  |  gru  |  mlp
```

where the topological operator (Cell) corresponds to the NAS-Bench-201 cell topology, `id` is the identity function, `zero` is the zero function, `transformer` is a typical transformer block, `gru` is a gated recurrent unit (GRU), and `mlp` is a 2-layer MLP that increases and decreases dimensionality (we chose a factor of 2) with GELU activation function. For simplicity, we have omitted the linear projection of the BERT embedding and the final few layers (GRU -> Dropout -> Linear).

Our search yielded a classifier head that achieved a test accuracy of 92.26%. We depict the found classifier head in Figure 2 of the attached PDF. Although this does not achieve state-of-the-art numbers, we would like to note that we conducted this experiment only as proof of concept (e.g., not optimal training pipeline, etc.) and expect improved numbers by adopting more recent training techniques, e.g., further pre-training with additional training data, also finetuning the BERT encoder, hyperparameter tuning, etc.

[1] Karpathy, A. “nanoGPT”. GitHub 2023.

---

### Decision · Program_Chairs · 2023-09-21

**Decision:**

Accept (poster)

**Comment:**

This paper presents a novel and promising approach to Neural Architecture Search (NAS) by introducing a unifying search space design framework based on context-free grammars. The authors demonstrate the effectiveness of their approach by efficiently searching over large and expressive hierarchical search spaces and achieving superior results compared to existing NAS approaches. Overall, this paper is well-written and makes a significant contribution to the field of NAS. The reviewers are positive in the rebuttal period. I recommend to accept this paper.